# Boosting Revisited:
# Benchmarking and Advancing LP-Based Ensemble Methods

**Fabian Akkerman**[ID]                                                               *f.r.akkerman@utwente.nl*
*Industrial Engineering and Management Science*
*University of Twente*

**Julien Ferry**[ID]                                                                  *julien.ferry@polymtl.ca*
*CIRRELT & SCALE-AI Chair in Data-Driven Supply Chains*
*Polytechnique Montréal*

**Christian Artigues**[ID]                                                            *christian.artigues@laas.fr*
*LAAS-CNRS*
*Université de Toulouse*

**Emmanuel Hébrard**[ID]                                                              *hebrard@laas.fr*
*LAAS-CNRS*
*Université de Toulouse*

**Thibaut Vidal**[ID]                                                                 *thibaut.vidal@polymtl.ca*
*CIRRELT & SCALE-AI Chair in Data-Driven Supply Chains*
*Polytechnique Montréal*

**Reviewed on OpenReview:** *https://openreview.net/forum?id=lscC4PZUE4*

## Abstract

Despite their theoretical appeal, totally corrective boosting methods based on linear programming have received limited empirical attention. In this paper, we conduct the first large-scale experimental study of six LP-based boosting formulations, including two novel methods, NM-Boost and QRLP-Boost, across 20 diverse datasets. We evaluate the use of both heuristic and optimal base learners within these formulations, and analyze not only accuracy, but also ensemble sparsity, margin distribution, anytime performance, and hyperparameter sensitivity. We show that totally corrective methods can outperform or match state-of-the-art heuristics like XGBoost and LightGBM when using shallow trees, while producing significantly sparser ensembles. We further show that these methods can thin pre-trained ensembles without sacrificing performance, and we highlight both the strengths and limitations of using optimal decision trees in this context.

## 1 Introduction

Despite the surge in deep learning, ensemble methods remain state-of-the-art for tabular data (Borisov et al., 2024). This is evident in recent prediction competitions, where ensemble methods often outperform deep neural networks (Bojer & Meldgaard, 2021; Makridakis et al., 2022; 2024). Among them, boosting methods, such as Adaboost, XGBoost, and LightGBM, have become the default choice due to their efficiency, accuracy, and practical success across domains (Freund & Schapire, 1997; Chen & Guestrin, 2016; Ke et al., 2017). While these methods are widely adopted, they rely on greedy, stage-wise updates that often obscure the optimization principles behind their effectiveness.

A more theoretically grounded alternative is offered by linear programming (LP) based boosting methods, which formulate ensemble training as a global optimization problem. These methods cast training as a

linear program solved via column generation (Uchoa et al., 2024), where the goal is to optimize the weights of all base learners simultaneously. At each iteration, a new base learner is added to improve classification margins —the signed distances between examples and the ensemble decision boundary. From the perspective of decomposition techniques, column generation provides a principled framework to decouple the selection of base learners (solutions of pricing subproblems) from the global reweighting of the learner weights (solution of the master problem). These two subproblems are linked through dual variables, which emphasize misclassified points. This iterative reoptimization is referred to as *total correctiveness*. Notably, variations in the main objective, such as maximizing the minimum margin or minimizing the variance across margins, lead to different selection criteria for the base learners.

Although LP-based boosting was introduced over two decades ago, comprehensive empirical analysis remains limited. While some studies have explored multiple formulations or larger datasets, most remain either narrowly scoped or primarily theoretical. Few works offer a broad and systematic comparison of LP-based methods, and even fewer examine their behavior in depth across varying conditions. As a result, key questions remain unanswered: how do different formulations behave in practice, how do they compare to modern heuristic methods, and how are they affected by factors such as tree depth, margin objectives, and the use of optimal versus heuristic base learners?

In this work, we conduct the first large-scale empirical study of totally corrective boosting methods. Focusing on binary classification tasks, we compare six optimization-based methods, which we refer to as LP-based for brevity, although some involve quadratic programs (QPs). This includes two novel formulations introduced in this paper. Their performance is benchmarked against three state-of-the-art heuristic boosting baselines. Our evaluation spans 20 benchmark datasets and explores a wide range of settings, encompassing varying tree depths, margin objectives, and types of base learners—namely, heuristic CART trees and optimal decision trees. Decision trees are a natural choice in boosting due to their interpretability, and ensemble sparsity is particularly relevant: smaller ensembles with fewer trees are easier to interpret and faster at inference time. Beyond accuracy alone, our goal is to better understand the structure and properties of totally corrective methods. To that end, we examine anytime performance, ensemble sparsity, margin distributions, sensitivity to hyperparameters, and reweighting dynamics. Our findings provide new insights into why certain formulations generalize better and offer practical guidance for designing interpretable and effective ensemble models.

To summarize, our main contributions are:

- We conduct an extensive empirical study across twenty datasets, comparing six LP-based boosting methods and three state-of-the-art heuristic baselines.

- We analyze generalization performance, anytime behavior, margin distribution, ensemble sparsity, and hyperparameter sensitivity of LP-based boosting methods.

- We study the use of three different types of base learners within boosting frameworks: CART trees with hard voting, CART trees with soft voting through confidence scores, and optimal decision trees.

- We present two novel formulations that achieve state-of-the-art performance: one that focuses on the negative margins (misclassifications), and another that adapts an existing method by introducing a new quadratic regularization term.

- The full source code is available under an MIT license at `https://github.com/frakkerman/colboost`. All methods are implemented in the unified, user-friendly `colboost` Python library to promote more systematic empirical comparisons between totally corrective methods and facilitate future research in this field, see `https://pypi.org/project/colboost/`. Furthermore, all experimental results and datasets used in this study can be found at `https://doi.org/10.4121/f82dcdaa-fc94-43c5-b66d-02579bd3de4f`.

The remainder of this paper is organized as follows. Section 2 introduces notation and background on boosting approaches. Section 3 reviews related work on totally corrective methods and highlights our contributions. In Section 4, we present two novel totally corrective formulations. Section 5 details our experimental

design. The results are then reported in two parts, based on the type of base learner. Section 6 focuses on ensembles built with CART trees, using either hard voting or confidence-based soft voting. Section 7 then evaluates the use of optimal decision trees. Finally, Section 8 concludes this paper.

## 2 Preliminaries

Let $\mathcal{D} = \{(x_1, y_1), (x_2, y_2), \ldots, (x_M, y_M)\}$ be a dataset of size $M$ where each example (also called data point, or sample) $x_i \in \mathcal{X}$ has a binary label $y_i \in \{-1, 1\}$. Also, let $\mathbf{H} = \{h_1, h_2, \ldots, h_T\}$ be an ensemble of classifiers, such that $h_j : \mathcal{X} \mapsto \{-1, 1\}$, $\forall j \in \{1, \ldots, T\}$. Boosting methods usually associate weights $\mathbf{w} = \{w_1, w_2, \ldots, w_T\}$ to the different classifiers —also called *base learners* or *weak learners*— to minimize classification error. Base learners are trained iteratively such that each new classifier $h_j$ is fitted using sample weights $\mathbf{u}_j = \{u_{j1}, u_{j2}, \ldots, u_{jM}\}$, which highlight misclassified points. In column generation-based methods, the weights $\mathbf{w}$ are either computed directly (in primal) or derived from dual variables, depending on the formulation. While totally corrective boosting methods are able to recompute the base learners' weights $\mathbf{w}$ to maintain the best-performing ensemble at each iteration, popular heuristic methods typically compute the weight of each classifier only once and never update it in further iterations.

Adaboost, introduced in Freund & Schapire (1997), is a classical heuristic boosting method. Its most common version relies on the SAMME algorithm (Hastie et al., 2009). For a pre-specified number of iterations $T$, which specifies the number of trees that will be used in the ensemble, the SAMME algorithm iteratively does the following. First, (i) it trains a tree using sample weights $\mathbf{u}_j$ to weight datapoints for importance (in the first iteration, $u_{1i} \leftarrow \frac{1}{M}$ $\forall i \in \{1, \ldots, M\}$), next, (ii) the weighted error of the new tree $h_j$ is calculated, and (iii) the weight $w_j$ of the new tree in the ensemble voting process is determined based on its error. Afterwards, (iv) new sample weights $\mathbf{u}_{j+1}$ are calculated by updating the former ones $\mathbf{u}_j$ based on the misclassifications of the new tree $h_j$ and a pre-defined learning rate hyperparameter, and the process continues at step (i) until $T$ trees have been added to the ensemble.

Totally corrective methods are different from Adaboost in several ways. We present a high-level and generic overview to summarize these distinctions in Algorithm 1 (see also Shen et al. 2013). The key differences from Adaboost (and other heuristic boosting methods) are: (i) the ensemble size is not pre-defined, as the stopping criterion, based on the dual constraint, ensures that the optimal ensemble has been found, (ii) no intermediate error term is calculated to determine the weights $\mathbf{w}$ and $\mathbf{u}$, as both are outputs from the LP, and (iii) the weights $\mathbf{w}$ are re-computed at each LP solve (instead of only the weight of the new tree). These differences offer clear advantages over heuristic boosting methods like Adaboost, as the stopping criterion is well-defined ensuring finite termination at a global optimum, sample weights $\mathbf{u}$ are determined using the complete current ensembles' performance, and all base learners' weights $\mathbf{w}$ are re-determined, which yields ensembles that perform better while also being sparser (with fewer trees).

---

**Algorithm 1** High-level totally corrective boosting algorithm

---

**Require:** Convergence threshold $\epsilon$
1: $\forall i \in \{1..M\}, \quad u_{1i} \leftarrow \frac{1}{M}, \quad \beta \leftarrow 0$          ▷ Initialize the variables
2: $j \leftarrow 1$
3: **while** True **do**
4:      Fit new tree $h_j$ to training data using weights $\mathbf{u}_j$          ▷ Solve the pricing problem
5:      **if** $\sum_{i=1}^{M} y_i u_{ji} h_j(x_i) \leq \beta + \epsilon$ **then** break          ▷ Check for stopping criterion
6:      Add $h_j$ to the master problem
7:      Obtain new voting weights $w_{j' \in \{1 \ldots j\}}$ and compute $\mathbf{u}_{j+1}$          ▷ Solve the restricted master problem
8: **return** A convex combination of base learners and voting weights $\mathbf{w}$

---

## 3 Related Works

In this section, we focus on totally corrective boosting methods for training ensemble models. A foundational contribution comes from Grove & Schuurmans (1998) who, motivated by the success of Adaboost (Freund

& Schapire, 1997) and the hypothesis that its effectiveness arises from focusing on margins (Bartlett et al., 1998), propose the use of LP to minimize the worst-case margin. In classification, the margin typically quantifies the confidence of the ensemble's prediction: it is the difference between the cumulative vote assigned to the correct label and the largest cumulative vote assigned to any incorrect label (Bartlett et al., 1998). A negative margin indicates a misclassified example —one that lies on the wrong side of the ensemble's decision boundary. Grove & Schuurmans (1998) propose the following LP formulation (notation adapted for consistency):

$$\text{maximize}_{w,\rho} \quad \rho. \tag{1}$$

$$\text{subject to} \quad y_i \sum_{j=1}^{T} w_j h_j(x_i) \geq \rho, \quad \forall i = 1, \ldots, M, \tag{2}$$

$$\sum_{j=1}^{T} w_j = 1, \tag{3}$$

$$w_j \geq 0, \quad \forall j = 1, \ldots, T, \tag{4}$$

where $\rho$ is the minimum margin, the term $y_i \sum_{j=1}^{T} w_j h_j(x_i)$ is positive when the current ensemble votes correctly for data point $(x_i, y_i)$, and negative when the ensemble vote is incorrect. The sum of weak learner weights is bounded by 1. Solving this problem using a column generation framework provides two main advantages: (i) it allows leveraging the dual solution to identify new weak learners at each iteration, and (ii) it offers a well-defined stopping criterion. The corresponding dual problem is given by:

$$\text{minimize}_{u,\beta} \quad \beta. \tag{5}$$

$$\text{subject to} \quad \sum_{i=1}^{M} u_i y_i h_j(x_i) \leq \beta, \quad \forall j = 1, \ldots, T, \tag{6}$$

$$\sum_{i=1}^{M} u_i = 1, \tag{7}$$

$$0 \leq u_i, \quad \forall i = 1, \ldots, M. \tag{8}$$

In this dual formulation, Constraint (6) defines the misclassification costs $u_i$ for each example $i$, emphasizing those with low or negative margins. Training a weak learner corresponds to finding a violated constraint (6) and a new variable $w_j$ of positive reduced cost. Furthermore, when no base learner satisfies $\sum_{j=1}^{M} u_i y_i h_j(x_i) > \beta$, the current combined classifier represents the optimal solution among all possible linear combinations of base learners. Despite these properties, Grove & Schuurmans (1998) report unstable empirical performance for this LP. Demiriz et al. (2002) address several of the issues by using a soft-margin LP variant, i.e., introducing a slack variable in Constraint (2). This adjustment improves robustness: unlike the hard-margin formulation, which can be highly degenerate when the number of weak learners is small relative to the number of data points, the soft-margin version reduces such degeneracies and is less sensitive to noisy or outlier data. The resulting formulation —known as **LP-Boost**— alleviates many limitations of the original hard-margin LP, though, as we will demonstrate in our experiments, it does not resolve all of them. In the remainder of this section, we discuss the contributions to ensemble learning using linear programming after LP-Boost (Grove & Schuurmans, 1998; Demiriz et al., 2002).

Theoretical analyses have identified fundamental limitations of LP-based boosting and guided the design of improved formulations. Shen & Li (2009) analyze boosting algorithms through the lens of their Lagrange duals, drawing connections between LP-Boost and entropy-maximization frameworks. They show that Adaboost, LogitBoost, and soft-margin LP-Boost can be viewed as entropy-regularized variants of the hard-margin LP-Boost formulation. A key insight from their analysis is that LP-Boost, despite its focus on maximizing the minimum margin, often exhibits inferior generalization performance compared to algorithms that optimize for average margin or margin distribution, such as Adaboost. The authors argue that entropy regularization, which enforces a more uniform weight distribution on training examples, contributes to Adaboost's superior generalization performance, especially in noisy or non-separable datasets.

This work provides theoretical insights into why LP-Boost's strict focus on the minimum margin may limit its effectiveness in practical settings, a conclusion further supported by the formal results in Gao & Zhou (2013).

Building on earlier theoretical insights, several works have proposed totally corrective variants of LP-Boost to improve both convergence and generalization. Warmuth et al. (2006) propose Total-Boost, which incorporates entropic regularization and adaptive margin constraints to achieve logarithmic convergence guarantees. The authors experimentally demonstrate that Total-Boost often requires significantly fewer iterations than LP-Boost, particularly in high-dimensional or redundant feature spaces, while producing smaller ensembles, making it advantageous for feature selection tasks. Total-Boost serves as a precursor to Soft-Boost (Rätsch et al., 2007), which extends the approach to the non-separable case by minimizing the relative entropy to the initial distribution, subject to linear constraints on the edges of all previously generated base learners —where the edge is defined as the weighted difference between correctly and incorrectly classified examples. Rätsch et al. (2007) show that LP-Boost may require up to $\mathcal{O}(M)$ iterations to converge in the worst case, particularly when the problem structure induces linear dependencies in the optimization process. In contrast, Soft-Boost's use of capping constraints and entropy updates leads to provably faster, logarithmic convergence, making it a potentially more efficient alternative in such settings. However, its conservative constraint-tightening can result in worse early-stage generalization performance.

To address this, Warmuth et al. (2008) introduce Entropy Regularized LP-Boost (**ERLP-Boost**), which modifies LP-Boost by incorporating a scaled relative entropy term into the objective. ERLP-Boost matches the performance of LP-Boost and Soft-Boost but offers faster early error reduction and guarantees convergence within $\mathcal{O}(\frac{1}{\epsilon^2} \ln \frac{M}{C})$ iterations, with $C$ being a tunable parameter. Additionally, Warmuth et al. (2006) observe that LP-Boost's empirical performance is sensitive to the choice of the solver: interior-point methods achieve faster convergence than simplex methods in certain scenarios.

More recently, Mitsuboshi et al. (2022) offer a unified view of LP-based boosting methods by framing LP-Boost and ERLP-Boost as instances of the Frank-Wolfe algorithm. They highlight a key trade-off: LP-Boost has low per-iteration cost but may require many iterations, while ERLP-Boost converges in fewer steps at a higher computational cost per iteration. To balance these strengths, they propose MLP-Boost, a hybrid algorithm that alternates between LP-Boost and ERLP-Boost steps. Experiments show that MLP-Boost retains the convergence guarantees of ERLP-Boost while achieving runtime comparable to LP-Boost.

Building on the limitations of focusing only on the minimum margin, a line of research has emerged to optimize the *entire* margin distribution. Shen & Li (2010) introduce **MD-Boost**, a totally corrective boosting algorithm that simultaneously maximizes the average margin and minimizes margin variance. Extending LP-Boost, MD-Boost retains the column generation approach but adopts a quadratic programming formulation, enabling it to overcome limitations inherent to linear programming methods. Empirical results show that MD-Boost outperforms LP-Boost in generalization performance on several benchmark datasets. The authors highlight that MD-Boost's ability to consider the full margin distribution, rather than focusing solely on the minimum margin like LP-Boost, contributes to its robustness and improved classification accuracy. In a similar spirit, Roy et al. (2016) propose Cq-Boost, a column generation algorithm that minimizes the PAC-Bayesian $\mathcal{C}$-bound, explicitly accounting for both the mean and variance of the margin distribution. Unlike LP-Boost, which focuses on minimizing the margin of misclassified points, Cq-Boost leverages quadratic programming to optimize the margin distribution while achieving sparser ensembles. Empirical results show that Cq-Boost outperforms LP-Boost in terms of accuracy and sparsity.

Bi et al. (2004) extend LP-Boost by developing **CG-Boost**, a column-generation boosting framework for constructing sparse mixture-of-kernels models. By 2-norm regularization, CG-Boost produces models that balance expressiveness and sparsity, achieving improved generalization and reduced testing time compared to single-kernel or composite-kernel methods. The approach generalizes LP-Boost to quadratic programs, enabling its application to a broader range of learning formulations. Experiments demonstrate CG-Boost's effectiveness in achieving high accuracy with fewer basis functions, outperforming standard composite-kernel methods in both performance and efficiency. Shen et al. (2013) propose CGBoost (not to be confused with CG-Boost by Bi et al. 2004), a fully corrective boosting framework that generalizes LP-Boost to accommodate arbitrary convex loss functions and regularization terms (e.g., $\ell_1$, $\ell_2$, and $\ell_\infty$-norms). Unlike LP-Boost,

which solves the dual problem, CGBoost focuses on the primal problem, offering simpler optimization with faster convergence. The authors demonstrate that CGBoost effectively balances sparsity and generalization, outperforming LP-Boost in terms of computational efficiency while maintaining or improving classification accuracy across various benchmark datasets.

To address class imbalance, Datta et al. (2020) propose LexiBoost, a lexicographic programming-based boosting framework that eliminates the need for manual cost tuning. LexiBoost solves a two-stage sequence of linear programs, first minimizing hinge loss separately for each class, then minimizing the deviation from these losses to balance misclassification rates. The dual formulation further adapts instance weights dynamically. The approach is scalable to multi-class settings and demonstrates strong performance across imbalanced datasets, hyperspectral images, and ImageNet subsets, outperforming traditional cost-sensitive boosting methods. Aziz et al. (2024) propose a dynamic data-reduction ensemble learning algorithm (DDA) that extends LP-Boost by incorporating dynamic data selection, phased learning, and nonlinear loss functions. Unlike standard LP-Boost, which considers the entire training dataset in every iteration, DDA uses sparse dual solutions to identify "active data points" in a bootstrap fashion, focusing computation on subsets of critical examples to improve efficiency. The algorithm operates in three phases: initialization (bootstrapping base learners), generation (error-based sampling without full master problem inputs), and refinement (iteratively adding base learners based on active data subsets). By incorporating nonlinear loss functions and explicitly promoting diversity, DDA reduces generalization error while maintaining or improving computational efficiency. Experiments show that DDA outperforms standard LP-Boost by achieving higher accuracy and better diversity in ensemble models.

Other work has adapted LP-Boosting for specific application domains or different base learners. Hinrichs et al. (2009) propose spatially augmented LP-Boosting, adding spatial smoothness constraints to promote contiguous regions in medical imaging data. Applied to the ADNI dataset, the method improves classification accuracy and interpretability compared to standard LP-Boost, achieving better generalization performance. Aglin et al. (2021) investigate optimal forests of decision trees by integrating optimal decision tree (ODT) learning into LP-Boost and MD-Boost frameworks. Their method, OptiBoost, uses a column generation process with ODTs to guarantee optimality for both LP-Boost and MD-Boost formulations. Experiments demonstrate that optimizing the entire margin distribution with MD-Boost often yields better generalization compared to maximizing the minimum margin in LP-Boost. The study highlights that incorporating ODTs into boosting frameworks not only improves optimization objectives but also leads to more accurate and sparse models, outperforming heuristic approaches on certain datasets.

To sum up, most works on totally corrective boosting propose formulations that optimize the margins of base learner ensembles. However, key aspects of LP-based boosting —including the ensemble accuracy-sparsity trade-off, anytime behavior, and sensitivity to hyperparameters— remain underexplored. In this work, we address these gaps through a comprehensive empirical study across twenty datasets, comparing six LP-based boosting methods against three state-of-the-art heuristic baselines. Our results provide a systematic understanding of how margin formulations affect not only final accuracy but also convergence speed, sparsity, and hyperparameter sensitivity. In addition, we study the influence of weak learner choice —comparing heuristic CART trees with either hard voting or soft (confidence-based) voting, and optimal decision trees— on boosting performance. We also introduce two novel formulations. The first one explicitly models negative margins, allowing direct control of the trade-off between generalization performance and training accuracy through a regularization hyperparameter. The second applies quadratic regularization to improve ensemble stability. Together, these contributions offer both practical insights and theoretical extensions that strengthen the foundation of LP-based boosting research.

## 4 New Totally Corrective Formulations

We introduce two novel boosting formulations that build on existing margin-based approaches. The first focuses on a previously underexplored aspect of the margin distribution, the negative margins. This formulation, detailed in Section 4.1, directly yields the ensemble weights $\mathbf{w}$ from the primal solution, while the sample weights $\mathbf{u}$ are derived from the dual. The second formulation, presented in Section 4.2, introduces

regularization on the sample weights $\mathbf{u}$, which are computed directly in the primal to keep the quadratic term in the objective. In this case, the ensemble weights $\mathbf{w}$ are recovered from the dual solution.

## 4.1 Negative Margins Boosting

The idea behind this formulation, which we coin Negative Margins Boost (NM-Boost), is to maximize the sum of all margins while assigning greater weight to negative margins in the objective. A coefficient $C$ controls the trade-off between reducing negative margins (which serve as a proxy for misclassifications) and increasing the overall sum of margins (which promotes better generalization). NM-Boost avoids the limitations of earlier approaches that focus solely on the worst-case margin —often overly sensitive to outliers— as well as the drawbacks of LP-Boost, where the margin of each example is decomposed into a "slack" and a worst-case margin, making the contribution of individual examples difficult to interpret.

In the primal formulation shown below, variables $\rho_i^{neg}$ represent the negative part of the individual margins: $\rho_i^{neg} = 0$ if example $i$ is correctly classified, and $\rho_i^{neg}$ is the (negative) margin $\rho_i$ between the ensemble's prediction for misclassified example $i$ and the decision boundary (minus a small offset) otherwise. Negative margins hence receive greater penalization in the objective function. This explicit modeling ensures that misclassified examples (with negative margins) directly influence the optimization, rather than being absorbed implicitly in the overall margin sum. In practice, they are computed as the negative part of the margins offset by a constant term $\frac{1}{T}$. This offset ensures a clear distinction between zero and strictly positive margins, since in the former case, an example may not be confidently classified —depending on the tie-breaking policy used. Note that this offset becomes tight when all trees contribute equally to the vote.[1] The optimization problem of NM-Boost is formalized as:

$$\text{maximize}_{w,\rho,\rho^{neg}} \quad \sum_{i=1}^{M} \rho_i^{neg} + C \sum_{i=1}^{M} \rho_i. \tag{9}$$

$$\text{subject to} \quad y_i \sum_{j=1}^{T} w_j h_j(x_i) \geq \rho_i, \quad \forall i = 1, \ldots, M, \tag{10}$$

$$\rho_i^{neg} \leq 0, \quad \forall i = 1, \ldots, M, \tag{11}$$

$$\rho_i^{neg} \leq \rho_i - \frac{1}{T}, \quad \forall i = 1, \ldots, M, \tag{12}$$

$$\sum_{j=1}^{T} w_j = 1, \quad w_j \geq 0, \quad \forall j = 1, \ldots, T. \tag{13}$$

Because (9) is an LP with $\mathbf{w}$ constrained to the probability simplex, an optimal solution can be chosen to place weight on only a subset of base learners; in practice, many $w_j$ are zero.

## 4.2 Quadratically Regularized LP-Boost

We also propose a boosting formulation with a quadratic regularization term, which we call QRLP-Boost. The formulation builds on prior work by Warmuth et al. (2006), Rätsch et al. (2007), and Warmuth et al. (2008). More precisely, we modify the objective function of ERLP-Boost, whose formulation is provided in the Appendix B.3. The original ERLP-Boost ensures stability through entropy-based smoothing. Moreover, it requires iterative refinement of the sample weight distribution, repeatedly computing KL-divergence adjustments and stabilizing small weight changes. In contrast, our proposed QRLP-Boost introduces a direct quadratic regularization that blends entropy and variance-like penalties, enabling larger, smoother updates within a single optimization step. This leads to a more numerically stable process, as the regularization naturally controls weight shifts without requiring repeated re-optimization or entropy-based corrections.

Our approach differs from ERLP-Boost in the following ways: (i) instead of minimizing a KL-divergence penalty, we use a combined entropy–quadratic regularization term that stabilizes updates while offering more

---

[1]For instance, this is usually not the case within a confidence-based soft voting scheme.

flexibility in reweighting training examples, (ii) the weight distribution is updated directly in a single QP solve per iteration, avoiding multiple re-solves with adjusted entropy terms, and (iii) convergence is checked based on the QP objective, eliminating the need for iterative entropy-based margin estimates or auxiliary thresholds. QRLP-Boost therefore solves the following convex optimization problem:

$$\text{minimize}_{u,\xi} \quad \sum_{i=1}^{M} \xi_i + \frac{1}{\eta} \sum_{i=1}^{M} \left( u_i \log u_i^0 + \frac{u_i^2}{2u_i^0} \right). \tag{14}$$

$$\text{subject to} \quad \sum_{i=1}^{M} u_i y_i h_j(x_i) \le \xi_i, \quad \forall j = 1, \dots, T, \tag{15}$$

$$\sum_{i=1}^{M} u_i = 1, \tag{16}$$

$$0 \le u_i \le \frac{1}{C}, \quad \forall i = 1, \dots, M. \tag{17}$$

Here, $\eta$ is defined as $\max(0.5, \ln M / \frac{1}{2}\epsilon^{\text{stop}})$, where $\epsilon^{\text{stop}}$ is a small constant. The term $u^0$ denotes the initial sample weight distribution, typically uniform. The quadratic regularizer $u_i^2/2u_i$ replaces the KL-divergence term in ERLP-Boost, serving to penalize sharp deviations from the initial weights. The quadratic component of the regularizer (with $u_i^0 > 0$) makes the objective strictly convex in $\mathbf{u}$, hence the minimizing $\mathbf{u}$ is unique. We solve the program directly with a standard convex optimizer; no generic closed-form expression is available. All other constraints are retained from the original ERLP-Boost formulation (see Appendix B.3).

## 5 Experimental Design

To obtain robust and generalizable insights into boosting with column generation methods, we conduct an extensive empirical study across 20 diverse datasets from widely used benchmark repositories in machine learning. These datasets were selected to ensure diversity in terms of size, feature dimensionality, and class balance. A complete description of the datasets is given in Appendix A.

Our study includes the following totally corrective methods, evaluated in the original versions proposed by their respective authors: LP-Boost (Demiriz et al., 2002), CG-Boost (Bi et al., 2004), ERLP-Boost (Warmuth et al., 2008), MD-Boost (Shen & Li, 2010), as well as our two novel formulations, NM-Boost and QRLP-Boost (introduced in Section 4). For LP-Boost, we adopt the formulation from Equation 2 in Demiriz et al. (2002), as we found the alternative version (Equation 4) to be highly sensitive to its hyperparameters and more prone to degeneracy. The detailed mathematical formulations of all totally corrective methods are provided in Appendix B. While we initially included Cq-Boost (Roy et al., 2016) in our experiments, we did not retain it in our final analyses for the sake of brevity, as it did not provide additional insights or accuracy improvements over other methods. For comparison, we also include three widely used heuristic boosting baselines: Adaboost (Freund & Schapire, 1997), XGBoost (Chen & Guestrin, 2016), and LightGBM (Ke et al., 2017).

To ensure a fair comparison between the different methods, we conduct hyperparameter tuning for which we vary the trade-off parameter $C$ (totally corrective methods) and the learning rate (heuristic boosting methods) across 10 possible values. Appendix C details the hyperparameter ranges considered for each method, consistent with the original papers' recommendations and refined based on our preliminary experiments. As each method involves tuning a single hyperparameter only, an exhaustive sweep over a fixed set of values is straightforward and ensures that we reliably measure performance across the relevant range.

Each dataset is split into 60% for training, 20% for validation, and 20% for testing. We select the best-performing hyperparameter value in terms of accuracy using the validation data, and report only the results on the testing data. For all results, we report averages over five random seeds, accounting for variability in data splits and method-specific randomness. We impose a time limit of 45 minutes and an iteration limit of 100 for all the methods. Early stopping is explicitly disabled for totally corrective approaches to ensure that all boosting strategies are evaluated under a comparable number of iterations. This setup ensures

fairness, as each method is allowed to construct ensembles with the same number of base learners. In practice, the iteration limit is reached much more frequently than the time limit, which is seldom binding. In Appendix D.1, we report the computational times per dataset for all studied methods. On average, the totally corrective methods require between 8 and 15 minutes of training time, whereas the heuristic variants typically finish within seconds. Although this represents a noticeable difference, the running times remain modest and well within practical limits for offline training.

All code is written in Python and experiments are executed on a Linux cluster using Python 3.11. Mathematical programs are solved using Gurobi 10.0.1. We rely on the `scikit-learn` library (Pedregosa et al., 2011) to build CART trees and fit Adaboost ensembles, and use the `Blossom` library for optimal decision trees (Demirović et al., 2023). XGBoost and LightGBM ensembles are fitted using their respective Python libraries, see Chen & Guestrin (2016) and Ke et al. (2017), respectively. We run individual experiments on a single thread of a compute node equipped with AMD Genoa 9654 cores @2.4GHz along with 2 GB of memory per thread. All totally corrective methods are implemented within the unified, user-friendly `colboost` Python library, available under an MIT license at `https://pypi.org/project/colboost/`.

The remaining experimental sections in this paper are organized as follows. Section 6 analyzes the results of ensembles built using CART trees with either hard or confidence-based soft voting. Section 7 then focuses on the use of optimal decision trees in boosting ensembles.

# 6 Experiments with CART Trees as Base Learners

In this section, we conduct experiments using CART trees as base learners —the default choice in most boosting applications due to their computational efficiency and reasonably strong predictive performance, despite being heuristic rather than optimal. Our goal is to provide a comprehensive empirical analysis of totally corrective boosting methods, comparing different mathematical formulations with each other and with the state-of-the-art heuristic benchmarks. To this end, we evaluate multiple facets of model behavior: accuracy-sparsity performance (Section 6.1), anytime performance (Section 6.2), ensemble sparsity (Section 6.3), margin distributions (Section 6.4), and sensitivity to hyperparameters (Section 6.5). We also include an experiment, in Section 6.6, for which we obtain an ensemble using an external method (Adaboost) and next let the totally corrective methods reweight the complete ensemble in a single shot. All main experiments are conducted using a hard voting scheme (i.e., each tree casts a vote of either -1 or +1). In Section 6.7, we investigate the impact of adopting a soft voting mechanism, in which trees contribute confidence-weighted scores (i.e., each tree votes in the domain $[-1, 1]$). Throughout all experiments, we vary the maximum tree depth between 1 (decision stumps), 3, 5, and 10. While decision stumps are commonly used in boosting due to their high bias and low variance, we broaden the scope to thoroughly compare performance across varying tree complexities.

## 6.1 Accuracy-Sparsity Performance

We evaluate the trained ensembles using two metrics: testing accuracy and number of columns (base learners) used. The latter serves as a measure of sparsity, indicating how many trees contribute non-zero weight to the ensemble's prediction. Sparser ensembles are generally preferred, as this improves interpretability and reduces inference time.

Figure 1 summarizes all results by showing the average testing accuracy and sparsity across all considered datasets[2], for each method and different tree depths. For depth 1 (decision stumps), we observe that NM-Boost, QRLP-Boost, CG-Boost, and XGBoost achieve comparable accuracy. However, NM-Boost matches or slightly outperforms the others while using significantly fewer trees. MD-Boost yields the sparsest ensembles but suffers from the lowest accuracy. Interestingly, for decision stumps (which are often the default weak learners of boosting approaches), totally corrective methods outperform the heuristic baselines (Adaboost, XGBoost, and LightGBM) in both accuracy and sparsity. At depths 3 and 5, some totally corrective methods

---

[2]While averaging across datasets may seem counterintuitive, since the resulting metrics lack a clear, interpretable meaning for any individual dataset, it is a widely used approach for summarizing and comparing methods (see, e.g., Freund & Schapire 1996, Demirović et al. 2023).

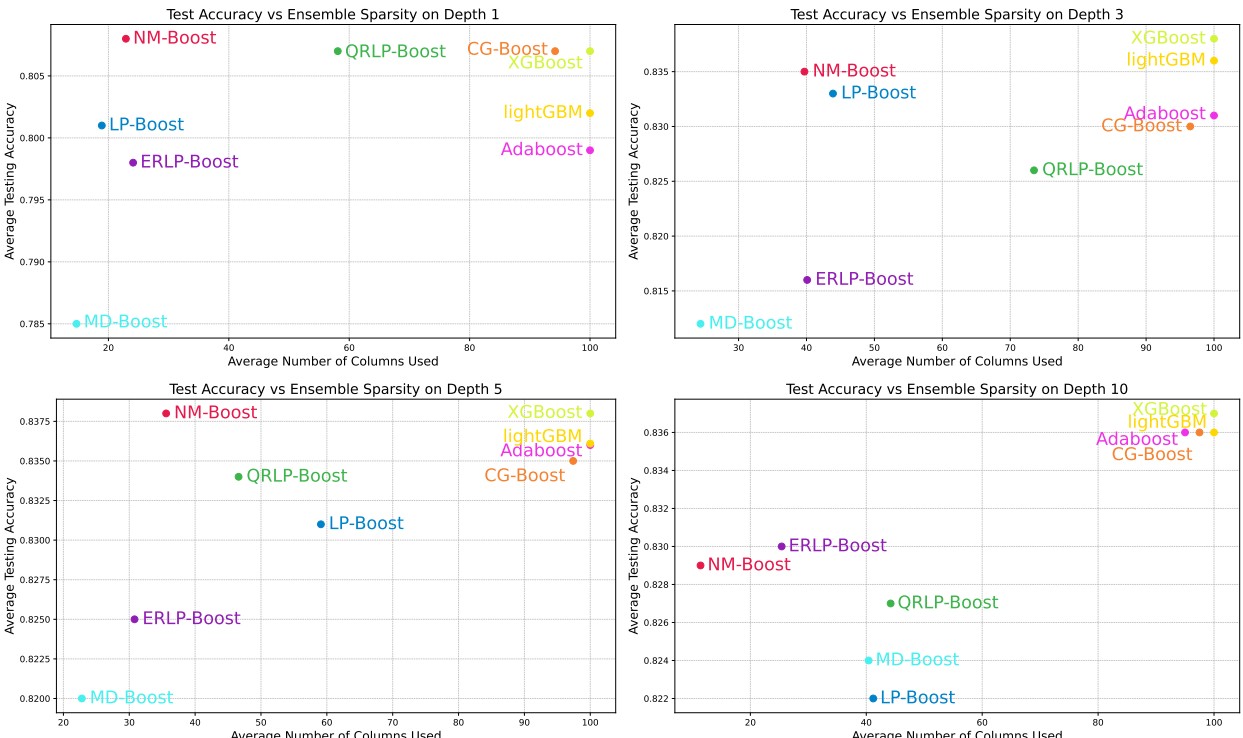

Figure 1: Average testing accuracy compared to average ensemble sparsity over **all datasets** for CART decision trees of depth 1, 3, 5, and 10.

(NM-Boost, QRLP-Boost, LP-Boost) exhibit close or similar accuracy as the heuristic benchmarks but use fewer trees, or find similar accuracy with the same number of trees (CG-Boost). For depth 10, the individual trees are strong and specialized, and using them all is the best policy. As a result, totally corrective methods fall short in accuracy, though CG-Boost remains competitive, partly due to its use of denser ensembles.

**Observation 1** *With shallow to moderately deep CART trees, totally corrective methods match or exceed the test accuracy of heuristic baselines while yielding significantly sparser ensembles.*

**Observation 2** *Among all totally corrective methods, our proposed NM-Boost achieves the best performance when using shallow to moderately deep CART trees.*

**Observation 3** *With deeper CART trees, a trade-off becomes apparent: heuristic methods achieve generally higher accuracy, but at the expense of sparsity.*

To allow for a more nuanced assessment of each method's strengths and weaknesses, we also report per-dataset results. Tables 1 and 2 detail the testing accuracy and sparsity across all considered datasets, totally corrective methods, and heuristic benchmarks for CART decision tree base learners of depth 1 and 5, respectively. Results for the other depth values are provided in the Appendix D.2, and are consistent with our main observations.

Focusing on depth-1 trees (Table 1), we observe that the superiority of totally corrective methods over state-of-the-art heuristic boosting approaches, which was previously observed on aggregate results, holds consistently across datasets. The proposed NM-Boost and QRLP-Boost are particularly competitive in this regime.

**Observation 4** *With decision stumps base learners, the totally corrective boosting methods consistently outperform or match the benchmarked heuristic approaches on 19 out of 20 datasets.*

Table 1: Testing accuracy and number of non-zero weights for different boosting methods using CART trees of **depth 1**, averaged over five seeds. **Bold** highlights the best *overall* accuracy, while a star* marks the best among *totally corrective* methods. The last row shows the mean and median for both statistics.

| Dataset | NM-Boost Acc. | Cols. | QRLP-Boost Acc. | Cols. | LP-Boost Acc. | Cols. | CG-Boost Acc. | Cols. | ERLP-Boost Acc. | Cols. | MD-Boost Acc. | Cols. | Adaboost Acc. | Cols. | XGBoost Acc. | Cols. | LightGBM Acc. | Cols. |
|---|---|---|---|---|---|---|---|---|---|---|---|---|---|---|---|---|---|---|
| banana | **0.625*** ± 0.016 | 5 | 0.620 ± 0.016 | 16 | **0.625*** ± 0.016 | 1 | **0.625*** ± 0.016 | 100 | 0.620 ± 0.016 | 16 | 0.620 ± 0.016 | 14 | **0.625** ± 0.016 | 100 | 0.620 ± 0.016 | 100 | 0.620 ± 0.016 | 100 |
| breast cancer | **0.819*** ± 0.049 | 5 | 0.800 ± 0.068 | 20 | 0.789 ± 0.053 | 4 | 0.789 ± 0.042 | 100 | 0.811 ± 0.053 | 17 | 0.808 ± 0.038 | 10 | 0.777 ± 0.045 | 100 | 0.804 ± 0.041 | 100 | **0.819** ± 0.059 | 100 |
| diabetes | **0.775*** ± 0.013 | 4 | 0.756 ± 0.031 | 98 | 0.742 ± 0.024 | 19 | 0.747 ± 0.028 | 100 | 0.770 ± 0.037 | 28 | 0.755 ± 0.023 | 10 | 0.769 ± 0.020 | 100 | 0.761 ± 0.022 | 100 | 0.768 ± 0.018 | 100 |
| german credit | 0.795 ± 0.020 | 16 | **0.808*** ± 0.024 | 27 | 0.800 ± 0.015 | 26 | 0.807 ± 0.012 | 100 | 0.801 ± 0.021 | 24 | 0.754 ± 0.011 | 11 | 0.798 ± 0.017 | 100 | 0.804 ± 0.015 | 100 | 0.806 ± 0.015 | 100 |
| heart | 0.841 ± 0.043 | 26 | 0.822 ± 0.019 | 34 | 0.811 ± 0.043 | 12 | **0.848*** ± 0.014 | 100 | 0.807 ± 0.028 | 11 | 0.804 ± 0.036 | 12 | 0.819 ± 0.032 | 100 | 0.789 ± 0.042 | 100 | 0.833 ± 0.023 | 100 |
| image | **0.851*** ± 0.009 | 20 | 0.844 ± 0.010 | 40 | 0.834 ± 0.015 | 11 | 0.844 ± 0.006 | 100 | 0.804 ± 0.006 | 8 | 0.733 ± 0.047 | 7 | 0.805 ± 0.012 | 100 | 0.840 ± 0.006 | 100 | 0.835 ± 0.011 | 100 |
| ringnorm | 0.930 ± 0.003 | 60 | **0.931*** ± 0.004 | 62 | 0.927 ± 0.004 | 58 | 0.929 ± 0.003 | 100 | 0.883 ± 0.008 | 20 | 0.877 ± 0.007 | 16 | 0.894 ± 0.006 | 100 | 0.929 ± 0.005 | 100 | 0.926 ± 0.003 | 100 |
| solar flare | 0.703 ± 0.047 | 7 | 0.710 ± 0.077 | 16 | 0.683 ± 0.091 | 2 | 0.738* ± 0.071 | 100 | 0.724 ± 0.058 | 10 | 0.683 ± 0.051 | 10 | **0.759** ± 0.031 | 100 | 0.724 ± 0.072 | 100 | 0.628 ± 0.096 | 100 |
| splice | 0.943 ± 0.009 | 55 | 0.943 ± 0.006 | 92 | 0.943 ± 0.010 | 31 | 0.946* ± 0.005 | 100 | 0.932 ± 0.011 | 16 | 0.918 ± 0.013 | 9 | 0.942 ± 0.010 | 100 | **0.947** ± 0.006 | 100 | 0.943 ± 0.008 | 100 |
| thyroid | 0.940 ± 0.035 | 5 | **0.949*** ± 0.031 | 8 | 0.944 ± 0.032 | 3 | 0.944 ± 0.032 | 100 | 0.944 ± 0.032 | 4 | 0.921 ± 0.024 | 8 | 0.940 ± 0.032 | 100 | 0.935 ± 0.034 | 100 | 0.898 ± 0.019 | 100 |
| titanic | 0.806* ± 0.015 | 11 | 0.790 ± 0.019 | 93 | 0.796 ± 0.021 | 3 | 0.797 ± 0.019 | 100 | 0.793 ± 0.026 | 16 | 0.801 ± 0.030 | 13 | 0.802 ± 0.035 | 100 | **0.808** ± 0.026 | 100 | 0.806 ± 0.031 | 100 |
| twonorm | **0.961*** ± 0.003 | 60 | 0.960 ± 0.004 | 62 | **0.961*** ± 0.004 | 57 | **0.961*** ± 0.004 | 100 | 0.924 ± 0.004 | 25 | 0.950 ± 0.004 | 43 | 0.945 ± 0.002 | 100 | 0.960 ± 0.004 | 100 | 0.954 ± 0.004 | 100 |
| waveform | **0.889*** ± 0.007 | 56 | 0.885 ± 0.008 | 65 | **0.889*** ± 0.005 | 35 | 0.888 ± 0.006 | 100 | 0.870 ± 0.006 | 20 | 0.850 ± 0.003 | 11 | 0.880 ± 0.007 | 100 | 0.888 ± 0.005 | 100 | 0.886 ± 0.004 | 100 |
| adult | 0.814 ± 0.003 | 16 | **0.816*** ± 0.002 | 39 | 0.814 ± 0.003 | 14 | 0.814 ± 0.003 | 80 | **0.816*** ± 0.002 | 32 | 0.814 ± 0.002 | 14 | 0.815 ± 0.002 | 100 | **0.816** ± 0.002 | 100 | **0.816** ± 0.002 | 100 |
| compas | 0.660 ± 0.016 | 7 | 0.671 ± 0.018 | 28 | 0.660 ± 0.013 | 5 | 0.660 ± 0.013 | 100 | 0.671 ± 0.018 | 28 | **0.673*** ± 0.016 | 13 | 0.671 ± 0.016 | 100 | 0.671 ± 0.016 | 100 | 0.669 ± 0.017 | 100 |
| employment CA2018 | 0.740 ± 0.003 | 16 | **0.747*** ± 0.002 | 83 | 0.740 ± 0.003 | 17 | 0.740 ± 0.003 | 100 | 0.744 ± 0.002 | 30 | 0.734 ± 0.003 | 9 | 0.723 ± 0.004 | 100 | **0.747** ± 0.002 | 100 | 0.746 ± 0.002 | 100 |
| employment TX2018 | 0.744 ± 0.003 | 18 | **0.755*** ± 0.003 | 86 | 0.744 ± 0.003 | 18 | 0.744 ± 0.003 | 76 | 0.749 ± 0.002 | 29 | 0.731 ± 0.005 | 8 | 0.731 ± 0.003 | 100 | 0.753 ± 0.004 | 100 | 0.751 ± 0.002 | 100 |
| public coverage CA2018 | 0.680 ± 0.006 | 4 | **0.704*** ± 0.003 | 88 | 0.680 ± 0.005 | 3 | 0.680 ± 0.006 | 96 | **0.704*** ± 0.003 | 72 | **0.704*** ± 0.003 | 54 | 0.695 ± 0.004 | 100 | 0.701 ± 0.004 | 100 | 0.700 ± 0.004 | 100 |
| public coverage TX2018 | 0.834 ± 0.008 | 4 | 0.849* ± 0.003 | 100 | 0.829 ± 0.002 | 3 | 0.829 ± 0.002 | 100 | 0.848 ± 0.003 | 52 | 0.847 ± 0.002 | 7 | 0.844 ± 0.003 | 100 | **0.850** ± 0.003 | 100 | 0.849 ± 0.002 | 100 |
| mushroom secondary | **0.817*** ± 0.004 | 62 | 0.787 ± 0.003 | 65 | 0.816 ± 0.004 | 56 | 0.816 ± 0.005 | 71 | 0.752 ± 0.003 | 23 | 0.726 ± 0.003 | 15 | 0.739 ± 0.002 | 100 | 0.784 ± 0.004 | 100 | 0.778 ± 0.004 | 100 |
| Mean/Median | **0.808***/**0.815*** | 22.9/16 | 0.807/0.804 | 58.1/63.5 | 0.801/0.806 | 18.9/13 | 0.807/0.810 | 94.2/100 | 0.798/0.802 | 24.1/21.5 | 0.785/0.778 | 14.7/11 | 0.799/0.800 | 100/100 | 0.807/0.804 | 100/100 | 0.802/0.811 | 100/100 |

**Observation 5** *With decision stumps base learners, NM-Boost and QRLP-Boost achieve the best performances of all totally corrective methods, with NM-Boost producing sparser ensembles.*

Table 2: Testing accuracy and number of non-zero weights for different boosting methods using CART trees of **depth 5**, averaged over five seeds. **Bold** highlights the best *overall* accuracy, while a star* marks the best among *totally corrective* methods. The last row shows the mean and median for both statistics.

| Dataset | NM-Boost Acc. | Cols. | QRLP-Boost Acc. | Cols. | LP-Boost Acc. | Cols. | CG-Boost Acc. | Cols. | ERLP-Boost Acc. | Cols. | MD-Boost Acc. | Cols. | Adaboost Acc. | Cols. | XGBoost Acc. | Cols. | LightGBM Acc. | Cols. |
|---|---|---|---|---|---|---|---|---|---|---|---|---|---|---|---|---|---|---|
| banana | **0.670*** ± 0.014 | 12 | **0.670*** ± 0.014 | 1 | **0.670*** ± 0.014 | 1 | **0.670*** ± 0.014 | 100 | **0.670*** ± 0.014 | 2 | **0.670*** ± 0.014 | 48 | 0.670 ± 0.014 | 100 | 0.670 ± 0.014 | 100 | 0.670 ± 0.014 | 100 |
| breast cancer | **0.875*** ± 0.028 | 9 | 0.838 ± 0.041 | 15 | 0.830 ± 0.027 | 20 | 0.819 ± 0.031 | 100 | 0.815 ± 0.042 | 11 | 0.826 ± 0.060 | 8 | 0.811 ± 0.041 | 100 | 0.819 ± 0.019 | 100 | 0.830 ± 0.032 | 100 |
| diabetes | 0.747 ± 0.012 | 29 | 0.753* ± 0.020 | 32 | 0.714 ± 0.020 | 67 | 0.738 ± 0.023 | 100 | 0.740 ± 0.025 | 20 | 0.734 ± 0.008 | 14 | 0.757 ± 0.009 | 100 | **0.774** ± 0.021 | 100 | 0.758 ± 0.019 | 100 |
| german credit | 0.876 ± 0.024 | 27 | 0.884* ± 0.029 | 22 | 0.866 ± 0.022 | 77 | 0.882 ± 0.010 | 100 | 0.852 ± 0.024 | 16 | 0.841 ± 0.019 | 12 | 0.882 ± 0.022 | 100 | 0.878 ± 0.027 | 100 | **0.887** ± 0.016 | 100 |
| heart | 0.874 ± 0.036 | 5 | 0.867 ± 0.046 | 12 | 0.863 ± 0.030 | 43 | **0.896*** ± 0.015 | 100 | 0.885 ± 0.022 | 12 | 0.859 ± 0.030 | 17 | 0.893 ± 0.022 | 100 | 0.893 ± 0.027 | 100 | **0.896** ± 0.019 | 100 |
| image | 0.953 ± 0.002 | 22 | 0.948 ± 0.015 | 23 | 0.952 ± 0.008 | 31 | 0.955* ± 0.004 | 100 | 0.933 ± 0.011 | 12 | 0.914 ± 0.020 | 33 | **0.957** ± 0.003 | 100 | 0.955 ± 0.007 | 100 | **0.957** ± 0.007 | 100 |
| ringnorm | 0.942* ± 0.005 | 93 | 0.919 ± 0.006 | 60 | 0.937 ± 0.005 | 79 | 0.939 ± 0.002 | 100 | 0.909 ± 0.008 | 24 | 0.886 ± 0.004 | 14 | 0.921 ± 0.006 | 100 | **0.948** ± 0.006 | 100 | 0.947 ± 0.002 | 100 |
| solar flare | **0.676*** ± 0.028 | 4 | 0.662 ± 0.051 | 16 | 0.641 ± 0.071 | 1 | 0.670* ± 0.063 | 100 | 0.628 ± 0.063 | 11 | 0.648 ± 0.101 | 11 | 0.655 ± 0.038 | 100 | 0.607 ± 0.099 | 100 | 0.593 ± 0.105 | 100 |
| splice | 0.975 ± 0.006 | 20 | 0.978 ± 0.005 | 48 | 0.979* ± 0.007 | 83 | 0.979* ± 0.006 | 100 | 0.976 ± 0.008 | 21 | 0.973 ± 0.008 | 22 | **0.984** ± 0.004 | 100 | 0.980 ± 0.005 | 100 | 0.982 ± 0.006 | 100 |
| thyroid | 0.944 ± 0.032 | 2 | **0.953*** ± 0.029 | 6 | 0.944 ± 0.019 | 1 | 0.944 ± 0.032 | 100 | 0.935 ± 0.031 | 4 | 0.940 ± 0.035 | 9 | 0.944 ± 0.032 | 100 | 0.944 ± 0.032 | 100 | 0.907 ± 0.025 | 100 |
| titanic | 0.806 ± 0.021 | 9 | 0.797 ± 0.017 | 25 | 0.794 ± 0.026 | 23 | 0.794 ± 0.031 | 100 | 0.802 ± 0.018 | 21 | 0.812* ± 0.030 | 15 | 0.803 ± 0.017 | 100 | **0.822** ± 0.036 | 100 | 0.821 ± 0.019 | 100 |
| twonorm | 0.962* ± 0.004 | 59 | 0.953 ± 0.003 | 45 | 0.960 ± 0.004 | 82 | 0.962* ± 0.002 | 100 | 0.951 ± 0.006 | 28 | 0.943 ± 0.005 | 37 | 0.965 ± 0.004 | 100 | **0.967** ± 0.002 | 100 | **0.967** ± 0.002 | 100 |
| waveform | 0.921* ± 0.006 | 75 | 0.908 ± 0.007 | 29 | 0.919 ± 0.014 | 96 | 0.917 ± 0.009 | 100 | 0.893 ± 0.009 | 21 | 0.886 ± 0.007 | 16 | 0.923 ± 0.009 | 100 | 0.927 ± 0.008 | 100 | **0.928** ± 0.010 | 100 |
| adult | 0.816 ± 0.002 | 72 | 0.817 ± 0.002 | 100 | 0.817 ± 0.001 | 90 | 0.817 ± 0.001 | 96 | 0.818 ± 0.002 | 100 | 0.819* ± 0.001 | 49 | 0.819 ± 0.001 | 100 | **0.820** ± 0.001 | 100 | **0.820** ± 0.002 | 100 |
| compas | 0.667 ± 0.012 | 17 | 0.669 ± 0.017 | 78 | 0.669 ± 0.017 | 44 | 0.669 ± 0.014 | 100 | **0.670*** ± 0.017 | 3 | 0.668 ± 0.016 | 40 | **0.670** ± 0.012 | 100 | 0.669 ± 0.014 | 100 | 0.669 ± 0.015 | 100 |
| employment CA2018 | 0.746 ± 0.002 | 69 | 0.751 ± 0.003 | 100 | 0.748 ± 0.001 | 75 | 0.748 ± 0.003 | 75 | 0.752* ± 0.001 | 82 | 0.742 ± 0.003 | 18 | 0.748 ± 0.001 | 100 | **0.755** ± 0.001 | 100 | 0.754 ± 0.002 | 100 |
| employment TX2018 | 0.759 ± 0.001 | 85 | 0.762* ± 0.002 | 99 | 0.759 ± 0.003 | 90 | 0.760 ± 0.001 | 91 | 0.761 ± 0.003 | 64 | 0.747 ± 0.005 | 17 | 0.758 ± 0.003 | 100 | 0.764 ± 0.003 | 100 | **0.765** ± 0.004 | 100 |
| public coverage CA2018 | 0.702 ± 0.011 | 30 | 0.707 ± 0.004 | 100 | 0.713 ± 0.003 | 100 | 0.713 ± 0.002 | 100 | 0.715* ± 0.003 | 100 | 0.707 ± 0.003 | 20 | 0.716 ± 0.002 | 100 | **0.718** ± 0.003 | 100 | 0.716 ± 0.002 | 100 |
| public coverage TX2018 | 0.851 ± 0.001 | 22 | 0.853* ± 0.002 | 91 | 0.852 ± 0.003 | 99 | 0.849 ± 0.001 | 100 | 0.853* ± 0.003 | 44 | 0.850 ± 0.001 | 13 | 0.851 ± 0.003 | 100 | **0.856** ± 0.003 | 100 | 0.855 ± 0.003 | 100 |
| mushroom secondary | **0.999*** ± 0.000 | 51 | 0.987 ± 0.002 | 31 | **0.999*** ± 0.001 | 81 | **0.999*** ± 0.000 | 86 | 0.936 ± 0.008 | 19 | 0.940 ± 0.035 | 43 | 0.998 ± 0.000 | 100 | **0.999** ± 0.000 | 100 | **0.999** ± 0.000 | 100 |
| Mean/Median | **0.838***/**0.863*** | 35.6/24.5 | 0.834/0.845 | 46.6/31.5 | 0.831/0.841 | 59.1/76 | 0.835/0.834 | 97.4/100 | 0.825/0.835 | 30.8/20.5 | 0.820/0.833 | 22.8/17 | 0.836/0.835 | 100/100 | 0.838/0.839 | 100/100 | 0.836/0.843 | 100/100 |

As the depth of the decision trees increases, we observe a shift in the relative performance of the boosting methods. Table 2, which reports results for depth-5 trees, illustrates this trend. These findings suggest that deeper trees tend to favor traditional heuristic methods such as XGBoost and LightGBM. Nevertheless, totally corrective methods —NM-Boost in particular— remain competitive, especially in terms of the trade-off between testing accuracy and ensemble sparsity. This pattern becomes even more apparent for depth-10 trees (Appendix D.2): as tree depth increases further, the trade-offs between ensemble size and accuracy grow more pronounced. We observe a gradual increase in the testing accuracy of the benchmarked heuristic boosting methods as tree depth increases. Nonetheless, for most datasets, the difference relative to the totally corrective boosting methods remains statistically insignificant, while the latter consistently yield much sparser ensembles. Specifically, for depth-5 trees, only 3 out of 20 datasets showed a statistically significant improvement in testing accuracy of heuristic methods over totally corrective ones. For depth-10 trees, this number increased slightly to 5 out of 20. In contrast, the totally corrective methods significantly outperformed the heuristic ones on 1 out of 20 datasets for both depths.

**Observation 6** *As the depth of decision trees increases, the relative accuracy of totally corrective boosting methods declines compared to heuristic ones. However, on most datasets, differences in testing accuracy are not statistically significant, while totally corrective methods consistently produce much sparser ensembles.*

**Observation 7** *NM-Boost remains the most accurate totally corrective boosting method with depth-3 and depth-5 trees. With depth-10 trees, CG-Boost and ERLP-Boost match its accuracy but produce significantly larger ensembles.*

## 6.2 Anytime Performance

We now evaluate the anytime performance of the different boosting methods, i.e., how their testing accuracy evolves over the iterations. To maintain readability and conserve space, we present results for two representative datasets and a focused subset of methods. The remaining results follow the same trends (in particular, supporting all our drawn observations) and will be released alongside our source code.

Figure 2 shows the testing accuracy of the considered boosting approaches at each iteration —i.e., after the addition of each new base learner— on the *image* dataset, across different decision tree depths. We observe that performance differences are more pronounced for shallower trees (e.g., depth 1) and tend to diminish as tree depth increases (e.g., depth 10). This suggests that the strategy used to generate and aggregate base learners has a greater impact when the individual learners are weak.

**Observation 8** *Differences in anytime performance between boosting approaches are most pronounced when using shallow trees. With decision stumps, totally corrective methods significantly outperform heuristic methods —especially Adaboost— during the early iterations.*

Furthermore, in the early iterations, totally corrective methods outperform the heuristic benchmarks. This trend is reversed on the *ringnorm* dataset (Figure 3), where XGBoost and LightGBM consistently outperform the totally corrective methods across all tree depths except depth 10. These results suggest that the ability of each boosting approach to accurately fit the data in early iterations is dataset-dependent. Interestingly, even when using deeper trees, totally corrective boosting methods often yield better performance in early iterations, although heuristic methods tend to catch up quickly as more learners are added.

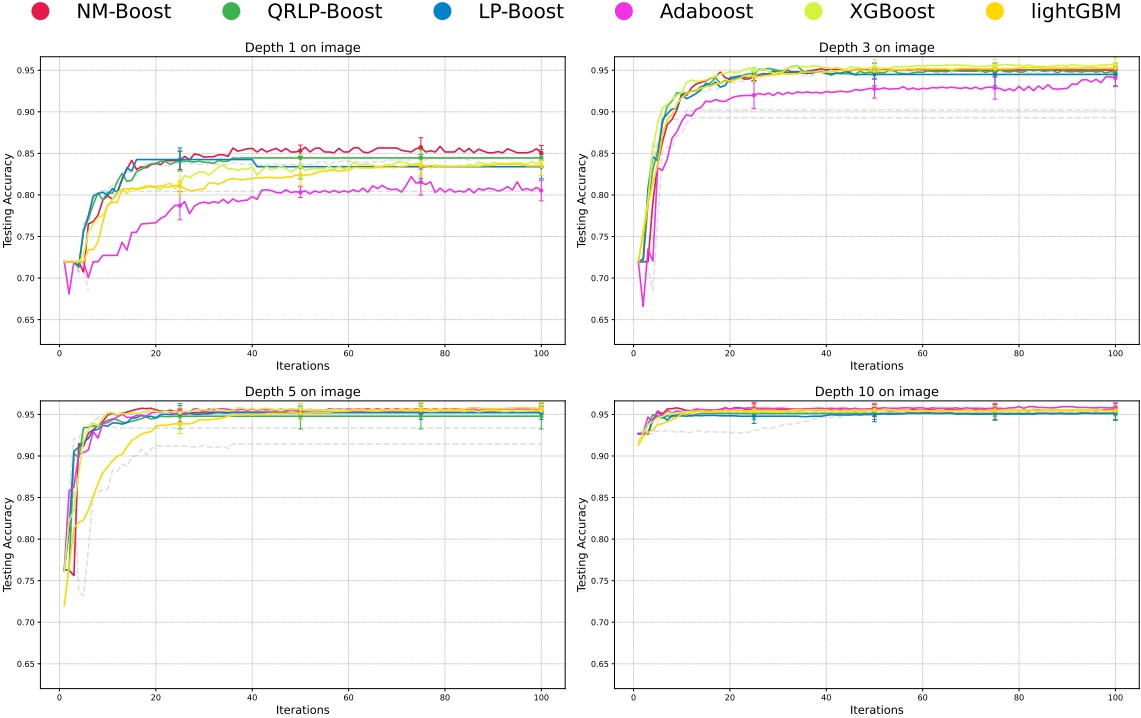

Figure 2: Anytime behavior on the **image** dataset for selected methods (all other methods in gray), for CART decision trees of depth 1, 3, 5, and 10. Error bars indicate the standard deviation over 5 seeds.

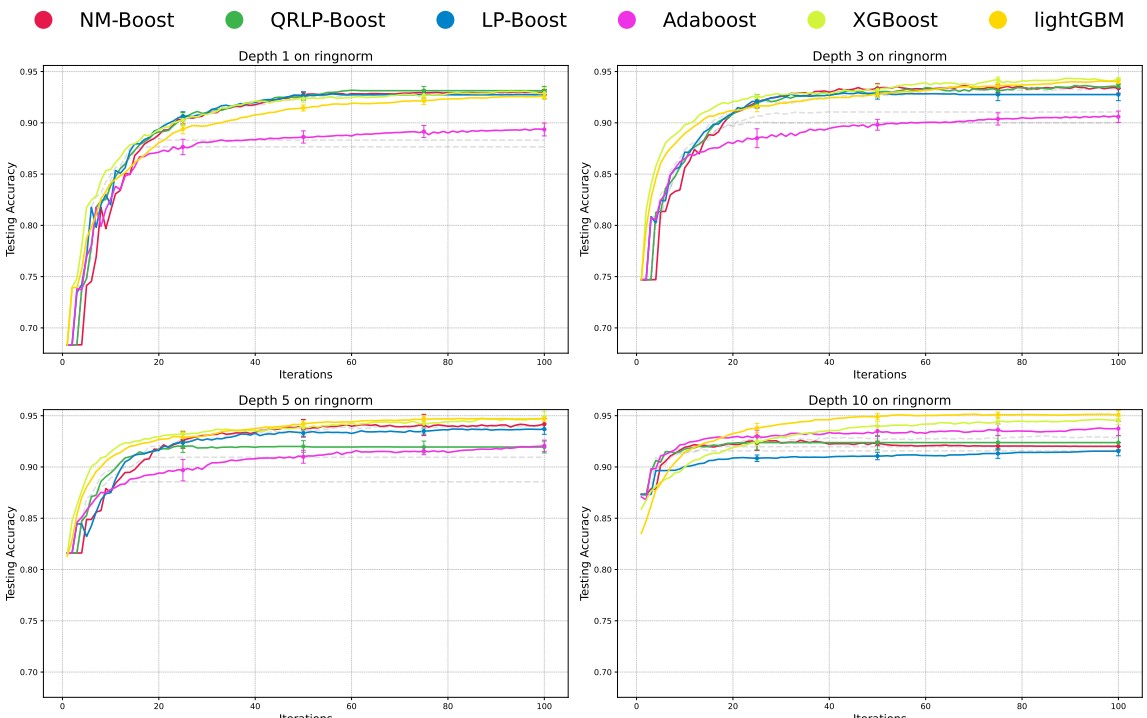

Figure 3: Anytime behavior on the **ringnorm** dataset for selected methods (all other methods in gray), for CART decision trees of depth 1, 3, 5, and 10. Error bars indicate the standard deviation over 5 seeds.

**Observation 9** *When using deeper CART trees as base learners, totally corrective methods often achieve better early-stage performance than heuristic approaches —though the latter typically converge to slightly higher final accuracy.*

The anytime performance of the boosting approaches may also be influenced by how the individual trees are constructed in the underlying algorithms. Indeed, XGBoost and LightGBM both use a custom leaf-wise growth strategy to build their base learners, while all other methods use a depth-wise approach (CART) to construct each decision tree. We observe that XGBoost and LightGBM perform significantly worse than the other methods during the initial iterations when using decision trees deeper than decision stumps. Their performance typically catches up to that of the other methods after around 10 iterations. This behavior is illustrated, for example, on the *adult* dataset, with detailed results provided in Appendix D.3.

### 6.3 Ensemble Sparsity

We now analyze ensemble sparsity, defined as the number of base learners with non-zero weights in the final ensemble. We exclude XGBoost and LightGBM from this analysis, as they use stage-wise additive boosting with implicit weights. In these methods, each tree contributes to the final prediction through its leaf values, scaled by the learning rate, and all trees remain active in the ensemble. Figure 4 presents barcharts showing the assigned weight of each tree across different methods for four representative datasets. The results reveal substantial variation in sparsity across methods. As expected, Adaboost always uses all trees by design, since it is not totally corrective and assigns weights sequentially. Surprisingly, CG-Boost —despite being a totally corrective method and an extension of LP-Boost with a regularization term on the ensemble weights— often uses all trees as well. This observation is consistent with prior findings in Roy et al. (2016) and suggests that CG-Boost's regularization may not sufficiently promote sparsity in practice. Other totally corrective methods are able to obtain the same or better performance with significantly fewer trees. Notably, NM-Boost, LP-Boost, and ERLP-Boost frequently yield sparse ensembles without sacrificing predictive accuracy. QRLP-Boost occasionally produces sparse ensembles but is less consistent —for example,

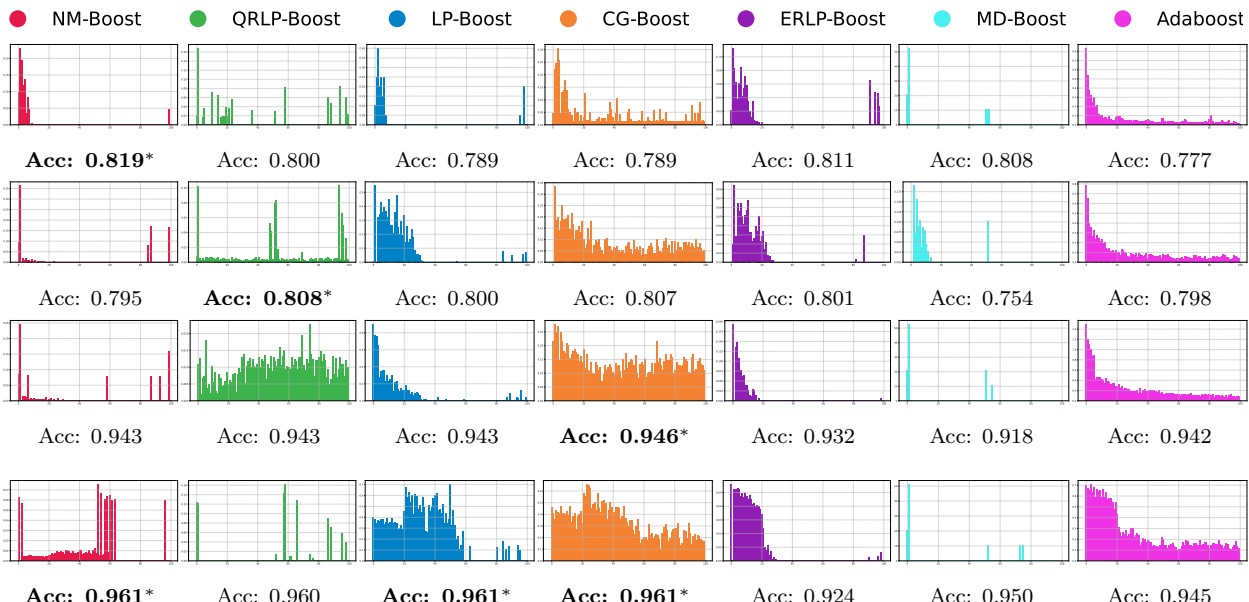

Figure 4: Ensemble weights for four datasets (top to bottom: **breast cancer**, **german credit**, **splice**, and **twonorm**) and depth 1 CART trees for totally corrective methods and Adaboost, over 5 seeds. **Bold** highlights the best *overall* accuracy, while a star* marks the best among *totally corrective* methods.

on the *splice* dataset, it assigns non-zero weights to all 100 trees. MD-Boost generates the sparsest ensembles overall, but this comes at the cost of noticeably reduced accuracy. NM-Boost consistently produces sparser ensembles than QRLP-Boost, as its objective implicitly promotes selecting only trees that perform well.

**Observation 10** *The methods that structurally return the sparsest ensembles while maintaining competitive accuracy are NM-Boost, LP-Boost, and ERLP-Boost.*

### 6.4 Margin Analysis

As mentioned in Section 3, most totally corrective methods, including those proposed in this work, focus on optimizing the margins. In this section, we analyze the cumulative margin distribution of the methods on the test data. This allows us to compare not only the proportion of misclassified instances (i.e., those with negative margins) but also the structure of the decision boundaries induced by different methods.

First, we consider the margins of the boosting methods with tree depths of 1, 3, and 5, for various datasets. Figure 5 shows the cumulative margin distributions on the *german credit* and *ringnorm* datasets. We observe that the minimum margin of the methods across the depths is not directly linked to test accuracy, echoing the insights of Shen & Li (2010). Similarly, the variance of the margin distribution does not appear to be the primary driver of performance: for example, QRLP-Boost often yields lower margin variance across depths, yet it is not always the top-performing method. This supports the doubts raised by Breiman (1999) and further discussed in Gao & Zhou (2013). QRLP-Boost and NM-Boost generally achieve the lowest margin variance, although this difference vanishes with deep decision trees, aligning with the absence of significant performance differences in that regime.

**Observation 11** *Differences in performance across boosting methods cannot be fully explained by their margin distribution. Methods with a heavy right-tail of the distribution (high confidence on correctly classified data points) do not necessarily have the highest accuracy and lowest minimum margin.*

Next, we take a closer look at NM-Boost and QRLP-Boost, and analyze how their margin distributions for the test data change for different values of their tradeoff hyperparameter ($C$). As a comparison, we also plot the margin distribution of Adaboost for different values of its hyperparameter (learning rate). We

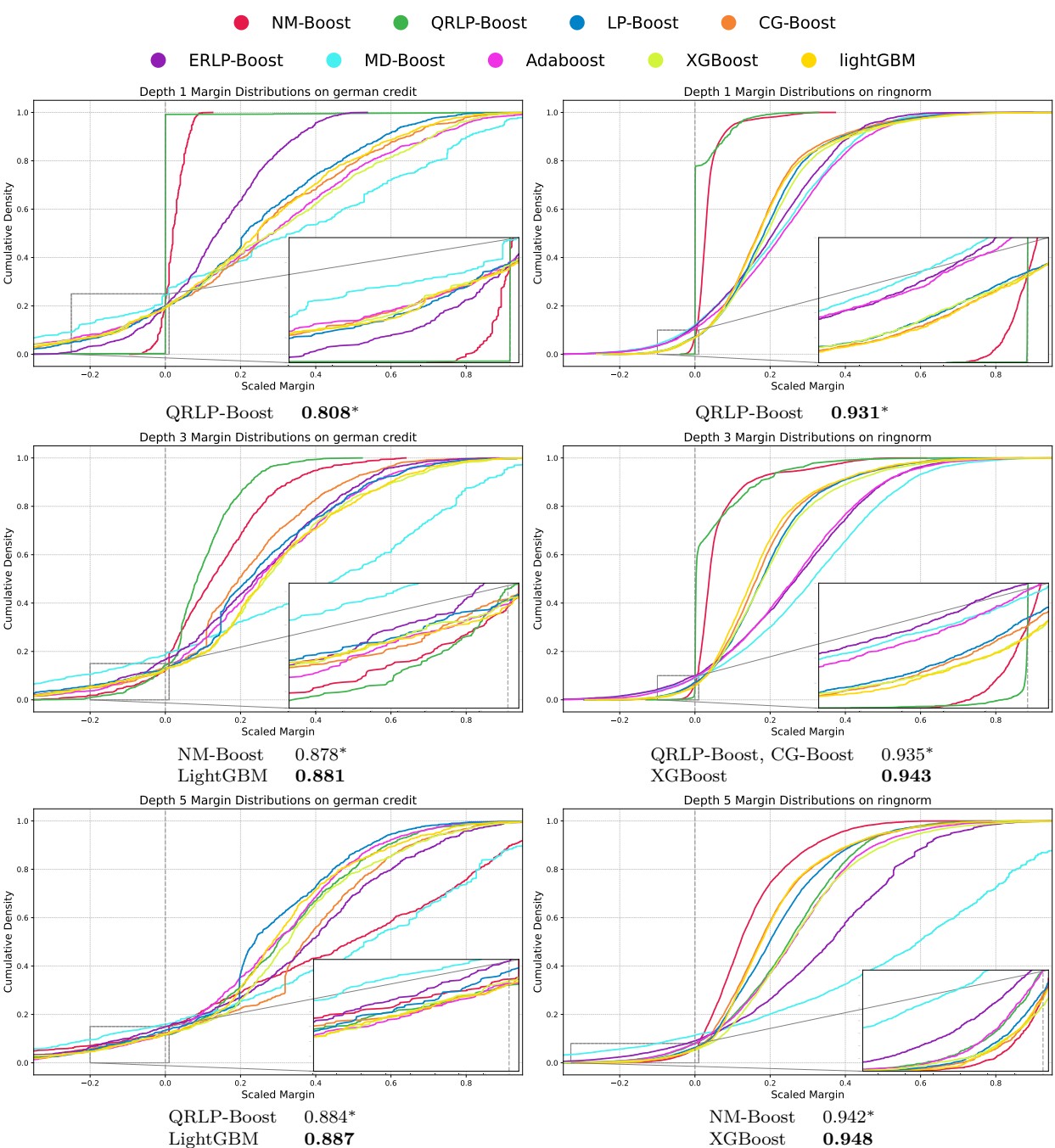

Figure 5: Test data margin distribution for **german credit** (left column plots) and **ringnorm** (right column plots) datasets using CART trees of depths 1, 3, and 5, over 5 seeds. **Bold** highlights the best *overall* accuracy, while a star* marks the best among *totally corrective* methods.

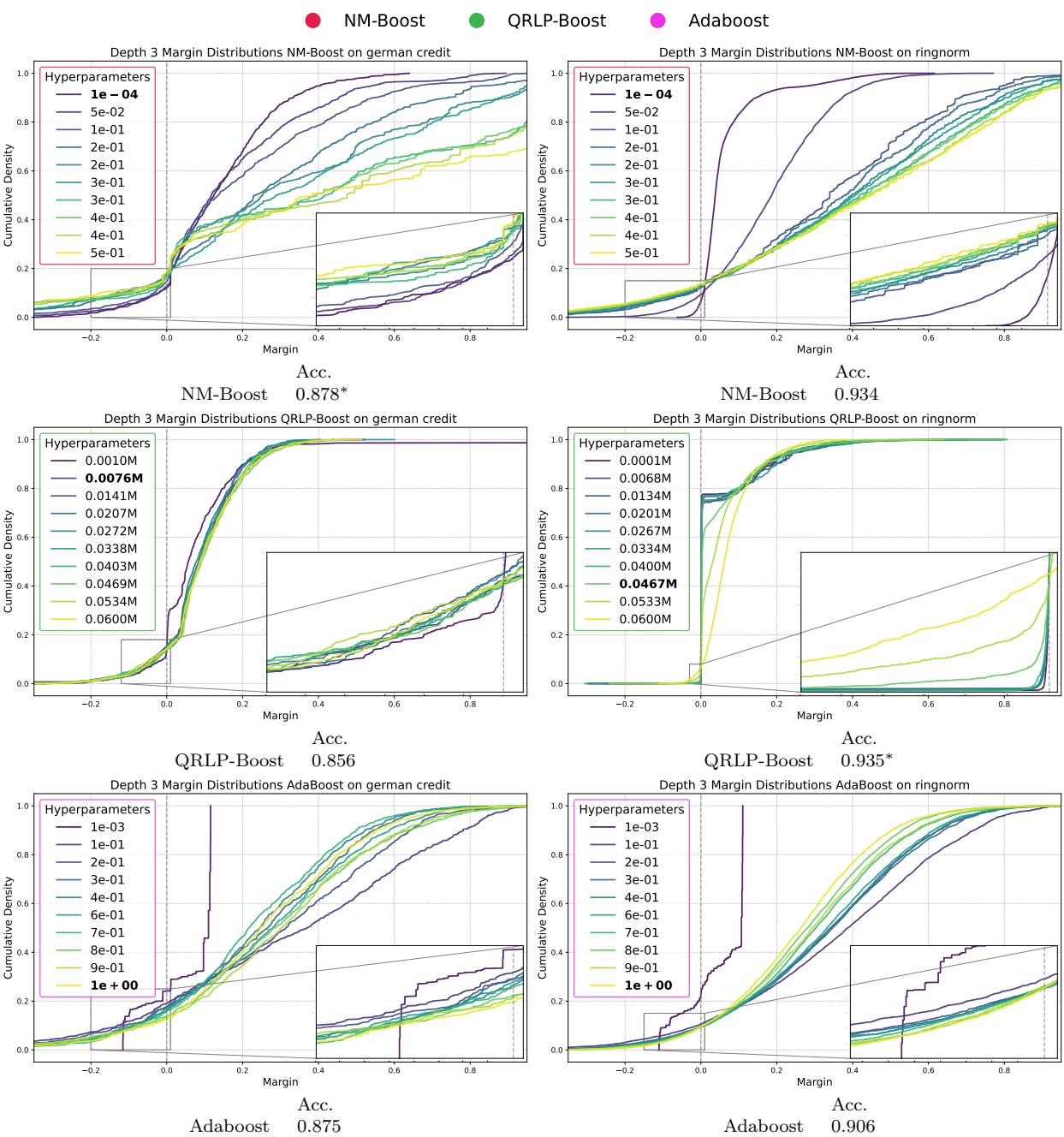

Figure 6: Test margin distribution over the hyperparameter for **german credit** (left column plots) and **ringnorm** (right column plots) datasets using depth 3 CART trees, over 5 seeds. The best hyperparameter value is marked **bold** in the legend. **Bold** accuracy highlights the best *overall*, while a star* marks the best among *totally corrective* methods. The legend's outline color indicates the used formulation.

display in Figure 6 the margin distributions for depth-3 trees, again for the *german credit* and *ringnorm* datasets. Remember that for NM-Boost, the hyperparameter $C$ controls the tradeoff between reducing negative margins and increasing the sum of margins. Lower values of $C$ focus on reducing the negative portion of the margin distribution, thereby minimizing misclassifications, whereas higher values enhance generalization by increasing the positive margins. For QRLP-Boost, $C$ controls the balance between the quadratic regularization of sample weights and the original LP-Boost linear worst-case margin.

For NM-Boost, the hyperparameter has a direct and large effect on the variance of margin distributions: larger values of $C$ place more emphasis on maximizing the sum of margins (i.e., assigning more weight to confidently classified examples), which leads to a wider spread in margin values and thus higher variance.

For QRLP-Boost, we observe a smaller impact on variance but a more noticeable effect on the *smoothness* of the margin distribution: increasing $C$ tightens the upper bound on individual sample weights, effectively increasing the regularization on the distribution. This promotes smoother distributions and moves the objective further from that of standard LP-Boost.

Interestingly, for NM-Boost, lower variance in the margin distribution tends to correlate with better generalization on the test set, which is analogous to insights in Shen & Li (2010). However, this result is not necessarily confirmed by the earlier results in Figure 5. The same goes for QRLP-Boost: a smoother curve does not necessarily yield better generalization. If we look at the Adaboost margins, this is confirmed, as the best performance is not at the lowest variance margin distribution.

**Observation 12** *Margin distribution variance is not necessarily correlated with testing accuracy.*

### 6.5 Hyperparameter Sensitivity

We further evaluate the sensitivity of methods to their hyperparameter. To do so, we plot the testing accuracy and number of used columns (as before) but now for *all hyperparameter values*, instead of only the best found on the validation data. We hereafter focus on two representative datasets, and plot results for all remaining ones in Appendix D.4. Figure 7 shows the performance of totally corrective boosting approaches on the *ringnorm* and *mushroom secondary* datasets using depth-1 and depth-5 trees. Across all 20 datasets, we consistently observe a trade-off between sparsity and testing accuracy. The interesting finding is that

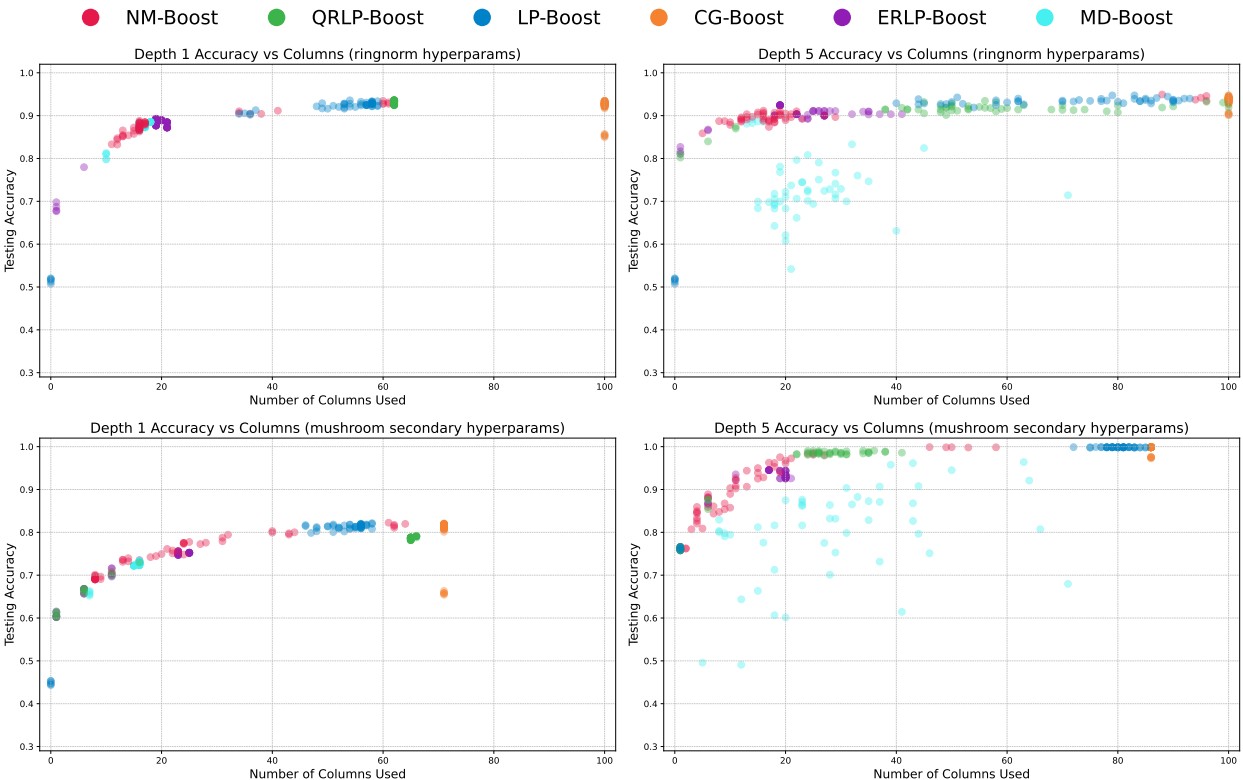

Figure 7: Average testing accuracy compared to average ensemble sparsity over two datasets (top: **ringnorm**, bottom: **mushroom secondary**) and **all hyperparameter values** for **depth 1** (left) and **depth 5** (right) CART tree base learners, over 5 seeds.

each method occupies a different point along this Pareto front[3], highlighting distinct performance–sparsity trade-offs. CG-Boost often lies on the rightmost part of the front, yielding ensembles that perform well but use most of the generated base learners. LP-Boost, QRLP-Boost, ERLP-Boost, and NM-Boost offer an interesting range of trade-offs, as they are often able to improve sparsity while preserving predictive performance. Overall, we observe that the different totally corrective approaches are complementary, i.e., they are placed in different parts of the Pareto front, and the selection of the method may be determined based on the specific accuracy-sparsity tradeoff requirement per dataset.

**Observation 13** *We observe a clear trade-off between accuracy and sparsity, and most methods are placed on the Pareto front of both metrics, i.e., they build non-dominated ensembles.*

For depth-5 trees, Figure 7 shows that MD-Boost is often dominated, i.e., there is always a method that finds better accuracy and sparsity. However, its best-performing hyperparameter typically lies on the frontier, underscoring the importance of careful tuning. In MD-Boost, the hyperparameter $C$ governs the trade-off between maximizing the first moment and minimizing the second moment of the margin distribution. As also highlighted in Demiriz et al. (2002), LP-Boost is prone to degenerate solutions when the set of weak learners is small. Moreover, it is highly sensitive to its hyperparameter: for a high value of $C$, LP-Boost reverts to hard-margin behavior, reintroducing the issues identified in Grove & Schuurmans (1998). These effects are clearly visible in Figure 7 (leftmost points, with trivial accuracy) and are consistent across the other datasets.

**Observation 14** *Hyperparameter tuning is crucial for some totally corrective boosting approaches — particularly MD-Boost and LP-Boost— as unsuitable values of their trade-off parameter can lead to poorly performing ensembles.*

### 6.6 Reweighting Behavior

We now consider a post-processing experiment where an ensemble of 100 trees is first generated using Adaboost. Each totally corrective method is then applied in a single shot to reweight this fixed ensemble, i.e., to optimize the base learner weights without generating new ones. Our goal is to assess whether totally corrective reweighting can yield sparser or more accurate ensembles compared to the original heuristic solution. Related work has shown promise in post-processing Adaboost ensembles, see Emine et al. (2025). As previously, this experiment is conducted for tree depths of 1, 3, 5, and 10. Figure 8 shows the average performance across all datasets, while Appendix D.5 details all results per dataset.

First, we observe that totally corrective methods can slightly improve the predictive performance of the original Adaboost ensembles, in terms of testing accuracy (particularly when using decision stumps) and in terms of accuracy-sparsity tradeoff. For deeper trees (depth 10), reweighting generally does not allow significant performance improvements. Overall, totally corrective methods appear as a viable strategy to thin existing ensembles while keeping up performance, especially for shallow tree ensembles.

**Observation 15** *Totally corrective methods can be an effective post-processing strategy to sparsify existing ensembles while preserving —or modestly improving— their predictive performance.*

However, the reweighted ensembles consistently underperform compared to those natively generated by the totally corrective methods (Figure 1). This indicates that the strength of totally corrective boosting lies not only in its base learner weight optimization, but also in its ability to iteratively generate informative base learners. The performance gains observed with totally corrective methods result from this joint optimization over both base learner selection and weight assignment.

### 6.7 Effect of Confidence-Rated Voting

Following the proposal by Demiriz et al. (2002), we investigate the impact of confidence-rated voting in boosting ensembles. In that setting, each base learner $h_j$ no longer outputs a binary label ($h_j : \mathcal{X} \mapsto$

---

[3]In multi-objective optimization, the Pareto front contains *non-dominated* solutions — none improve one objective without worsening another. Here, an ensemble is non-dominated if accuracy or sparsity cannot improve without degrading the other.

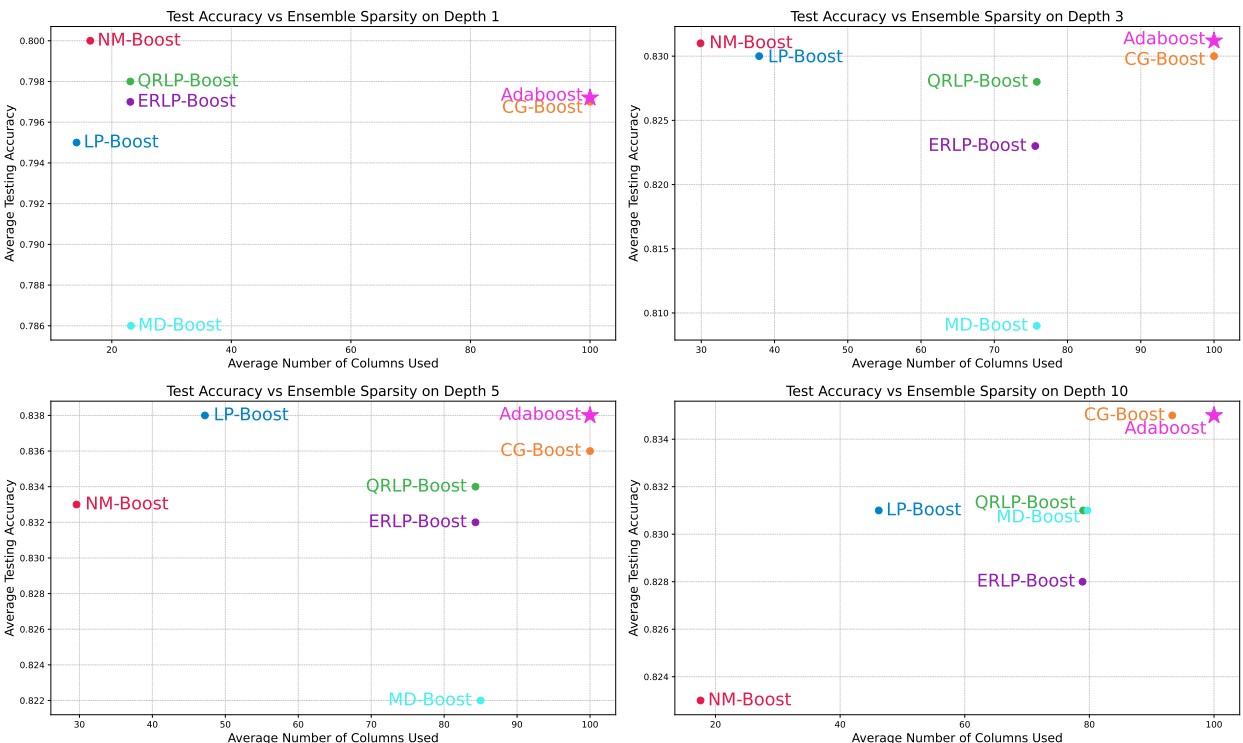

Figure 8: Reweighting of an existing Adaboost ensemble of 100 trees: average testing accuracy compared to average ensemble sparsity over **all datasets** for decision trees of depth 1, 3, 5, and 10.

$\{-1, +1\}$), but rather a real-valued score ($h_j : \mathcal{X} \mapsto [-1, 1]$) representing its confidence in the prediction. These scores are typically derived from the class proportions in the training examples reaching each leaf during tree construction. Confidence-rated outputs allow for finer-grained voting, where each tree contributes proportionally to (an estimate of) its certainty.

Figure 9 provides a summary of the results using boxplots across datasets. Additional analyses —including anytime performance, as well as sparsity–accuracy trade-offs on individual datasets— are presented in Appendix D.6. In line with the observations of Demiriz et al. (2002), we find no significant improvement in final test performance when using confidence-rated trees as opposed to standard CART trees. However, we do observe that the relative final performance of the methods sometimes slightly changes. For instance, MD-Boost often benefits from confidence-rated outputs, while QRLP-Boost and ERLP-Boost tend to degrade in performance when using confidence-rated decision stumps. This likely stems from the limited capacity of shallow trees to produce meaningful confidence estimates. For deeper trees, these effects diminish. For the remaining methods, performance differences between binary and confidence-rated voting are negligible. Appendix D.6 further shows that confidence-rated outputs do not consistently improve anytime performance. That said, for moderately deep trees (depths 3 and 5), we occasionally observe slightly improved anytime behavior of confidence-rated trees in the early iterations, especially for NM-Boost, QRLP-Boost, LP-Boost and ERLP-Boost.

**Observation 16** *We do not observe a significant performance difference when using confidence-rated trees versus normal CART trees.*

## 7 Experiments with Optimal Trees as Base Learners

Thus far, we have relied on the popular CART heuristic algorithm to generate decision tree base learners. In this section, we evaluate the impact of replacing CART with *optimal decision trees* (ODTs) instead,

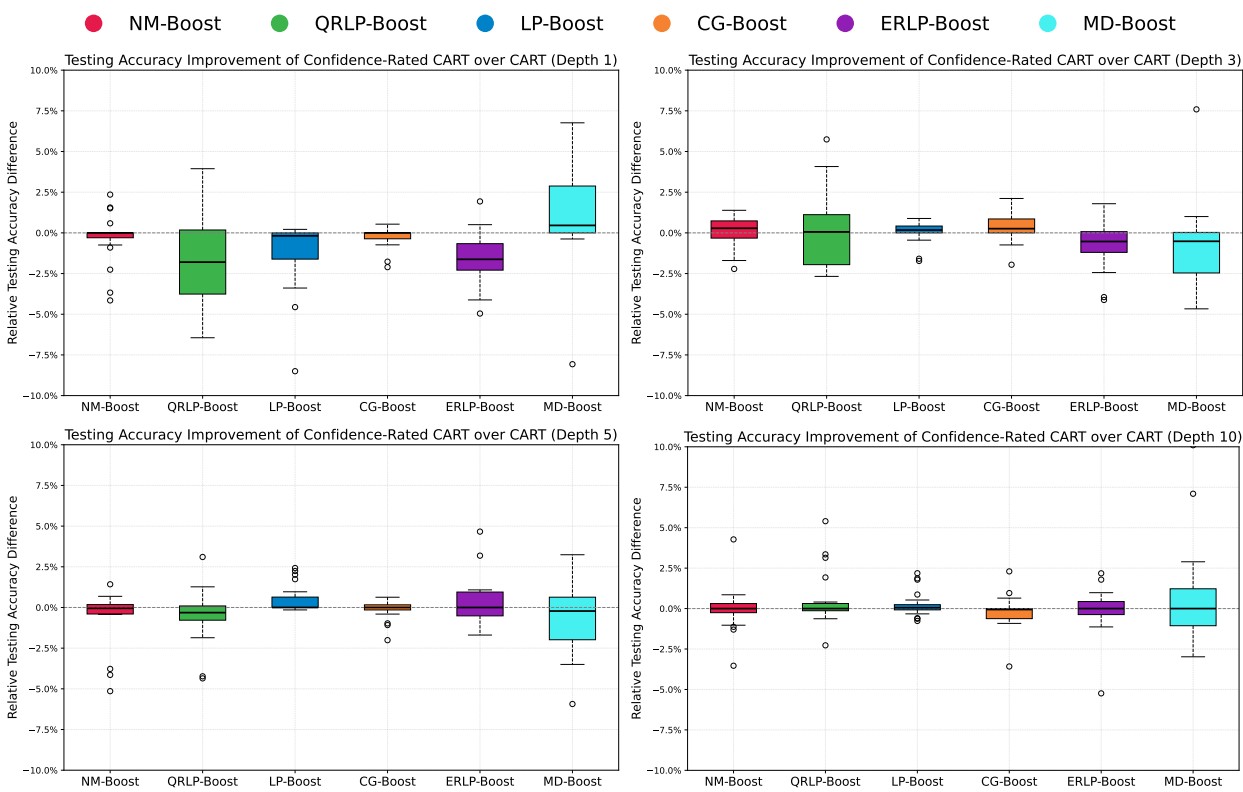

Figure 9: Boxplot showing relative improvement of using confidence-rated CART trees over regular CART trees in the trained ensembles over **all datasets**, for decision trees of depth 1, 3, 5, and 10, over 5 seeds. For visualization purpose, a few extreme outliers were removed.

leveraging recent advances in ODTs learning algorithms. Indeed, in the last years, the scalability of these algorithms has improved, making them a suitable alternative to heuristics like CART (van der Linden et al., 2025). Examples of these scalable algorithms are `LDS-DL8.5` (Kiossou et al., 2022), `MurTree` (Demirović et al., 2022), and `Blossom` (Demirović et al., 2023). For a broader empirical analysis of ODTs, we refer to van der Linden et al. (2025). We decide to conduct experiments with `Blossom`, due to its memory efficiency at deeper depths and its native support for sample weighting. `Blossom` maximizes the training accuracy given the maximum tree depth.

To our knowledge, the only prior study evaluating ensembles with optimal trees is Aglin et al. (2021), who investigate LP-Boost and MD-Boost on a limited number of small datasets (up to 958 data points) with trees of depth at most 3. They show that on some datasets, optimal trees can improve performance, but they might also decrease performance on others. They conclude that the objective of the totally corrective method has a large effect on the performance of optimal decision tree ensembles. Our findings support this conclusion across a broader variety of totally corrective methods and ODT depths.

Figure 10 shows the anytime behavior of NM-Boost, QRLP-Boost, and LP-Boost on the *ringnorm* dataset. In general, we find that the final test accuracy when using optimal decision trees is often similar to CART trees.

**Observation 17** *We do not observe any significant or consistent difference in anytime behavior between ODT and CART ensembles.*

Figure 11 illustrates the improvement in predictive performance (aggregated over all datasets) achieved when using ODTs instead of CART trees in the trained ensembles. The results confirm that the performance gain of ODTs is dataset dependent. These findings refine and extend the observations from Aglin et al.

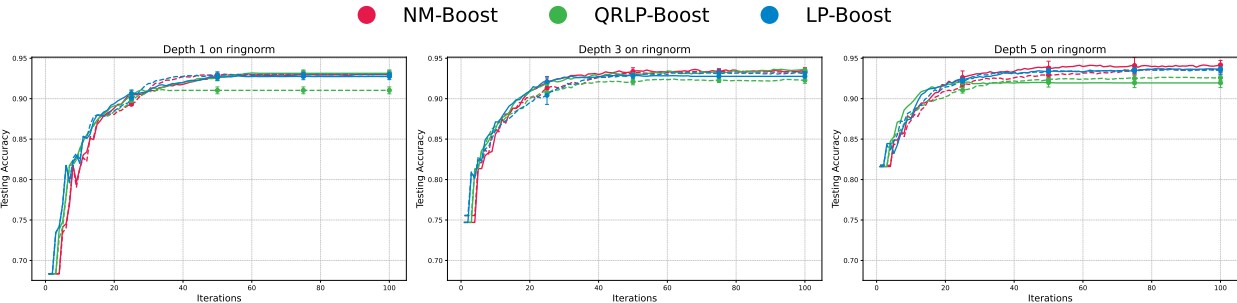

Figure 10: Anytime behavior of several totally corrective boosting methods on the **ringnorm** dataset, using base learner decision trees of depth 1, 3, or 5 trained either with CART (full line) or with an optimal algorithm (dashed line), the error bars indicate standard deviation over 5 seeds.

(2021): performance improvements can be observed on specific datasets, but the average effect of using ODTs yields similar results to CART. Detailed per-dataset results that support these conclusions are provided in Appendix D.7, together with sparsity-accuracy trade-off summaries, which show that ODT ensembles offer no consistent sparsity advantage nor disadvantage compared to CART-based ensembles.

**Observation 18** *Across all tree depths and nearly all datasets, the final performance of totally corrective methods is similar when using optimal decision trees compared to heuristic CART trees.*

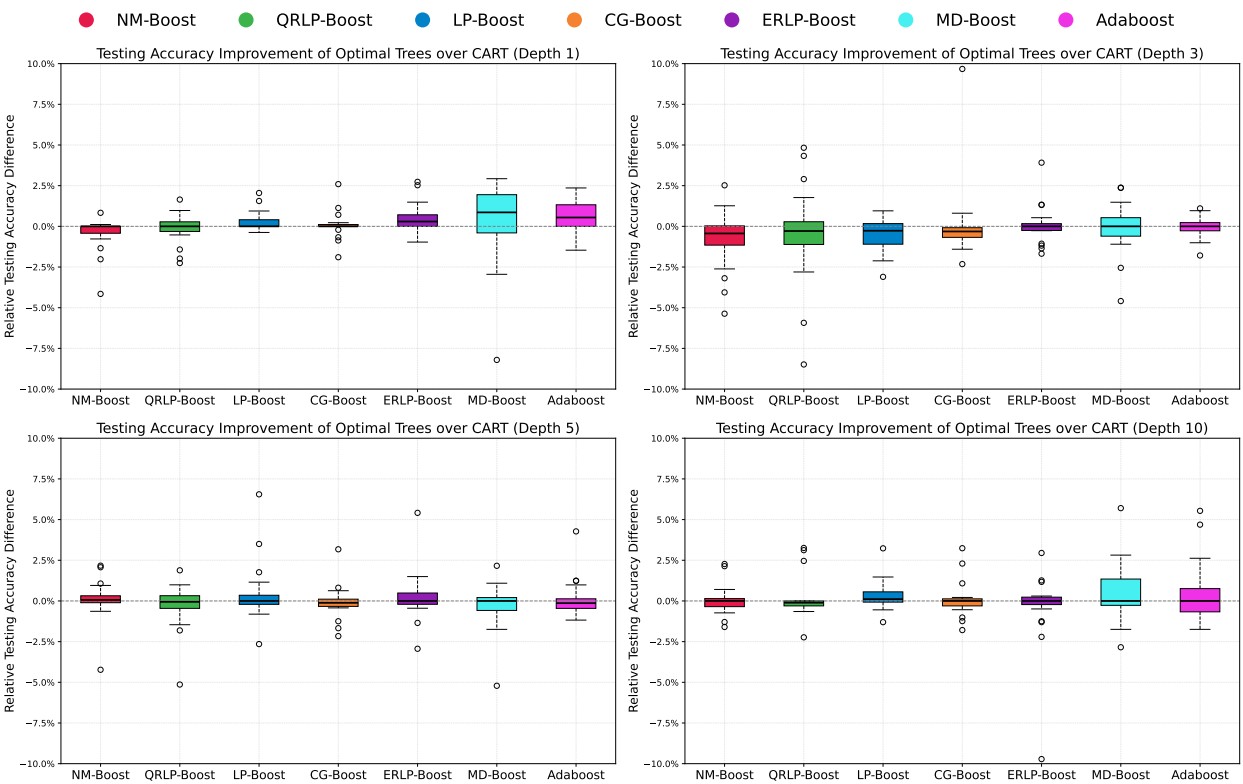

Figure 11: Relative improvement of the testing accuracy of boosting approaches when using optimal decision trees vs. CART trees over **all datasets**, for depth 1, 3, 5, and 10 tree ensembles, over 5 seeds.

We conjecture that optimal decision trees are likely stronger individual predictors with lower bias, but their construction may lead to less diversity across iterations. This reduced diversity may ultimately hinder the ensemble's overall predictive performance, yielding limited improvement potential over CART.

## 8    Conclusions

In this paper, we conducted an extensive empirical study of totally corrective boosting methods based on column generation. We compared four existing methods, two novel formulations, and three state-of-the-art heuristic baselines across 20 datasets, varying tree depth and base learners. Our evaluation focused on two key dimensions: testing accuracy and sparsity of the ensemble. We found that totally corrective methods can outperform or match heuristic benchmarks, especially when using shallow trees. With decision stumps —the common default in boosting— the proposed NM-Boost and QRLP-Boost achieved the best performance among totally corrective methods, with NM-Boost achieving state-of-the-art accuracy using significantly fewer trees.

As tree depth increases, heuristic methods such as XGBoost and LightGBM tend to gain an advantage in predictive performance. Nevertheless, totally corrective methods remain competitive and often offer a favorable accuracy-sparsity trade-off. We observed that the best-performing methods are typically those that balance ensemble size with careful margin-based optimization, especially NM-Boost, LP-Boost, and QRLP-Boost. These methods often lie on the Pareto frontier between accuracy and sparsity, achieving high accuracy with compact ensembles.

Our margin analysis revealed that neither minimum margin nor margin variance alone is sufficient to explain generalization behavior. QRLP-Boost often yields smooth margin distributions, while NM-Boost promotes focus on misclassified datapoints. Yet, both strategies can generalize well, depending on the dataset. We also studied hyperparameter sensitivity and found that while most totally corrective methods are stable under tuning, their performance can be strongly influenced by it. In particular, MD-Boost and LP-Boost can suffer when their hyperparameters are not well aligned with the dataset.

We further evaluated the reweighting ability of totally corrective methods in a post-processing scenario, where they reassign weights to pre-trained Adaboost ensembles. While this procedure can improve sparsity and sometimes accuracy, it does not match the performance of fully-trained totally corrective ensembles. This underscores that the strength of totally corrective approaches lies not just in weight optimization, but also in the dynamic generation of new base learners during training. When replacing CART trees with optimal decision trees, we found that performance may increase depending on the dataset, but often is similar to CART.

Our study has a few limitations. All methods were trained using the same base learners (CART or Blossom) to ensure fairness, though this may favor or disadvantage certain approaches. Despite extensive hyperparameter tuning, some methods may benefit from even finer calibration. Finally, our conclusions are based on binary classification problems and may not fully transfer to regression, multi-class settings, or real-world applications with different structures.

Future work could explore more flexible formulations that adapt ensemble size to the strength of individual learners, or investigate how different types of base learners (e.g., mixed CART and optimal trees or CART trees of different depths) can be used together. Studying totally corrective methods in regression and multi-class tasks, stopping criteria, and applying these techniques in practical domains are also promising directions. Our reweighting experiments also motivate further research into voting schemes, ensemble thinning, and hybrid strategies combining heuristic and totally corrective learning to achieve both sparsity and performance.

### Broader Impact Statement

Some of the datasets used in our experiments (in particular, *adult* and *compas propublica*) contain sensitive attributes and are known to reflect societal biases. We stress that our use of these datasets is for methodological benchmarking, with no intention for the trained models to be directly deployed. We caution against

an uncritical application of our work and emphasize that any real-world use in sensitive domains should be accompanied by thorough fairness assessments and, where necessary, appropriate mitigation strategies. Finally, we argue that sparse ensembles —such as those obtained through linear programming boosting approaches— offer practical advantages for the detection and auditing of potential biases.

### Acknowledgments

This work used the Dutch national e-infrastructure with the support of the SURF Cooperative using grant no. EINF-13069. It was also enabled by support provided by Calcul Québec and the Digital Research Alliance of Canada, the SCALE AI Chair in Data-Driven Supply Chains, and the *Fonds de recherche du Québec – Nature et technologies (FRQNT)* through a Team Research Project *(327090)*. We thank Gurobi Optimization for extending our license to support large-scale experiments.

### Post-Publication Changes

After publication, we were notified of an implementation error in the `Blossom` library, used for obtaining optimal decision trees (ODTs). Therefore, we ran all experiments with ODTs again and updated the conclusions of Section 7.

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

# A    Datasets

In this section, we provide more details about the datasets used. We study 13 datasets commonly used in related works (Bi et al., 2004; Rätsch et al., 2007; Warmuth et al., 2008; Shen & Li, 2010; Mitsuboshi et al., 2022) which were proposed as binary classification benchmark by Rätsch et al. (2001) and originally sourced from UCI (Kelly et al., 2025), DELVE (Rasmussen et al., 1996) and STAT-LOG (Feng et al., 1993). We retrieved some datasets from the KEEL repository (Alcalá-Fdez et al., 2009; 2011). Some of the challenges were transformed from multi-class to binary classification by Rätsch et al. (2001). Given that many of these datasets are synthetic, we decided to supplement them with seven well-known and publicly available real-world datasets. Table 3 summarizes the key statistics of the benchmark datasets and the supplementary datasets, after preprocessing.

All features are binarized as this streamlines presentation and comparison of models, and it also allows us to conduct experiments with optimal decision trees, whose state-of-the-art learning methods require binary features. Features are separated into numerical and categorical types. Numerical features are discretized into four equal-sized bins and transformed via one-hot encoding, producing binary indicators for each bin. Categorical features are one-hot encoded with the first category dropped. In some cases, binarization leads to a large number of features. To reduce computational effort, we therefore conduct feature selection if needed. Features are ranked by their contribution to reducing impurity in a random forest model, and subsets of features are iteratively evaluated for validation accuracy using a forward stepwise selection approach. The subset is selected as the smallest feature set that maximizes predictive performance, see James et al. (2013). Below, we explicitly indicate when we did feature selection.

**Banana** is an artificial dataset where instances belong to two clusters with a banana shape. There are two features corresponding to the x and y axis, respectively. See `https://sci2s.ugr.es/keel/dataset.php?cod=182` for the data repository.

**Breast cancer** is a dataset sourced from the University Medical Centre, Institute of Oncology in Ljubljana (Slovenia) (Zwitter & Soklic, 1988). The challenge is to forecast recurrence events for patients with breast cancer stages I to III, see `https://doi.org/10.24432/C51P4M`.

**Diabetes** is a dataset from the National Institute of Diabetes and Digestive and Kidney Disease for which we need to predict —based on diagnostic measurements— whether a patient has diabetes (Turney, 1990). All patients here are females at least 21 years old of Pima Indian heritage, see `https://10.0.68.224/7zcc8v6hvp.1`.

A **solar flare** is a sudden burst of energy and radiation from the sun. The data contains observations of solar activity, specifically focusing on sunspots and solar flares, and the challenge is to predict the occurrence of a type of solar flare, see `http://dx.doi.org/10.5281/zenodo.18110`.

**German credit** is a dataset of customers who take a credit from a bank. Each customer is classified as having good or bad credit risks according to a set of customer attributes (Hofmann, 1994), see `https://doi.org/10.24432/C5NC77`.

**Image** is a dataset with the task to determine the type of surface of each region in an image. The instances are drawn randomly from a database of 7 outdoor images. The images were hand-segmented to create a classification for every pixel. Each instance is a $3 \times 3$ region, see `https://doi.org/10.24432/C5GP4N`.

The **heart** dataset has as challenge to predict heart disease in patients, it is a combined set from four locations: Cleveland, Hungary, Switzerland, and Long Beach VA (Janosi et al., 1989), see `https://doi.org/10.24432/C52P4X`.

The **ringnorm** dataset is an artificial binary classification dataset. The challenge is to classify two 20-dimensional Gaussian distributions with $N(0, 4\mathbf{I})$ and $N(\mu, \mathbf{I})$, with $\mu = (a, a, \ldots, a)$ and $a = 1/\sqrt{20}$, see `http://dx.doi.org/10.5281/zenodo.18110`.

The problem posed in the **splice** dataset is to recognize the boundaries between exons (the parts of DNA sequence retained after splicing) and introns (the parts of the DNA sequence that are spliced out). Splicing

is the removal of superfluous points on a DNA sequence during protein creation. We reduced the number of binarized features from 240 to 61, see `https://doi.org/10.24432/C5M888`.

The task in the **thyroid** dataset is to detect if a given patient is healthy or suffers from hypothyroidism. The data is originally sourced from the Garavan Institute in Sydney, Australia (Quinlan, 1986), see `https://doi.org/10.24432/C5D010`.

The task for the **titanic** dataset is to predict if a passenger survived the Titanic shipwreck based on passenger attributes. We did not use the benchmark version by Rätsch et al. (2001), as this contains only 24 data points, and used the version publicly available with 887 data points, see `http://dx.doi.org/10.5281/zenodo.18110`. We reduced the number of binarized features from 333 to 94.

The **twonorm** dataset is an artificial binary classification dataset. The challenge is to classify two 20-dimensional Gaussian distributions with means $(a, a, \ldots, a)$ and $(-a, -a, \ldots, -a)$ and $a = 2/\sqrt{20}$, see `http://dx.doi.org/10.5281/zenodo.18110`.

**Waveform** is an artificial binary classification dataset of waveform data, with each sample containing 40 attributes, with added noise. Each class is a random convex combination of two of the waveforms, see `http://dx.doi.org/10.5281/zenodo.18110`.

Below, we describe the datasets that are not part of the benchmark by Rätsch et al. (2001), but were selected by us as a supplementary challenge.

**Adult** is a well-known classification dataset for predicting whether annual income of an individual exceeds \$50K per year based on census data (Becker & Kohavi, 1996), see `https://doi.org/10.24432/C5XW20`. The dataset is studied in many related binary classification studies, among others in Demiriz et al. (2002) and Zhai et al. (2013).

Furthermore, we utilize the `folktables` library to construct four additional datasets based on census data. This library was constructed to address limitations of the *adult* dataset, such as outdated feature encodings and overly specific or unrepresentative target thresholds. We use two standardized challenges, see Ding et al. (2024): (i) predict whether a low-income individual, not eligible for Medicare, has coverage from public health insurance (**public coverage**), and (ii) predict whether an adult is employed (**employment**). We study both challenges in two states, California (CA) and Texas (TX), which exhibit significantly different population characteristics in terms of demographics, income distributions, and socio-economic diversity. For both challenges, we randomly sampled 25% of the 2018 data to keep the dataset size manageable. For the public coverage datasets, we left out redundant features and reduced from 622 to 57 (California) and from 591 to 130 (Texas). Datasets can be retrieved via the `folktables` Python library, using the specific challenges as mentioned in Ding et al. (2024).

**Compas** (Correctional Offender Management Profiling for Alternative Sanctions) is a dataset for which we need to predict criminal defendant's likelihood of reoffending in Broward County, Florida, see `https://www.kaggle.com/datasets/danofer/compass`. We preprocessed the data such that we remain with 14 features.

Finally, we include **secondary mushroom**, a dataset of simulated mushrooms for binary classification into edible and poisonous. Compared to the primary mushroom dataset, this simulated data contains $7.5\times$ more examples, which helps us to study scalability of algorithms to larger datasets (Wagner et al., 2021), see `https://doi.org/10.24432/C5FP5Q`. We reduced the number of binarized features from 111 to 63.

Table 3: Summary of the datasets

|  | Dataset | Features | Data Points | Class distribution |
|---|---|---|---|---|
| Benchmarks | banana | 8 | 5300 | 55.1% |
|  | breast cancer | 36 | 263 | 72.6% |
|  | diabetes | 112 | 768 | 65.1% |
|  | german credit | 80 | 1000 | 72.1% |
|  | heart | 52 | 270 | 57.4% |
|  | image | 72 | 2086 | 55.0% |
|  | ringnorm | 80 | 7400 | 50.9% |
|  | solar flare | 36 | 144 | 57.6% |
|  | splice | 61 | 2991 | 55.0% |
|  | thyroid | 20 | 215 | 70.2% |
|  | titanic | 94 | 887 | 61.4% |
|  | twonorm | 80 | 7400 | 50.7% |
|  | waveform | 84 | 5000 | 66.6% |
| Supplementary | adult | 19 | 48,842 | 76.1% |
|  | compas | 14 | 7206 | 54.9% |
|  | employment CA2018 | 84 | 75,660 | 57.3% |
|  | employment TX2018 | 84 | 52,089 | 58.0% |
|  | public coverage CA2018 | 57 | 34,638 | 63.2% |
|  | public coverage TX2018 | 130 | 24,732 | 81.2% |
|  | secondary mushroom | 63 | 61,069 | 55.2% |

## B  Formulations

In this section, we provide the mathematical programming formulations of the benchmarked LP-based boosting techniques. In most cases, except for QRLP-Boost and ERLP-Boost, we solve the primal formulation because these are often simpler in terms of variables and constraints than their dual counterpart and therefore, faster to solve, as shown in Shen et al. (2013). Nonetheless, we provide both the primal and dual formulations for most methods.

Table 4: Notation overview for primal formulations.

| Primal notation | Meaning |
|---|---|
| $x_i$ | features for data point $i$ |
| $y_i$ | label for data point $i$ |
| $h_j$ | base learner $j$, part of the set $\mathbf{H}$ |
| $w_j$ | weight for each base learner $h_j$ in the ensemble |
| $\rho_i$ | margin for data point $i$ |
| $\xi_i$ | slack variable for each data point $i$ |
| $M$ | number of data points in the training set |
| $T$ | number of base learners in $\mathbf{H}$ |
| $C$ | regularization hyperparameter |

We adapt and unify the notation from the original papers to match the one introduced in Section 2. The notation used for the primal formulations is summarized in Table 4. For the duals, we use the variable $u_i$ to denote the sample weights, i.e., a score provided to each data point $i$ in the dataset that indicates misclassification or distance to the decision boundary for the current ensemble $\mathbf{H}$ (and is usually used to weight the examples when fitting a new base learner).

### B.1 LP-Boost

LP-Boost is proposed in Demiriz et al. (2002). The tradeoff hyperparameter $C$ is tuned to balance between misclassification error and margin maximization. In Demiriz et al. (2002), multiple alternative but equivalent formulations are provided; we found the following formulation to be the most robust in terms of hyperparameter sensitivity. The LP-Boost primal problem is formulated as:

$$\text{minimize}_{w,\xi} \quad \sum_{j=1}^{T} w_j + C \sum_{i=1}^{M} \xi_i \tag{18}$$

$$\text{subject to} \quad y_i \sum_{j=1}^{T} w_j \, h_j(x_i) + \xi_i \geq 1, \quad \forall i = 1, \ldots, M, \tag{19}$$

$$w_j \geq 0, \quad \forall j = 1, \ldots, T, \tag{20}$$

$$\xi_i \geq 0, \quad \forall i = 1, \ldots, M. \tag{21}$$

The corresponding dual formulation is:

$$\text{maximize}_u \quad \sum_{i=1}^{M} u_i \tag{22}$$

$$\text{subject to} \quad \sum_{i=1}^{M} u_i \, y_i \, h_j(x_i) \leq 1, \quad \forall j = 1, \ldots, T, \tag{23}$$

$$0 \leq u_i \leq C, \quad \forall i = 1, \ldots, M. \tag{24}$$

### B.2 CG-Boost

CG-Boost is a quadratic program proposed in Bi et al. (2004). Just as in LP-Boost, $C$ is tuned to balance misclassification error and margin maximization, but CG-Boost regularizes the weights $w$ with the $L_2$ norm instead of the $L_1$ norm.

$$\text{minimize}_{w,\xi} \quad \frac{1}{2} \sum_{j=1}^{T} w_j^2 + C \sum_{i=1}^{M} \xi_i \tag{25}$$

$$\text{subject to} \quad y_i \sum_{j=1}^{T} w_j \, h_j(x_i) + \xi_i \geq 1, \quad \forall i = 1, \ldots, M, \tag{26}$$

$$w_j \geq 0, \quad \forall j = 1, \ldots, T, \tag{27}$$

$$\xi_i \geq 0, \quad \forall i = 1, \ldots, M. \tag{28}$$

For the corresponding dual, the primal variable $w$ remains in the dual:

$$\text{maximize}_u \; \text{minimize}_w \quad \sum_{i=1}^{M} u_i - \frac{1}{2} \sum_{j=1}^{T} w^2 \tag{29}$$

$$\text{subject to} \quad \sum_{i=1}^{M} u_i \, y_i \, h_j(x_i) \leq w_j, \quad \forall j = 1, \ldots, T, \tag{30}$$

$$0 \leq u_i \leq C, \quad \forall i = 1, \ldots, M. \tag{31}$$

### B.3 ERLP-Boost

A line of research has focused on developing alternative LP-Boost formulations with proven iteration bounds. Total-Boost (Warmuth et al., 2006) and Soft-Boost (Rätsch et al., 2007) have, in contrast to LP-Boost, an iteration bound, of $\mathcal{O}(\frac{2}{\epsilon^2} \ln \frac{M}{C})$ iterations to produce a linear combination of base learners for which the soft margin is within $\epsilon$ of the maximum minimum soft margin. Total-Boost and Soft-Boost only differ in the capping parameter, as Total-Boost uses $C = 1$, whereas Soft-Boost has $C \in [1, M]$.

As mentioned in Section 3, ERLP-Boost (Warmuth et al., 2008) performs similarly to LP-Boost and Soft-Boost overall but offers a theoretical bound on the number of iterations and typically achieves a faster reduction in error than Soft-Boost during the early stages of training. In ERLP-Boost, the hyperparameter $C$ balances the tradeoff between entropy regularization and the original LP-Boost objective (the worst-case margin), by constraining the maximum weight assigned to each example in the distribution.

We solve the following convex programming dual in an iterative fashion, i.e., we solve and obtain a new distribution $\mathbf{u}$ in each iteration until convergence. This makes ERLP-Boost potentially slower in terms of computational time per generated base learner.

$$\text{minimize}_{u,\xi} \quad \sum_{i=1}^{M} \xi_i + \frac{1}{\eta}\Delta(\mathbf{u}, \mathbf{u}^0), \tag{32}$$

$$\text{subject to} \quad \sum_{i=1}^{M} u_i y_i h_j(x_i) \leq \xi_i, \quad \forall j = 1, \ldots, T, \tag{33}$$

$$\sum_{i=1}^{M} u_i = 1, \tag{34}$$

$$0 \leq u_i \leq \frac{1}{C}, \quad \forall i = 1, \ldots, M. \tag{35}$$

Here, $\eta$ is a small constant calculated by:

$$\max\left(0.5, \frac{\ln M}{\frac{1}{2}\epsilon^{\text{stop}}}\right), \tag{36}$$

with $\epsilon^{\text{stop}}$ a small constant also used to set the maximum number of iterations and the convergence criterion:

$$\text{Maximum iterations} = \max\left(\frac{4}{\epsilon^{\text{stop}}/2}, \frac{8}{(\epsilon^{\text{stop}}/2)^2}\right). \tag{37}$$

We obtain the relative entropy term $\Delta(u^0, u)$ as follows:

$$\sum_{i=1}^{M} u_i \left(\log(u_i^0 + \epsilon) + \frac{u_i - u_i^0}{u_i^0 + \epsilon}\right), \tag{38}$$

which is the first order expanded KL-divergence with a small perturbation on $\mathbf{u}^0$ to prevent issues near zero. Note that we obtain the weights $w_j$ as the dual values of the soft margin constraint after convergence or the maximum number of iterations are reached.

### B.4 MD-Boost

MD-Boost is proposed in Shen & Li (2010) and focuses on the *distribution* of margins. Here, the tradeoff hyperparameter $C$ balances the normalized average margin and the normalized margin variance, i.e., the first and second moments of the margin distribution, respectively. The matrix $A \in \mathbb{R}^{M \times M}$ is given by:

$$A = \begin{bmatrix} 1 & -\frac{1}{M-1} & \cdots & -\frac{1}{M-1} \\ -\frac{1}{M-1} & 1 & \cdots & -\frac{1}{M-1} \\ \vdots & \vdots & \ddots & \vdots \\ -\frac{1}{M-1} & -\frac{1}{M-1} & \cdots & 1 \end{bmatrix}. \tag{39}$$

The calculations with $A$ can place a large computational burden when the number of data points $M$ is large. As suggested in Shen & Li (2010), we can approximate $A$ by its identity matrix. In line with the original work, we decided to use this approximation for all datasets, as we observed no performance decrease but a significant speedup.

$$\text{maximize}_{w,\rho} \quad \sum_{i=1}^{M} \rho_i - \frac{1}{2}\, \rho^{\top} A \rho \tag{40}$$

$$\text{subject to} \quad \rho_i = y_i \sum_{j=1}^{T} w_j\, h_j(x_i), \quad \forall\, i = 1,\dots,M, \tag{41}$$

$$\sum_{j=1}^{T} w_j = C, \quad \forall\, j = 1,\dots,T \tag{42}$$

$$w_j \geq 0, \quad \forall\, j = 1,\dots,T. \tag{43}$$

The dual formulation is given by:

$$\text{minimize}_{r,u} \quad r + \frac{1}{2C}\,(\mathbf{u}-\mathbf{1})^{\top} A^{-1}(\mathbf{u}-\mathbf{1}) \tag{44}$$

$$\text{subject to} \quad \sum_{i=1}^{M} u_i\, y_i\, h_j(x_i) \leq r, \quad \forall\, j = 1,\dots,T. \tag{45}$$

Note that the dual variable $\mathbf{u}$ is unbounded and is no longer a distribution, see Shen & Li (2010).

## C  Hyperparameters

Hyperparameter tuning is conducted as follows. We split the dataset into a 60% training set, 20% validation set, and 20% test set. We perform 5-fold cross-validation for each dataset and method, evaluating 10 hyperparameter values uniformly spaced within the ranges shown in Table 5. This way, we ensure that each method has a similar opportunity to find good models. We select the best hyperparameter based on the validation set performance, and report statistics on the test set.

Note that, except for ERLP-Boost and QRLP-Boost, each totally corrective method has only a single hyperparameter. For both ERLP-Boost and QRLP-Boost, we decided to only tune the trade-off parameter $C$ between relative entropy and the maximum edge, and set $\epsilon$ to 0.01, as this is also done this way in the original ERLP-Boost paper, see Warmuth et al. (2008). For MD-Boost, we decided to use the Moore-Penrose pseudo inverse approximation of the matrix $A$, as this did not yield an observable performance decrease and does decrease computational time significantly.

For the Adaboost, XGBoost, and LightGBM benchmarks, we solely tune the learning rate.

## D  Complementary Results

In this section, we provide detailed and complementary results. First, in Section D.1, we report computational times per dataset and method, and in Section D.2, we report the global performances (testing accuracy and sparsity) of all studied boosting approaches across all considered datasets, for depths-3 and depth-10 CART decision tree base learners, respectively. We display in Section D.3 the anytime predictive performance of the different benchmarked methods on the *adult* dataset. Afterward, in Section D.4, we show the performance across hyperparameter values for *all* datasets. Next, we provide the detailed results for the reweighting of Adaboost ensembles experiment in Section D.5. We display the complete results for the confidence-rated voting experiment in Section D.6, and finally, we provide the full results for the optimal decision trees experiments in Section D.7.

Table 5: Hyperparameter ranges used for each method (applied consistently across datasets).

| Method | Hyperparameter range |
|---|---|
| NM-Boost | $\{10^{-4}, \ldots, 10^{-0.33}\}$ |
| QRLP-Boost | $\{1, \ldots, 0.06M\}$ |
| LP-Boost | $\{10^{-4}, \ldots, 10^{-0.33}\}$ |
| CG-Boost | $\{10^{-4}, \ldots, 10^{-0.33}\}$ |
| ERLP-Boost | $\{1, \ldots, 0.06M\}$ |
| MD-Boost | $\{1, \ldots, 120\}$ |
| Adaboost | $\{10^{-3}, \ldots, 1\}$ |
| XGBoost | $\{10^{-3}, \ldots, 1\}$ |
| LightGBM | $\{10^{-3}, \ldots, 1\}$ |

## D.1 Computational Time

Table 6 shows the total CPU time for obtaining an ensemble of CART trees for each dataset and method. We only report results for depth-1 base learners as we do not observe a significant difference in CPU times between different depths. Note that this was expected, since considering more complex base learners does not increase the search space (number of variables and their domains) of the optimization problems underlying totally corrective formulations (it only affects the base learners' building, which is negligible using CART trees). Note that we disabled early stopping for all totally corrective methods to allow each to construct ensembles with the same number of base learners. Therefore, CPU times could be significantly reduced, as in most cases the totally corrective methods converged before 100 iterations, see, for instance, Figures 2 and 3.

As expected, the heuristic methods require only a fraction of the time compared to the totally corrective methods to train. For the smaller datasets ($\leq 1000$ examples) the training time is less than a minute, but the larger datasets require significantly more time. Nevertheless, even for the largest datasets (with up to 75,660 examples and 130 features, as detailed in Table 3), the running times remain practically reasonable, especially given the observed improvement in the accuracy/sparsity trade-off (as summarized in Figure 1).

Table 6: Computational times in seconds for different boosting methods using CART trees of **depth 1**, averaged over five seeds. The last row shows the mean and median CPU time in seconds.

| Dataset | NM-Boost CPU time | QRLP-Boost CPU time | LP-Boost CPU time | CG-Boost CPU time | ERLP-Boost CPU time | MD-Boost CPU time | Adaboost CPU time | XGBoost CPU time | LightGBM CPU time |
|---|---|---|---|---|---|---|---|---|---|
| banana | $434.5 \pm 2.1$ | $235.8 \pm 1.0$ | $425.3 \pm 2.4$ | $433.4 \pm 2.0$ | $156.2 \pm 1.3$ | $436.1 \pm 3.9$ | $0.2 \pm 0.0$ | $0.2 \pm 0.0$ | $0.0 \pm 0.0$ |
| breast cancer | $21.5 \pm 0.3$ | $13.0 \pm 0.3$ | $21.7 \pm 0.4$ | $22.3 \pm 0.1$ | $9.2 \pm 0.3$ | $22.4 \pm 0.2$ | $0.1 \pm 0.0$ | $0.2 \pm 0.0$ | $0.0 \pm 0.0$ |
| diabetes | $63.1 \pm 0.5$ | $37.0 \pm 0.3$ | $63.4 \pm 0.7$ | $66.3 \pm 0.7$ | $24.3 \pm 0.4$ | $65.3 \pm 0.6$ | $0.2 \pm 0.0$ | $0.2 \pm 0.0$ | $0.0 \pm 0.0$ |
| german credit | $83.1 \pm 1.5$ | $47.8 \pm 0.5$ | $83.1 \pm 1.0$ | $86.5 \pm 1.0$ | $31.5 \pm 0.6$ | $84.1 \pm 0.9$ | $0.2 \pm 0.0$ | $0.2 \pm 0.0$ | $0.0 \pm 0.0$ |
| heart | $22.9 \pm 0.3$ | $13.5 \pm 0.1$ | $22.5 \pm 0.2$ | $23.2 \pm 0.3$ | $9.1 \pm 0.4$ | $23.0 \pm 0.4$ | $0.1 \pm 0.0$ | $0.2 \pm 0.0$ | $0.0 \pm 0.0$ |
| image | $173.5 \pm 1.0$ | $95.8 \pm 0.5$ | $169.3 \pm 0.7$ | $172.0 \pm 1.6$ | $62.3 \pm 0.5$ | $104.4 \pm 83.5$ | $0.2 \pm 0.0$ | $0.2 \pm 0.0$ | $0.0 \pm 0.0$ |
| ringnorm | $640.4 \pm 6.0$ | $345.2 \pm 2.2$ | $620.6 \pm 5.0$ | $611.6 \pm 2.5$ | $215.2 \pm 2.4$ | $610.2 \pm 3.9$ | $0.5 \pm 0.0$ | $0.2 \pm 0.0$ | $0.0 \pm 0.0$ |
| solar flare | $12.5 \pm 0.1$ | $7.6 \pm 0.3$ | $12.2 \pm 0.1$ | $12.6 \pm 0.1$ | $5.2 \pm 0.6$ | $12.5 \pm 0.2$ | $0.1 \pm 0.0$ | $0.2 \pm 0.0$ | $0.0 \pm 0.0$ |
| splice | $257.6 \pm 2.4$ | $140.6 \pm 1.0$ | $244.9 \pm 1.7$ | $246.2 \pm 2.6$ | $89.9 \pm 1.0$ | $245.8 \pm 2.0$ | $0.3 \pm 0.0$ | $0.2 \pm 0.0$ | $0.0 \pm 0.0$ |
| thyroid | $18.1 \pm 0.3$ | $10.5 \pm 0.3$ | $18.0 \pm 0.3$ | $18.6 \pm 0.1$ | $7.2 \pm 0.2$ | $18.3 \pm 0.2$ | $0.1 \pm 0.0$ | $0.2 \pm 0.0$ | $0.0 \pm 0.0$ |
| titanic | $73.5 \pm 0.6$ | $42.3 \pm 0.2$ | $71.5 \pm 0.4$ | $76.5 \pm 0.8$ | $27.3 \pm 0.5$ | $75.1 \pm 0.6$ | $0.2 \pm 0.0$ | $0.2 \pm 0.0$ | $0.0 \pm 0.0$ |
| twonorm | $646.3 \pm 3.8$ | $343.1 \pm 1.5$ | $618.3 \pm 4.8$ | $611.7 \pm 4.1$ | $217.4 \pm 1.2$ | $613.3 \pm 4.7$ | $0.5 \pm 0.0$ | $0.2 \pm 0.0$ | $0.0 \pm 0.0$ |
| waveform | $433.1 \pm 2.6$ | $231.5 \pm 2.0$ | $408.6 \pm 3.2$ | $410.1 \pm 3.8$ | $148.2 \pm 0.7$ | $409.6 \pm 2.3$ | $0.4 \pm 0.0$ | $0.2 \pm 0.0$ | $0.0 \pm 0.0$ |
| adult | $2481.4 \pm 89.6$ | $2220.2 \pm 23.5$ | $2618.7 \pm 99.7$ | $2617.9 \pm 96.1$ | $1485.8 \pm 5.4$ | $2670.4 \pm 18.4$ | $1.1 \pm 0.2$ | $0.3 \pm 0.0$ | $0.1 \pm 0.0$ |
| compas | $588.4 \pm 3.2$ | $320.6 \pm 2.0$ | $579.7 \pm 2.0$ | $586.9 \pm 5.6$ | $214.1 \pm 1.2$ | $589.4 \pm 6.0$ | $0.2 \pm 0.0$ | $0.2 \pm 0.0$ | $0.0 \pm 0.0$ |
| employment CA2018 | $2618.4 \pm 19.5$ | $2516.0 \pm 118.4$ | $2467.1 \pm 15.0$ | $2370.0 \pm 8.1$ | $2273.2 \pm 13.3$ | $2363.6 \pm 17.6$ | $2.6 \pm 0.0$ | $0.7 \pm 0.0$ | $0.3 \pm 0.0$ |
| employment TX2018 | $2614.5 \pm 126.1$ | $2413.0 \pm 34.2$ | $2548.1 \pm 9.1$ | $2507.8 \pm 27.7$ | $1562.1 \pm 12.8$ | $2496.8 \pm 28.7$ | $1.8 \pm 0.0$ | $0.7 \pm 0.2$ | $0.2 \pm 0.0$ |
| public coverage CA2018 | $2527.5 \pm 3.2$ | $1594.0 \pm 8.2$ | $2612.0 \pm 96.2$ | $2647.7 \pm 10.7$ | $1083.4 \pm 7.0$ | $2661.9 \pm 8.3$ | $1.1 \pm 0.1$ | $0.4 \pm 0.1$ | $0.1 \pm 0.0$ |
| public coverage TX2018 | $2068.9 \pm 19.8$ | $1125.5 \pm 4.6$ | $2039.5 \pm 12.7$ | $2052.6 \pm 13.0$ | $755.0 \pm 3.9$ | $2033.1 \pm 16.4$ | $1.4 \pm 0.0$ | $0.4 \pm 0.0$ | $0.1 \pm 0.0$ |
| mushroom secondary | $2451.2 \pm 7.4$ | $2581.5 \pm 116.2$ | $2599.1 \pm 127.5$ | $2586.1 \pm 16.4$ | $1835.9 \pm 16.9$ | $2569.6 \pm 17.8$ | $2.1 \pm 0.3$ | $0.5 \pm 0.0$ | $0.2 \pm 0.0$ |
| Mean/Median | 911.5/433.8 | 716.7/233.7 | 912.2/417.0 | 908.0/421.8 | 510.6/152.2 | 905.2/422.9 | 0.7/0.2 | 0.3/0.2 | 0.1/0.0 |

## D.2 Accuracy-Sparsity Performances

We report in Tables 7 and 8 the global performances (testing accuracy and sparsity) of the different totally corrective boosting methods and benchmarked heuristics, for all considered datasets, for depths-3 and depth-10 CART decision tree base learners, respectively. They complement the average results and the per-dataset results for depths-1 and depth-5 CART decision tree base learners provided and discussed in Section 6.1.

As can be observed in Table 8, Adaboost stopped after one iteration on the *heart* dataset with depth-10 trees due to perfect training accuracy. While this results in a particularly sparse and interpretable model (i.e., a single decision tree), we note that including additional trees enhances generalization —as for this experiment, the other boosting approaches all reach better testing accuracies than Adaboost.

Table 7: Testing accuracy and number of non-zero weights for different boosting methods using CART trees of **depth 3**, averaged over five seeds. **Bold** highlights the best accuracy among *all* methods, while a star* marks the best among *totally corrective* methods. The last row shows the mean and median for both statistics.

| Dataset | NM-Boost Acc. | Cols. | QRLP-Boost Acc. | Cols. | LP-Boost Acc. | Cols. | CG-Boost Acc. | Cols. | ERLP-Boost Acc. | Cols. | MD-Boost Acc. | Cols. | Adaboost Acc. | Cols. | XGBoost Acc. | Cols. | LightGBM Acc. | Cols. |
|---|---|---|---|---|---|---|---|---|---|---|---|---|---|---|---|---|---|---|
| banana | 0.670 ± 0.014 | 14 | 0.670 ± 0.014 | 77 | 0.670 ± 0.014 | 11 | **0.675* ± 0.018** | 100 | 0.670 ± 0.014 | 77 | 0.670 ± 0.014 | 16 | 0.671 ± 0.015 | 100 | 0.672 ± 0.015 | 100 | **0.675 ± 0.018** | 100 |
| breast cancer | **0.857* ± 0.049** | 11 | 0.834 ± 0.040 | 86 | 0.819 ± 0.035 | 15 | 0.819 ± 0.035 | 100 | 0.792 ± 0.036 | 12 | 0.804 ± 0.054 | 9 | 0.838 ± 0.041 | 100 | 0.823 ± 0.031 | 100 | 0.842 ± 0.009 | 100 |
| diabetes | 0.765* ± 0.023 | 9 | 0.729 ± 0.027 | 91 | 0.755 ± 0.017 | 21 | 0.735 ± 0.008 | 100 | 0.751 ± 0.022 | 30 | 0.748 ± 0.018 | 26 | 0.774 ± 0.016 | 100 | 0.771 ± 0.020 | 100 | **0.777 ± 0.020** | 100 |
| german credit | 0.878* ± 0.021 | 58 | 0.856 ± 0.021 | 93 | 0.870 ± 0.023 | 62 | 0.869 ± 0.029 | 100 | 0.833 ± 0.022 | 23 | 0.814 ± 0.023 | 14 | 0.875 ± 0.029 | 100 | 0.870 ± 0.016 | 100 | **0.881 ± 0.018** | 100 |
| heart | 0.870 ± 0.020 | 9 | 0.848 ± 0.038 | 38 | 0.881* ± 0.034 | 23 | 0.867 ± 0.034 | 100 | 0.837 ± 0.061 | 11 | 0.863 ± 0.042 | 23 | 0.885 ± 0.027 | 100 | **0.911 ± 0.014** | 100 | **0.911 ± 0.025** | 100 |
| image | 0.950* ± 0.005 | 41 | 0.947 ± 0.005 | 67 | 0.945 ± 0.014 | 34 | 0.946 ± 0.011 | 100 | 0.902 ± 0.011 | 12 | 0.893 ± 0.032 | 12 | 0.941 ± 0.010 | 100 | **0.957 ± 0.007** | 100 | 0.953 ± 0.006 | 100 |
| ringnorm | 0.934 ± 0.005 | 93 | 0.935* ± 0.002 | 100 | 0.928 ± 0.006 | 55 | 0.935* ± 0.005 | 100 | 0.900 ± 0.009 | 27 | 0.911 ± 0.007 | 31 | **0.943 ± 0.001** | 100 | **0.943 ± 0.001** | 100 | 0.940 ± 0.002 | 100 |
| solar flare | 0.690 ± 0.062 | 4 | 0.648 ± 0.070 | 24 | 0.669 ± 0.077 | 3 | 0.641 ± 0.089 | 100 | 0.655 ± 0.038 | 9 | 0.697* ± 0.086 | 9 | **0.697 ± 0.083** | 100 | 0.614 ± 0.091 | 100 | 0.593 ± 0.105 | 100 |
| splice | 0.969 ± 0.008 | 41 | 0.978* ± 0.007 | 100 | 0.973 ± 0.006 | 89 | 0.972 ± 0.007 | 100 | 0.964 ± 0.006 | 25 | 0.945 ± 0.008 | 16 | 0.979 ± 0.006 | 100 | **0.981 ± 0.006** | 100 | 0.980 ± 0.006 | 100 |
| thyroid | **0.944* ± 0.038** | 3 | 0.940 ± 0.035 | 7 | **0.944* ± 0.032** | 4 | 0.940 ± 0.035 | 100 | 0.940 ± 0.035 | 6 | **0.944* ± 0.032** | 8 | 0.935 ± 0.027 | 100 | **0.944 ± 0.032** | 100 | 0.907 ± 0.025 | 100 |
| titanic | 0.793 ± 0.030 | 3 | 0.766 ± 0.021 | 98 | 0.802 ± 0.024 | 36 | 0.751 ± 0.008 | 100 | 0.806* ± 0.026 | 19 | 0.798 ± 0.034 | 12 | 0.808 ± 0.024 | 100 | 0.816 ± 0.018 | 100 | **0.820 ± 0.028** | 100 |
| twonorm | 0.960 ± 0.004 | 95 | 0.960 ± 0.004 | 85 | 0.957 ± 0.002 | 52 | 0.961* ± 0.005 | 100 | 0.941 ± 0.003 | 24 | 0.924 ± 0.006 | 29 | 0.954 ± 0.006 | 100 | **0.964 ± 0.004** | 100 | 0.960 ± 0.005 | 100 |
| waveform | 0.900 ± 0.010 | 98 | 0.903 ± 0.008 | 100 | 0.903 ± 0.008 | 82 | 0.905* ± 0.011 | 100 | 0.888 ± 0.011 | 22 | 0.875 ± 0.008 | 18 | 0.893 ± 0.009 | 100 | **0.923 ± 0.006** | 100 | 0.915 ± 0.008 | 100 |
| adult | 0.815 ± 0.003 | 81 | 0.785 ± 0.013 | 2 | 0.817 ± 0.001 | 86 | 0.817 ± 0.001 | 91 | 0.820 ± 0.001 | 100 | **0.821* ± 0.001** | 40 | 0.820 ± 0.001 | 100 | 0.820 ± 0.001 | 100 | 0.820 ± 0.001 | 100 |
| compas | 0.668 ± 0.010 | 15 | 0.667 ± 0.015 | 100 | **0.671* ± 0.015** | 40 | 0.668 ± 0.018 | 100 | 0.667 ± 0.015 | 96 | 0.665 ± 0.014 | 58 | 0.668 ± 0.012 | 100 | 0.670 ± 0.013 | 100 | **0.671 ± 0.014** | 100 |
| employment CA2018 | 0.740 ± 0.003 | 21 | 0.751* ± 0.001 | 100 | 0.743 ± 0.002 | 21 | 0.744 ± 0.002 | 71 | 0.751* ± 0.002 | 73 | 0.736 ± 0.003 | 12 | 0.744 ± 0.002 | 100 | **0.754 ± 0.001** | 100 | 0.753 ± 0.001 | 100 |
| employment TX2018 | 0.747 ± 0.002 | 49 | 0.760 ± 0.005 | 29 | 0.751 ± 0.004 | 62 | 0.751 ± 0.001 | 86 | 0.761* ± 0.004 | 58 | 0.745 ± 0.012 | 23 | 0.753 ± 0.002 | 100 | **0.764 ± 0.003** | 100 | 0.763 ± 0.004 | 100 |
| public coverage CA2018 | 0.703 ± 0.006 | 71 | 0.689 ± 0.011 | 5 | 0.710 ± 0.004 | 98 | 0.709 ± 0.003 | 100 | 0.712* ± 0.002 | 100 | 0.708 ± 0.005 | 90 | 0.709 ± 0.003 | 100 | **0.716 ± 0.002** | 100 | 0.714 ± 0.001 | 100 |
| public coverage TX2018 | 0.849 ± 0.002 | 6 | 0.850 ± 0.003 | 100 | 0.851 ± 0.002 | 6 | 0.850 ± 0.002 | 100 | 0.852* ± 0.002 | 54 | 0.852* ± 0.002 | 22 | 0.851 ± 0.002 | 100 | **0.855 ± 0.002** | 100 | 0.854 ± 0.003 | 100 |
| mushroom secondary | 0.994 ± 0.003 | 72 | 0.995 ± 0.001 | 98 | 0.995 ± 0.002 | 77 | 0.996* ± 0.001 | 81 | 0.887 ± 0.009 | 24 | 0.830 ± 0.026 | 20 | 0.928 ± 0.014 | 100 | **0.997 ± 0.000** | 100 | 0.991 ± 0.000 | 100 |
| Mean/Median | **0.835*/0.853*** | 39.7/31 | 0.826/0.841 | 73.5/92 | 0.833/0.835 | 43.9/38 | 0.830/0.835 | 96.5/100 | 0.816/0.827 | 40.1/24.5 | 0.812/0.817 | 24.4/19 | 0.831/0.845 | 100/100 | **0.838**/0.839 | 100/100 | 0.836/0.848 | 100/100 |

Table 8: Testing accuracy and number of non-zero weights for different boosting methods using CART trees of **depth 10**, averaged over five seeds. **Bold** highlights the best accuracy among *all* methods, while a star* marks the best among *totally corrective* methods. The last row shows the mean and median for both statistics.

| Dataset | NM-Boost Acc. | Cols. | QRLP-Boost Acc. | Cols. | LP-Boost Acc. | Cols. | CG-Boost Acc. | Cols. | ERLP-Boost Acc. | Cols. | MD-Boost Acc. | Cols. | Adaboost Acc. | Cols. | XGBoost Acc. | Cols. | LightGBM Acc. | Cols. |
|---|---|---|---|---|---|---|---|---|---|---|---|---|---|---|---|---|---|---|
| banana | **0.670* ± 0.014** | 14 | **0.670* ± 0.014** | 2 | **0.670* ± 0.014** | 1 | **0.670* ± 0.014** | 100 | **0.670* ± 0.014** | 4 | **0.670* ± 0.014** | 49 | **0.670 ± 0.014** | 100 | **0.670 ± 0.014** | 100 | **0.670 ± 0.014** | 100 |
| breast cancer | 0.849* ± 0.043 | 4 | 0.830 ± 0.038 | 22 | 0.804 ± 0.029 | 41 | 0.838 ± 0.046 | 100 | 0.815 ± 0.028 | 7 | 0.789 ± 0.047 | 34 | **0.853 ± 0.044** | 100 | 0.838 ± 0.035 | 100 | 0.830 ± 0.032 | 100 |
| diabetes | 0.708 ± 0.011 | 3 | 0.716 ± 0.016 | 23 | 0.691 ± 0.032 | 67 | 0.740* ± 0.018 | 100 | 0.725 ± 0.027 | 10 | 0.738 ± 0.038 | 81 | **0.760 ± 0.000** | 100 | 0.738 ± 0.032 | 100 | 0.755 ± 0.019 | 100 |
| german credit | 0.875 ± 0.029 | 4 | 0.860 ± 0.030 | 19 | 0.875 ± 0.029 | 76 | 0.886* ± 0.021 | 100 | 0.882 ± 0.030 | 12 | 0.852 ± 0.031 | 99 | **0.894 ± 0.019** | 100 | 0.879 ± 0.024 | 100 | 0.890 ± 0.016 | 100 |
| heart | 0.837 ± 0.022 | 1 | 0.833 ± 0.031 | 1 | 0.833 ± 0.012 | 1 | 0.841 ± 0.025 | 100 | 0.844 ± 0.015 | 1 | 0.859* ± 0.049 | 15 | 0.830 ± 0.022 | 1 | **0.896 ± 0.025** | 100 | 0.893 ± 0.038 | 100 |
| image | 0.956 ± 0.008 | 7 | 0.951 ± 0.007 | 26 | 0.951 ± 0.008 | 53 | 0.951 ± 0.009 | 100 | 0.951 ± 0.009 | 10 | 0.953 ± 0.011 | 15 | **0.958 ± 0.005** | 100 | 0.955 ± 0.005 | 100 | 0.955 ± 0.006 | 100 |
| ringnorm | 0.920 ± 0.004 | 25 | 0.924 ± 0.007 | 90 | 0.915 ± 0.005 | 95 | 0.929* ± 0.006 | 100 | 0.920 ± 0.004 | 33 | 0.916 ± 0.010 | 34 | 0.937 ± 0.007 | 100 | 0.945 ± 0.004 | 100 | **0.951 ± 0.004** | 100 |
| solar flare | 0.655 ± 0.079 | 6 | 0.648 ± 0.051 | 74 | 0.641 ± 0.056 | 8 | 0.648 ± 0.059 | 100 | **0.710* ± 0.071** | 59 | 0.614 ± 0.126 | 24 | 0.611 ± 0.064 | 100 | 0.614 ± 0.088 | 100 | 0.593 ± 0.105 | 100 |
| splice | 0.974 ± 0.004 | 2 | 0.969 ± 0.011 | 17 | 0.966 ± 0.008 | 76 | **0.984* ± 0.004** | 100 | 0.968 ± 0.008 | 10 | 0.969 ± 0.015 | 31 | 0.980 ± 0.004 | 100 | 0.982 ± 0.004 | 100 | 0.983 ± 0.005 | 100 |
| thyroid | **0.944* ± 0.032** | 2 | 0.935 ± 0.027 | 14 | **0.944* ± 0.028** | 1 | 0.940 ± 0.035 | 100 | 0.940 ± 0.024 | 5 | **0.944* ± 0.032** | 16 | **0.944 ± 0.032** | 100 | **0.944 ± 0.032** | 100 | **0.944 ± 0.032** | 100 |
| titanic | 0.796 ± 0.025 | 6 | 0.792 ± 0.016 | 67 | 0.794 ± 0.036 | 37 | 0.807* ± 0.032 | 100 | 0.780 ± 0.031 | 14 | 0.793 ± 0.024 | 22 | 0.802 ± 0.022 | 100 | 0.809 ± 0.019 | 100 | **0.819 ± 0.018** | 100 |
| twonorm | 0.945 ± 0.008 | 12 | 0.954 ± 0.004 | 30 | 0.935 ± 0.007 | 95 | **0.968* ± 0.003** | 100 | 0.952 ± 0.004 | 36 | 0.957 ± 0.004 | 44 | 0.963 ± 0.004 | 100 | 0.966 ± 0.001 | 100 | **0.968 ± 0.003** | 100 |
| waveform | 0.924 ± 0.006 | 20 | 0.919 ± 0.006 | 29 | 0.913 ± 0.007 | 95 | 0.928* ± 0.005 | 100 | 0.917 ± 0.010 | 17 | 0.910 ± 0.007 | 15 | **0.931 ± 0.006** | 100 | 0.930 ± 0.008 | 100 | 0.928 ± 0.008 | 100 |
| adult | 0.815 ± 0.003 | 31 | 0.816* ± 0.002 | 1 | 0.815 ± 0.003 | 1 | 0.816* ± 0.002 | 96 | 0.816* ± 0.002 | 1 | 0.816 ± 0.003 | 86 | 0.816 ± 0.002 | 100 | 0.816 ± 0.003 | 100 | **0.820 ± 0.001** | 100 |
| compas | 0.664 ± 0.014 | 2 | 0.667* ± 0.012 | 100 | 0.664 ± 0.015 | 1 | 0.666 ± 0.014 | 100 | 0.667* ± 0.014 | 100 | 0.666 ± 0.015 | 93 | 0.664 ± 0.015 | 100 | 0.666 ± 0.012 | 100 | **0.668 ± 0.015** | 100 |
| employment CA2018 | 0.746 ± 0.003 | 16 | 0.744 ± 0.001 | 98 | 0.742 ± 0.002 | 1 | 0.754* ± 0.001 | 76 | 0.746 ± 0.002 | 53 | 0.749 ± 0.001 | 17 | 0.752 ± 0.002 | 100 | **0.757 ± 0.001** | 100 | 0.754 ± 0.001 | 100 |
| employment TX2018 | 0.754 ± 0.005 | 13 | 0.755 ± 0.003 | 88 | 0.750 ± 0.003 | 1 | 0.754* ± 0.003 | 91 | 0.754 ± 0.002 | 42 | 0.757 ± 0.004 | 17 | 0.762 ± 0.004 | 100 | **0.766 ± 0.003** | 100 | 0.765 ± 0.003 | 100 |
| public coverage CA2018 | 0.704 ± 0.003 | 34 | 0.709 ± 0.005 | 99 | 0.695 ± 0.005 | 100 | 0.721* ± 0.005 | 100 | 0.707 ± 0.002 | 60 | 0.713 ± 0.004 | 18 | 0.717 ± 0.003 | 100 | **0.719 ± 0.003** | 100 | 0.718 ± 0.003 | 100 |
| public coverage TX2018 | 0.850 ± 0.002 | 8 | 0.855 ± 0.003 | 56 | 0.846 ± 0.002 | 1 | 0.856* ± 0.002 | 100 | 0.849 ± 0.003 | 21 | 0.848 ± 0.003 | 12 | 0.856 ± 0.002 | 100 | **0.859 ± 0.003** | 100 | 0.856 ± 0.004 | 100 |
| mushroom secondary | 0.999* ± 0.000 | 19 | 0.999* ± 0.000 | 29 | 0.999* ± 0.000 | 73 | 0.999* ± 0.000 | 86 | 0.993 ± 0.002 | 14 | 0.968 ± 0.024 | 85 | **0.999 ± 0.000** | 100 | **0.999 ± 0.000** | 100 | **0.999 ± 0.000** | 100 |
| Mean/Median | 0.829/0.843* | 11.4/7.5 | 0.827/0.831 | 44.2/29 | 0.822/0.824 | 41.2/39 | 0.836*/0.839 | 97.5/100 | 0.830/0.830 | 25.4/14 | 0.824/0.832 | 40.4/27.5 | 0.836/0.841 | 95/100 | **0.837**/0.849 | 100/100 | 0.836/0.843 | 100/100 |

## D.3 Anytime Performance

We report in Figure 12 the testing accuracy of the different considered boosting approaches at each performed iteration during the ensemble training, for the *adult* dataset. This complements the results provided in Section 6.2 for the *image* and *ringnorm* datasets. Consistent with our previous observations, we note that the anytime performance of methods is especially different in early iterations, but gets close together for all methods later.

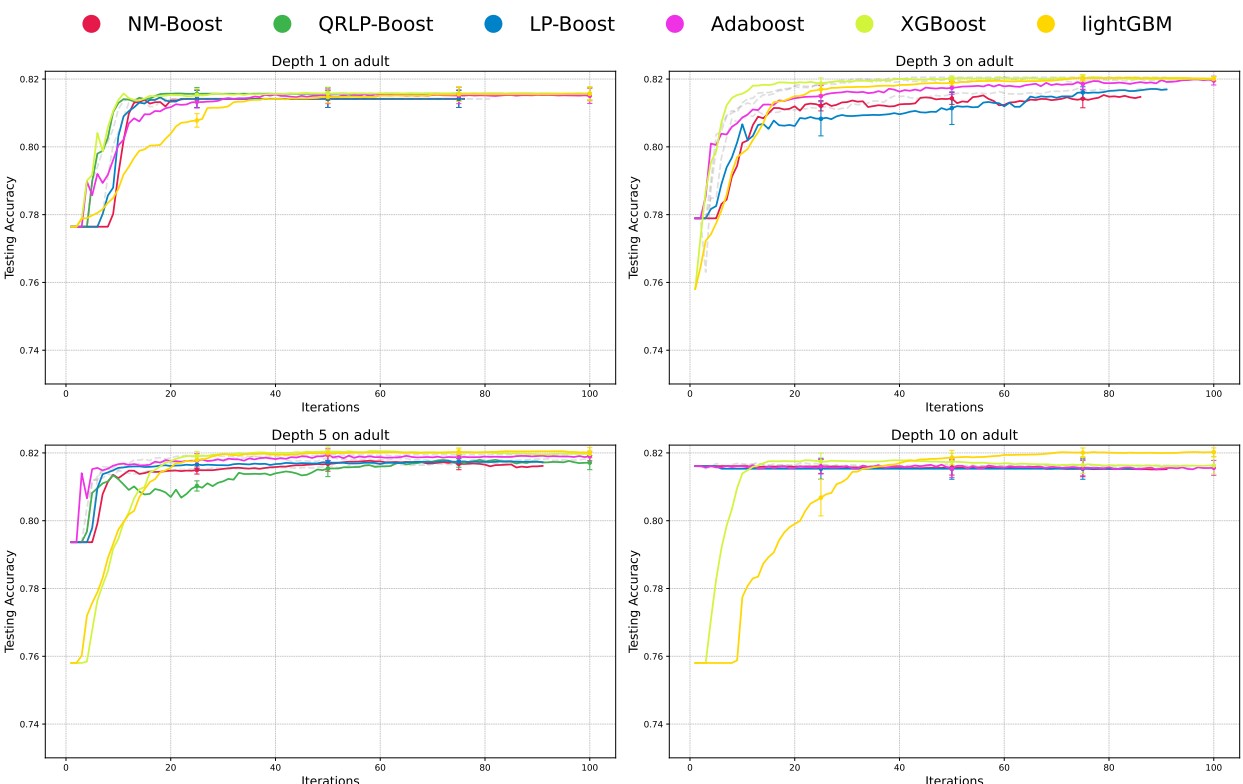

Figure 12: Anytime behavior on the **adult** dataset for selected methods (all other methods in gray), for CART decision trees of depth 1, 3, 5, and 10. Error bars indicate the standard deviation over 5 seeds.

### D.4   Hyperparameter Sensitivity

Figure 13 displays the trade-offs between testing accuracy and number of used columns (i.e., base learners), for the considered totally corrective boosting methods, for depth-1 decision trees, over all tested datasets and all hyperparameter values. This complements the results presented in Section 6.5 for two datasets and depth-1 and depth-5 trees.

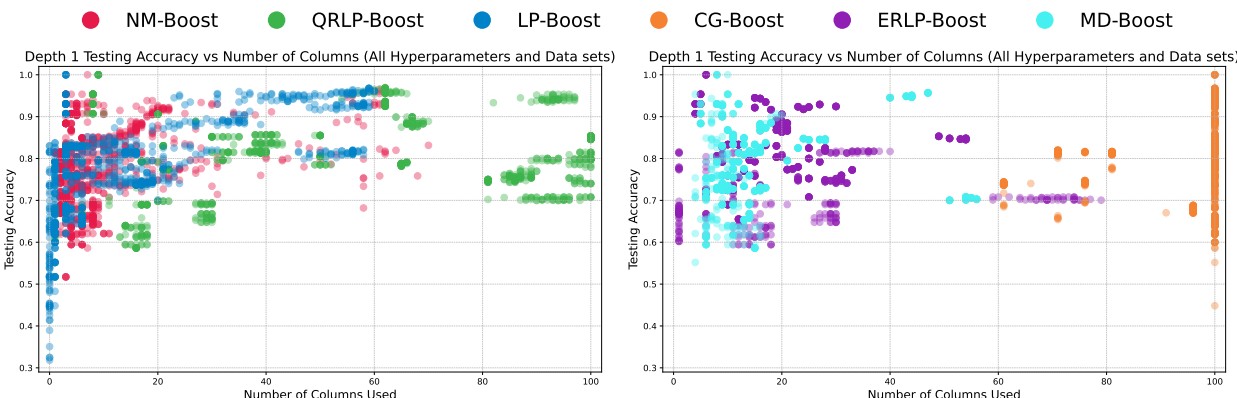

Figure 13: Average testing accuracy compared to average ensemble sparsity over **all datasets** and **all hyperparameter values** for depth 1 trees, over 5 seeds. The methods are split over two columns for visualization purposes.

## D.5    Reweighting Behavior

We provide the per-dataset results of our reweighting experiments in Tables 9, 10, 11, and 12, for trees of depth 1, 3, 5, and 10, respectively. This complements the aggregate results provided in Section 6.6.

Table 9: Testing accuracy and number of non-zero weights for different boosting methods for a **reweighting experiment** using 100 **depth 1** trees generated by Adaboost. A star* marks the best among *totally corrective* methods. The last row shows mean and median for both statistics. [†]Adaboost hyperparameters were not tuned for the reweighting experiment.

| | NM-Boost | | QRLP-Boost | | LP-Boost | | CG-Boost | | ERLP-Boost | | MD-Boost | | Adaboost[†] | |
|---|---|---|---|---|---|---|---|---|---|---|---|---|---|---|
| Dataset | Acc. | Cols. | Acc. | Cols. | Acc. | Cols. | Acc. | Cols. | Acc. | Cols. | Acc. | Cols. | Acc. | Cols. |
| banana | $0.625^* \pm 0.016$ | 2 | $0.620 \pm 0.016$ | 11 | $0.625^* \pm 0.016$ | 1 | $0.625^* \pm 0.016$ | 100 | $0.620 \pm 0.016$ | 11 | $0.620 \pm 0.016$ | 11 | $0.620 \pm 0.016$ | 100 |
| breast cancer | $0.830^* \pm 0.068$ | 5 | $0.792 \pm 0.075$ | 12 | $0.770 \pm 0.058$ | 0 | $0.789 \pm 0.054$ | 100 | $0.808 \pm 0.056$ | 12 | $0.800 \pm 0.054$ | 12 | $0.792 \pm 0.040$ | 100 |
| diabetes | $0.769^* \pm 0.029$ | 7 | $0.752 \pm 0.044$ | 36 | $0.755 \pm 0.033$ | 27 | $0.743 \pm 0.031$ | 100 | $0.752 \pm 0.038$ | 36 | $0.766 \pm 0.011$ | 36 | $0.761 \pm 0.033$ | 100 |
| german credit | $0.790 \pm 0.034$ | 20 | $0.803^* \pm 0.023$ | 33 | $0.793 \pm 0.017$ | 29 | $0.795 \pm 0.021$ | 100 | $0.800 \pm 0.038$ | 33 | $0.771 \pm 0.036$ | 33 | $0.808 \pm 0.022$ | 100 |
| heart | $0.807 \pm 0.043$ | 10 | $0.837^* \pm 0.022$ | 20 | $0.819 \pm 0.022$ | 11 | $0.826 \pm 0.019$ | 100 | $0.830 \pm 0.022$ | 20 | $0.763 \pm 0.036$ | 20 | $0.819 \pm 0.032$ | 100 |
| image | $0.816 \pm 0.027$ | 10 | $0.815 \pm 0.013$ | 13 | $0.813 \pm 0.029$ | 10 | $0.818 \pm 0.023$ | 100 | $0.821^* \pm 0.018$ | 13 | $0.787 \pm 0.023$ | 13 | $0.805 \pm 0.012$ | 100 |
| ringnorm | $0.912^* \pm 0.005$ | 29 | $0.899 \pm 0.006$ | 29 | $0.910 \pm 0.005$ | 29 | $0.910 \pm 0.004$ | 100 | $0.882 \pm 0.006$ | 29 | $0.885 \pm 0.005$ | 29 | $0.894 \pm 0.006$ | 100 |
| solar flare | $0.690 \pm 0.062$ | 7 | $0.710 \pm 0.035$ | 9 | $0.703 \pm 0.035$ | 3 | $0.717^* \pm 0.074$ | 100 | $0.703 \pm 0.047$ | 9 | $0.634 \pm 0.056$ | 9 | $0.710 \pm 0.056$ | 100 |
| splice | $0.944 \pm 0.008$ | 41 | $0.944 \pm 0.008$ | 44 | $0.945 \pm 0.006$ | 39 | $0.948^* \pm 0.005$ | 100 | $0.938 \pm 0.011$ | 44 | $0.944 \pm 0.005$ | 45 | $0.942 \pm 0.010$ | 100 |
| thyroid | $0.940 \pm 0.035$ | 6 | $0.926 \pm 0.009$ | 8 | $0.944^* \pm 0.032$ | 3 | $0.940 \pm 0.035$ | 100 | $0.926 \pm 0.009$ | 8 | $0.944^* \pm 0.032$ | 8 | $0.949 \pm 0.034$ | 100 |
| titanic | $0.794 \pm 0.019$ | 18 | $0.791 \pm 0.028$ | 28 | $0.799^* \pm 0.023$ | 3 | $0.789 \pm 0.024$ | 100 | $0.796 \pm 0.025$ | 28 | $0.797 \pm 0.022$ | 28 | $0.802 \pm 0.033$ | 100 |
| twonorm | $0.954 \pm 0.005$ | 50 | $0.940 \pm 0.001$ | 50 | $0.956^* \pm 0.002$ | 48 | $0.955 \pm 0.003$ | 100 | $0.924 \pm 0.003$ | 50 | $0.935 \pm 0.021$ | 50 | $0.945 \pm 0.002$ | 100 |
| waveform | $0.889^* \pm 0.006$ | 32 | $0.885 \pm 0.002$ | 34 | $0.888 \pm 0.005$ | 32 | $0.889^* \pm 0.005$ | 100 | $0.878 \pm 0.006$ | 34 | $0.856 \pm 0.013$ | 34 | $0.883 \pm 0.002$ | 100 |
| adult | $0.815 \pm 0.001$ | 10 | $0.816^* \pm 0.002$ | 21 | $0.814 \pm 0.003$ | 11 | $0.814 \pm 0.003$ | 100 | $0.815 \pm 0.002$ | 22 | $0.813 \pm 0.003$ | 22 | $0.815 \pm 0.002$ | 100 |
| compas propublica | $0.665 \pm 0.010$ | 9 | $0.672 \pm 0.017$ | 16 | $0.660 \pm 0.013$ | 4 | $0.660 \pm 0.013$ | 100 | $0.672 \pm 0.017$ | 16 | $0.673^* \pm 0.016$ | 16 | $0.670 \pm 0.016$ | 100 |
| employment CA2018 | $0.734 \pm 0.003$ | 6 | $0.737^* \pm 0.002$ | 12 | $0.734 \pm 0.003$ | 6 | $0.734 \pm 0.003$ | 100 | $0.736 \pm 0.000$ | 12 | $0.723 \pm 0.002$ | 12 | $0.723 \pm 0.004$ | 100 |
| employment TX2018 | $0.736 \pm 0.003$ | 6 | $0.738 \pm 0.001$ | 12 | $0.736 \pm 0.003$ | 6 | $0.736 \pm 0.003$ | 100 | $0.739^* \pm 0.003$ | 12 | $0.713 \pm 0.004$ | 12 | $0.731 \pm 0.003$ | 100 |
| public coverage CA2018 | $0.702^* \pm 0.004$ | 20 | $0.699 \pm 0.004$ | 29 | $0.680 \pm 0.005$ | 3 | $0.680 \pm 0.006$ | 100 | $0.698 \pm 0.004$ | 29 | $0.699 \pm 0.003$ | 29 | $0.695 \pm 0.004$ | 100 |
| public coverage TX2018 | $0.847^* \pm 0.003$ | 26 | $0.846 \pm 0.001$ | 28 | $0.825 \pm 0.006$ | 3 | $0.825 \pm 0.006$ | 100 | $0.846 \pm 0.001$ | 29 | $0.847^* \pm 0.002$ | 29 | $0.844 \pm 0.002$ | 100 |
| mushroom secondary | $0.739 \pm 0.002$ | 13 | $0.747 \pm 0.000$ | 16 | $0.739 \pm 0.002$ | 13 | $0.739 \pm 0.002$ | 100 | $0.748^* \pm 0.003$ | 16 | $0.741 \pm 0.002$ | 16 | $0.739 \pm 0.002$ | 100 |
| Mean/Median | $\mathbf{0.800^*}$ / 0.800 | 16.4 / 10.0 | 0.798 / 0.798 | 23.1 / 20.5 | 0.795 / 0.796 | 14.1 / 8.0 | 0.797 / 0.792 | 100.0 / 100.0 | 0.797 / $\mathbf{0.804}^*$ | 23.1 / 21.0 | 0.786 / 0.779 | 23.2 / 21.0 | 0.797 / 0.804 | 100 / 100 |

Table 10: Testing accuracy and number of non-zero weights for different boosting methods for a **reweighting experiment** using 100 **depth 3** trees generated by Adaboost. A star* marks the best among *totally corrective* methods. The last row shows the mean and median for both statistics. [†]Adaboost hyperparameters were not tuned for the reweight experiment.

| | NM-Boost | | QRLP-Boost | | LP-Boost | | CG-Boost | | ERLP-Boost | | MD-Boost | | Adaboost[†] | |
|---|---|---|---|---|---|---|---|---|---|---|---|---|---|---|
| Dataset | Acc. | Cols. | Acc. | Cols. | Acc. | Cols. | Acc. | Cols. | Acc. | Cols. | Acc. | Cols. | Acc. | Cols. |
| banana | $0.670^* \pm 0.014$ | 11 | $0.670^* \pm 0.014$ | 53 | $0.670^* \pm 0.014$ | 10 | $0.670^* \pm 0.014$ | 100 | $0.670^* \pm 0.014$ | 53 | $0.670^* \pm 0.014$ | 53 | $0.675 \pm 0.005$ | 100 |
| breast cancer | $0.823 \pm 0.009$ | 46 | $0.830^* \pm 0.024$ | 46 | $0.826 \pm 0.037$ | 24 | $0.830^* \pm 0.040$ | 100 | $0.823 \pm 0.015$ | 46 | $0.815 \pm 0.051$ | 46 | $0.838 \pm 0.041$ | 100 |
| diabetes | $0.762^* \pm 0.035$ | 24 | $0.762^* \pm 0.026$ | 99 | $0.751 \pm 0.036$ | 60 | $0.743 \pm 0.022$ | 100 | $0.753 \pm 0.034$ | 99 | $0.753 \pm 0.017$ | 99 | $0.756 \pm 0.025$ | 100 |
| german credit | $0.876^* \pm 0.037$ | 50 | $0.865 \pm 0.023$ | 90 | $0.857 \pm 0.020$ | 65 | $0.864 \pm 0.018$ | 100 | $0.850 \pm 0.024$ | 90 | $0.802 \pm 0.020$ | 90 | $0.875 \pm 0.029$ | 100 |
| heart | $0.900^* \pm 0.028$ | 10 | $0.881 \pm 0.019$ | 99 | $0.874 \pm 0.022$ | 47 | $0.889 \pm 0.026$ | 100 | $0.874 \pm 0.025$ | 99 | $0.852 \pm 0.063$ | 99 | $0.881 \pm 0.022$ | 100 |
| image | $0.947^* \pm 0.011$ | 28 | $0.929 \pm 0.008$ | 62 | $0.944 \pm 0.010$ | 31 | $0.946 \pm 0.009$ | 100 | $0.922 \pm 0.017$ | 62 | $0.911 \pm 0.010$ | 62 | $0.941 \pm 0.010$ | 100 |
| ringnorm | $0.924^* \pm 0.003$ | 46 | $0.910 \pm 0.004$ | 66 | $0.924^* \pm 0.005$ | 48 | $0.924^* \pm 0.004$ | 100 | $0.894 \pm 0.004$ | 66 | $0.856 \pm 0.007$ | 66 | $0.906 \pm 0.006$ | 100 |
| solar flare | $0.676 \pm 0.091$ | 7 | $0.683 \pm 0.101$ | 31 | $0.697^* \pm 0.086$ | 6 | $0.683 \pm 0.086$ | 100 | $0.676 \pm 0.099$ | 31 | $0.655 \pm 0.076$ | 31 | $0.697 \pm 0.077$ | 100 |
| splice | $0.979^* \pm 0.006$ | 35 | $0.976 \pm 0.007$ | 100 | $0.977 \pm 0.008$ | 90 | $0.978 \pm 0.006$ | 100 | $0.966 \pm 0.007$ | 100 | $0.940 \pm 0.008$ | 100 | $0.979 \pm 0.007$ | 100 |
| thyroid | $0.944 \pm 0.032$ | 2 | $0.944 \pm 0.032$ | 18 | $0.944 \pm 0.032$ | 2 | $0.940 \pm 0.024$ | 100 | $0.944 \pm 0.028$ | 18 | $0.949^* \pm 0.034$ | 18 | $0.944 \pm 0.032$ | 100 |
| titanic | $0.800 \pm 0.017$ | 7 | $0.810^* \pm 0.025$ | 90 | $0.809 \pm 0.017$ | 41 | $0.801 \pm 0.017$ | 100 | $0.808 \pm 0.029$ | 90 | $0.808 \pm 0.023$ | 90 | $0.807 \pm 0.020$ | 100 |
| twonorm | $0.956^* \pm 0.003$ | 71 | $0.946 \pm 0.005$ | 98 | $0.953 \pm 0.005$ | 66 | $0.955 \pm 0.005$ | 100 | $0.937 \pm 0.004$ | 98 | $0.906 \pm 0.008$ | 98 | $0.950 \pm 0.002$ | 100 |
| waveform | $0.894 \pm 0.014$ | 49 | $0.888 \pm 0.009$ | 70 | $0.892 \pm 0.014$ | 46 | $0.895^* \pm 0.015$ | 100 | $0.885 \pm 0.006$ | 70 | $0.872 \pm 0.007$ | 70 | $0.893 \pm 0.009$ | 100 |
| adult | $0.812 \pm 0.002$ | 30 | $0.812 \pm 0.001$ | 100 | $0.814 \pm 0.003$ | 57 | $0.814 \pm 0.003$ | 100 | $0.819^* \pm 0.001$ | 100 | $0.810 \pm 0.005$ | 100 | $0.820 \pm 0.001$ | 100 |
| compas propublica | $0.664 \pm 0.016$ | 27 | $0.663 \pm 0.017$ | 88 | $0.668^* \pm 0.015$ | 22 | $0.668^* \pm 0.016$ | 100 | $0.663 \pm 0.018$ | 88 | $0.666 \pm 0.012$ | 88 | $0.668 \pm 0.014$ | 100 |
| employment CA2018 | $0.738 \pm 0.002$ | 9 | $0.746^* \pm 0.003$ | 62 | $0.738 \pm 0.002$ | 9 | $0.738 \pm 0.002$ | 100 | $0.746^* \pm 0.003$ | 62 | $0.736 \pm 0.012$ | 62 | $0.744 \pm 0.002$ | 100 |
| employment TX2018 | $0.748 \pm 0.006$ | 39 | $0.752^* \pm 0.002$ | 76 | $0.742 \pm 0.004$ | 8 | $0.742 \pm 0.004$ | 100 | $0.752^* \pm 0.002$ | 73 | $0.746 \pm 0.005$ | 76 | $0.753 \pm 0.002$ | 100 |
| public coverage CA2018 | $0.708^* \pm 0.004$ | 81 | $0.700 \pm 0.005$ | 99 | $0.706 \pm 0.004$ | 73 | $0.707 \pm 0.004$ | 100 | $0.708^* \pm 0.003$ | 99 | $0.703 \pm 0.005$ | 99 | $0.710 \pm 0.003$ | 100 |
| public coverage TX2018 | $0.851 \pm 0.003$ | 8 | $0.851 \pm 0.002$ | 97 | $0.851 \pm 0.001$ | 4 | $0.851 \pm 0.002$ | 100 | $0.852^* \pm 0.003$ | 97 | $0.849 \pm 0.002$ | 97 | $0.851 \pm 0.002$ | 100 |
| mushroom secondary | $0.952 \pm 0.017$ | 49 | $0.945 \pm 0.000$ | 73 | $0.953^* \pm 0.017$ | 48 | $0.953^* \pm 0.017$ | 100 | $0.926 \pm 0.012$ | 71 | $0.889 \pm 0.023$ | 71 | $0.928 \pm 0.014$ | 100 |
| Mean/Median | $\mathbf{0.831^*}$ / 0.837 | 29.9 / 27.5 | 0.828 / $\mathbf{0.841}^*$ | 75.8 / 82.0 | 0.830 / 0.839 | 37.9 / 43.5 | 0.830 / $\mathbf{0.841}^*$ | 100.0 / 100.0 | 0.823 / 0.837 | 75.6 / 80.5 | 0.809 / 0.812 | 75.8 / 82.0 | 0.831 / 0.845 | 100 / 100 |

## D.6    Effect of Confidence-Rated Voting

In this appendix, we provide additional results of our experiments using confidence-rated voting for CART tree-based learners (Section 6.7). Figures 14 and 15 show the anytime behavior of the considered boosting approaches using standard CART trees and confidence-rated CART trees on depth 1, 3, 5, and 10 (top to bottom) decision trees, for the *twonorm* and *waveform* datasets, respectively. For visualization purposes, we split the methods over three columns, ensuring the axes have the same range. As mentioned in Demiriz et al. (2002), tree stumps have too little confidence information and therefore confidence-rated boosting has little effect. On depths 3 and 5, we sometimes see slightly improved anytime behavior of confidence-rated trees in the early iterations, especially for NM-Boost, QRLP-Boost, LP-Boost, and ERLP-Boost. However, the performance difference is never significant or structurally observed for all datasets, and the final performance

Table 11: Testing accuracy and number of non-zero weights for different boosting methods for a **reweighting experiment** using 100 **depth 5** trees generated by Adaboost. A star* marks the best among *totally corrective* methods. The last row shows the mean and median for both statistics. †Adaboost hyperparameters were not tuned for the reweight experiment.

| Dataset | NM-Boost Acc. | Cols. | QRLP-Boost Acc. | Cols. | LP-Boost Acc. | Cols. | CG-Boost Acc. | Cols. | ERLP-Boost Acc. | Cols. | MD-Boost Acc. | Cols. | Adaboost† Acc. | Cols. |
|---|---|---|---|---|---|---|---|---|---|---|---|---|---|---|
| banana | $0.670 \pm 0.014$ | 12 | $0.670 \pm 0.014$ | 73 | $0.670 \pm 0.014$ | 1 | $0.670 \pm 0.014$ | 100 | $0.670 \pm 0.014$ | 73 | $0.676^* \pm 0.000$ | 86 | $0.670 \pm 0.014$ | 100 |
| breast cancer | $0.857 \pm 0.026$ | 7 | $0.853 \pm 0.028$ | 46 | $0.868 \pm 0.017$ | 17 | $0.872^* \pm 0.022$ | 100 | $0.849 \pm 0.032$ | 46 | $0.853 \pm 0.054$ | 46 | $0.849 \pm 0.034$ | 100 |
| diabetes | $0.729 \pm 0.014$ | 23 | $0.751 \pm 0.015$ | 100 | $0.745 \pm 0.019$ | 97 | $0.745 \pm 0.020$ | 100 | $0.753 \pm 0.017$ | 100 | $0.756^* \pm 0.015$ | 100 | $0.753 \pm 0.022$ | 100 |
| german credit | $0.880 \pm 0.021$ | 22 | $0.895^* \pm 0.023$ | 100 | $0.884 \pm 0.026$ | 90 | $0.886 \pm 0.010$ | 100 | $0.888 \pm 0.024$ | 100 | $0.857 \pm 0.021$ | 100 | $0.893 \pm 0.022$ | 100 |
| heart | $0.863 \pm 0.015$ | 3 | $0.874 \pm 0.032$ | 100 | $0.896^* \pm 0.009$ | 42 | $0.893 \pm 0.007$ | 100 | $0.878 \pm 0.015$ | 100 | $0.870 \pm 0.031$ | 100 | $0.900 \pm 0.015$ | 100 |
| image | $0.956 \pm 0.007$ | 21 | $0.954 \pm 0.008$ | 50 | $0.957^* \pm 0.004$ | 24 | $0.957 \pm 0.002$ | 100 | $0.948 \pm 0.009$ | 50 | $0.935 \pm 0.010$ | 50 | $0.955 \pm 0.005$ | 100 |
| ringnorm | $0.928^* \pm 0.003$ | 52 | $0.916 \pm 0.004$ | 91 | $0.927 \pm 0.004$ | 52 | $0.928 \pm 0.004$ | 100 | $0.905 \pm 0.004$ | 91 | $0.863 \pm 0.007$ | 91 | $0.920 \pm 0.005$ | 100 |
| solar flare | $0.655 \pm 0.079$ | 6 | $0.641 \pm 0.052$ | 29 | $0.676^* \pm 0.097$ | 5 | $0.641 \pm 0.056$ | 100 | $0.634 \pm 0.064$ | 29 | $0.641 \pm 0.056$ | 29 | $0.648 \pm 0.046$ | 100 |
| splice | $0.978 \pm 0.005$ | 26 | $0.983 \pm 0.003$ | 100 | $0.982 \pm 0.005$ | 91 | $0.984^* \pm 0.003$ | 100 | $0.970 \pm 0.008$ | 100 | $0.963 \pm 0.009$ | 100 | $0.983 \pm 0.005$ | 100 |
| thyroid | $0.940 \pm 0.024$ | 3 | $0.940 \pm 0.024$ | 18 | $0.944^* \pm 0.019$ | 1 | $0.940 \pm 0.024$ | 100 | $0.935 \pm 0.027$ | 18 | $0.935 \pm 0.027$ | 18 | $0.940 \pm 0.024$ | 100 |
| titanic | $0.793 \pm 0.024$ | 7 | $0.802 \pm 0.022$ | 100 | $0.797 \pm 0.029$ | 43 | $0.788 \pm 0.024$ | 100 | $0.811^* \pm 0.024$ | 100 | $0.804 \pm 0.028$ | 100 | $0.792 \pm 0.027$ | 100 |
| twonorm | $0.960 \pm 0.004$ | 47 | $0.959 \pm 0.002$ | 100 | $0.963 \pm 0.003$ | 79 | $0.968^* \pm 0.003$ | 100 | $0.951 \pm 0.003$ | 100 | $0.921 \pm 0.008$ | 100 | $0.965 \pm 0.004$ | 100 |
| waveform | $0.926^* \pm 0.006$ | 61 | $0.925 \pm 0.007$ | 98 | $0.926^* \pm 0.010$ | 85 | $0.926^* \pm 0.009$ | 100 | $0.916 \pm 0.003$ | 98 | $0.882 \pm 0.006$ | 98 | $0.928 \pm 0.008$ | 100 |
| adult | $0.811 \pm 0.002$ | 59 | $0.807 \pm 0.004$ | 100 | $0.814^* \pm 0.002$ | 72 | $0.814^* \pm 0.004$ | 100 | $0.812 \pm 0.000$ | 100 | $0.814^* \pm 0.001$ | 100 | $0.668 \pm 0.002$ | 100 |
| compas propublica | $0.665 \pm 0.014$ | 59 | $0.663 \pm 0.010$ | 100 | $0.669 \pm 0.017$ | 56 | $0.667 \pm 0.017$ | 100 | $0.664 \pm 0.012$ | 100 | $0.670^* \pm 0.017$ | 100 | $0.668 \pm 0.016$ | 100 |
| employment CA2018 | $0.747 \pm 0.002$ | 58 | $0.748^* \pm 0.003$ | 92 | $0.739 \pm 0.002$ | 100 | $0.739 \pm 0.003$ | 100 | $0.748^* \pm 0.002$ | 92 | $0.739 \pm 0.002$ | 92 | $0.748 \pm 0.002$ | 100 |
| employment TX2018 | $0.757 \pm 0.004$ | 59 | $0.758^* \pm 0.003$ | 94 | $0.743 \pm 0.003$ | 9 | $0.743 \pm 0.003$ | 100 | $0.757 \pm 0.005$ | 95 | $0.750 \pm 0.005$ | 94 | $0.758 \pm 0.003$ | 100 |
| public coverage CA2018 | $0.706 \pm 0.002$ | 61 | $0.691 \pm 0.004$ | 100 | $0.713^* \pm 0.004$ | 97 | $0.713^* \pm 0.006$ | 100 | $0.709 \pm 0.004$ | 100 | $0.704 \pm 0.006$ | 100 | $0.714 \pm 0.002$ | 100 |
| public coverage TX2018 | $0.850 \pm 0.001$ | 4 | $0.852 \pm 0.001$ | 99 | $0.849 \pm 0.002$ | 6 | $0.849 \pm 0.001$ | 100 | $0.853^* \pm 0.001$ | 99 | $0.850 \pm 0.001$ | 99 | $0.850 \pm 0.001$ | 100 |
| mushroom secondary | $0.999^* \pm 0.000$ | 45 | $0.996 \pm 0.001$ | 96 | $0.999^* \pm 0.000$ | 66 | $0.999^* \pm 0.000$ | 100 | $0.992 \pm 0.002$ | 96 | $0.962 \pm 0.008$ | 96 | $0.998 \pm 0.000$ | 100 |
| Mean/Median | 0.833 / 0.853 | 29.6 / 22.5 | 0.834 / 0.853 | 84.3 / 98.5 | **0.838*** / 0.859 | 47.2 / 47.5 | 0.836 / **0.861*** | 100.0 / 100.0 | 0.832 / 0.851 | 84.3 / 98.5 | 0.822 / 0.851 | 85.0 / 98.5 | 0.838 / 0.849 | 100 / 100 |

Table 12: Testing accuracy and number of non-zero weights for different boosting methods for a **reweighting experiment** using 100 **depth 10** trees generated by Adaboost. A star* marks the best among *totally corrective* methods. The last row shows the mean and median for both statistics. †Adaboost hyperparameters were not tuned for the reweight experiment.

| Dataset | NM-Boost Acc. | Cols. | QRLP-Boost Acc. | Cols. | LP-Boost Acc. | Cols. | CG-Boost Acc. | Cols. | ERLP-Boost Acc. | Cols. | MD-Boost Acc. | Cols. | Adaboost† Acc. | Cols. |
|---|---|---|---|---|---|---|---|---|---|---|---|---|---|---|
| banana | $0.670 \pm 0.014$ | 10 | $0.670 \pm 0.014$ | 62 | $0.670 \pm 0.014$ | 1 | $0.670 \pm 0.014$ | 100 | $0.670 \pm 0.014$ | 62 | $0.684^* \pm 0.000$ | 78 | $0.670 \pm 0.014$ | 100 |
| breast cancer | $0.834 \pm 0.032$ | 2 | $0.842^* \pm 0.039$ | 39 | $0.830 \pm 0.062$ | 1 | $0.838 \pm 0.046$ | 100 | $0.842^* \pm 0.039$ | 39 | $0.842^* \pm 0.042$ | 39 | $0.849 \pm 0.040$ | 100 |
| diabetes | $0.671 \pm 0.029$ | 2 | $0.719 \pm 0.029$ | 85 | $0.740 \pm 0.030$ | 59 | $0.762^* \pm 0.007$ | 85 | $0.695 \pm 0.026$ | 85 | $0.732 \pm 0.030$ | 85 | $0.752 \pm 0.021$ | 100 |
| german credit | $0.848 \pm 0.027$ | 3 | $0.880 \pm 0.024$ | 100 | $0.886^* \pm 0.023$ | 74 | $0.886^* \pm 0.020$ | 100 | $0.871 \pm 0.026$ | 100 | $0.879 \pm 0.026$ | 100 | $0.890 \pm 0.020$ | 100 |
| heart | $0.830^* \pm 0.022$ | 1 | $0.830^* \pm 0.022$ | 1 | $0.830^* \pm 0.022$ | 1 | $0.830^* \pm 0.022$ | 1 | $0.830^* \pm 0.022$ | 1 | $0.830^* \pm 0.022$ | 1 | $0.830 \pm 0.022$ | 100 |
| image | $0.948 \pm 0.005$ | 6 | $0.952 \pm 0.006$ | 66 | $0.953 \pm 0.006$ | 14 | $0.955^* \pm 0.004$ | 100 | $0.951 \pm 0.007$ | 66 | $0.951 \pm 0.007$ | 66 | $0.955 \pm 0.006$ | 100 |
| ringnorm | $0.908 \pm 0.009$ | 5 | $0.929^* \pm 0.008$ | 100 | $0.927 \pm 0.006$ | 100 | $0.928 \pm 0.006$ | 100 | $0.928 \pm 0.008$ | 100 | $0.924 \pm 0.006$ | 100 | $0.931 \pm 0.008$ | 100 |
| solar flare | $0.662 \pm 0.051$ | 4 | $0.662 \pm 0.026$ | 65 | $0.655 \pm 0.022$ | 1 | $0.662 \pm 0.059$ | 100 | $0.641 \pm 0.035$ | 65 | $0.676^* \pm 0.083$ | 65 | $0.648 \pm 0.055$ | 100 |
| splice | $0.965 \pm 0.008$ | 2 | $0.971 \pm 0.010$ | 78 | $0.974^* \pm 0.012$ | 57 | $0.972 \pm 0.013$ | 80 | $0.968 \pm 0.008$ | 78 | $0.972 \pm 0.011$ | 78 | $0.979 \pm 0.007$ | 100 |
| thyroid | $0.944^* \pm 0.032$ | 3 | $0.940 \pm 0.024$ | 8 | $0.940 \pm 0.024$ | 1 | $0.944^* \pm 0.019$ | 100 | $0.940 \pm 0.024$ | 8 | $0.935 \pm 0.027$ | 8 | $0.940 \pm 0.024$ | 100 |
| titanic | $0.787 \pm 0.010$ | 4 | $0.793 \pm 0.015$ | 89 | $0.791 \pm 0.017$ | 14 | $0.803^* \pm 0.018$ | 100 | $0.789 \pm 0.015$ | 89 | $0.791 \pm 0.014$ | 89 | $0.792 \pm 0.010$ | 100 |
| twonorm | $0.944 \pm 0.008$ | 18 | $0.963 \pm 0.002$ | 100 | $0.966 \pm 0.003$ | 100 | $0.967^* \pm 0.003$ | 100 | $0.957 \pm 0.002$ | 100 | $0.955 \pm 0.004$ | 100 | $0.966 \pm 0.005$ | 100 |
| waveform | $0.914 \pm 0.011$ | 10 | $0.926 \pm 0.010$ | 100 | $0.927 \pm 0.008$ | 98 | $0.929^* \pm 0.007$ | 100 | $0.923 \pm 0.007$ | 100 | $0.923 \pm 0.009$ | 100 | $0.931 \pm 0.010$ | 100 |
| adult | $0.816 \pm 0.002$ | 2 | $0.812 \pm 0.002$ | 100 | $0.816 \pm 0.002$ | 40 | $0.817^* \pm 0.002$ | 100 | $0.815 \pm 0.002$ | 100 | $0.816 \pm 0.002$ | 100 | $0.816 \pm 0.003$ | 100 |
| compas propublica | $0.664 \pm 0.012$ | 57 | $0.665^* \pm 0.013$ | 100 | $0.664 \pm 0.015$ | 1 | $0.664 \pm 0.015$ | 100 | $0.665^* \pm 0.013$ | 100 | $0.665^* \pm 0.010$ | 100 | $0.665 \pm 0.013$ | 100 |
| employment CA2018 | $0.746 \pm 0.002$ | 89 | $0.748 \pm 0.004$ | 100 | $0.748 \pm 0.002$ | 84 | $0.748 \pm 0.001$ | 100 | $0.750^* \pm 0.002$ | 99 | $0.747 \pm 0.001$ | 99 | $0.752 \pm 0.001$ | 100 |
| employment TX2018 | $0.753 \pm 0.011$ | 82 | $0.753 \pm 0.004$ | 98 | $0.758^* \pm 0.004$ | 77 | $0.758^* \pm 0.003$ | 100 | $0.756 \pm 0.003$ | 98 | $0.753 \pm 0.003$ | 98 | $0.760 \pm 0.002$ | 100 |
| public coverage CA2018 | $0.704 \pm 0.005$ | 35 | $0.709 \pm 0.001$ | 100 | $0.707 \pm 0.002$ | 100 | $0.715^* \pm 0.004$ | 100 | $0.714 \pm 0.003$ | 100 | $0.703 \pm 0.004$ | 100 | $0.714 \pm 0.003$ | 100 |
| public coverage TX2018 | $0.847 \pm 0.002$ | 2 | $0.853 \pm 0.004$ | 99 | $0.850 \pm 0.004$ | 68 | $0.848 \pm 0.003$ | 100 | $0.855^* \pm 0.004$ | 99 | $0.847 \pm 0.002$ | 99 | $0.853 \pm 0.003$ | 100 |
| mushroom secondary | $0.999^* \pm 0.000$ | 15 | $0.999^* \pm 0.000$ | 89 | $0.999^* \pm 0.000$ | 34 | $0.999^* \pm 0.000$ | 100 | $0.999^* \pm 0.000$ | 89 | $0.995 \pm 0.002$ | 89 | $0.999 \pm 0.000$ | 100 |
| Mean/Median | 0.823 / 0.832 | 17.6 / 4.5 | 0.831 / **0.836*** | 79.0 / 93.5 | 0.831 / 0.830 | 46.2 / 48.5 | **0.835*** / 0.834 | 93.3 / 100.0 | 0.828 / **0.836*** | 78.9 / 93.5 | 0.831 / **0.836*** | 79.7 / 93.5 | 0.835 / 0.839 | 100 / 100 |

is often identical. Figure 16 displays the testing accuracy and sparsity of the different boosting approaches, averaged across all datasets, for both standard CART trees and confidence-rated CART trees, for the different tree depths. Finally, Tables 13, 14, 15 and 16 show the testing accuracy and sparsity of each tested approach for each individual dataset, for confidence-rated trees of depth 1, 3, 5 and 10, respectively.

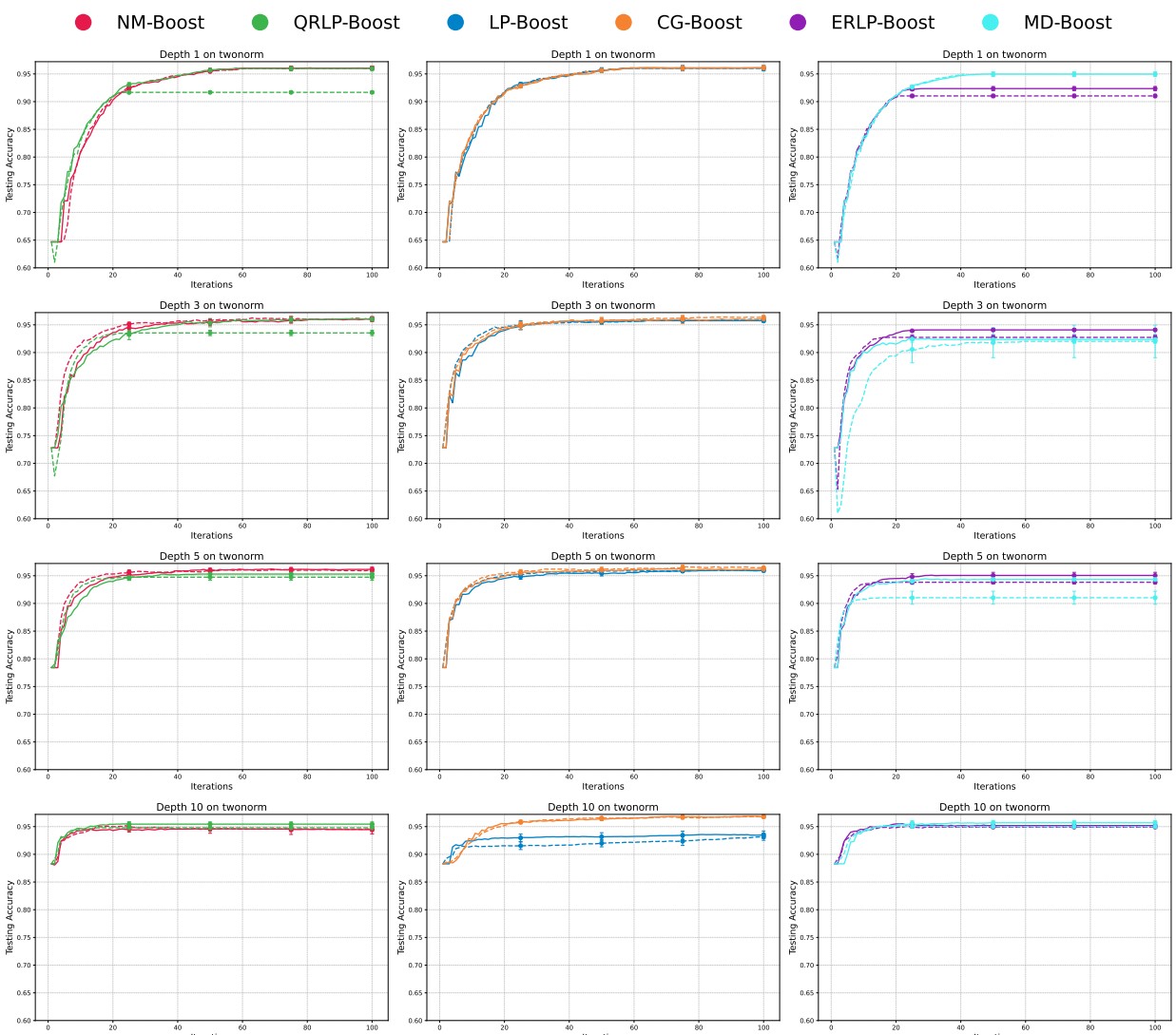

Figure 14: Anytime behavior for the **twonorm** dataset and 1, 3, 5, and 10 depth CART decision trees compared to confidence-rated trees (dashed line), error bars indicate standard deviation over 5 seeds. The methods are split over three columns for visualization purposes.

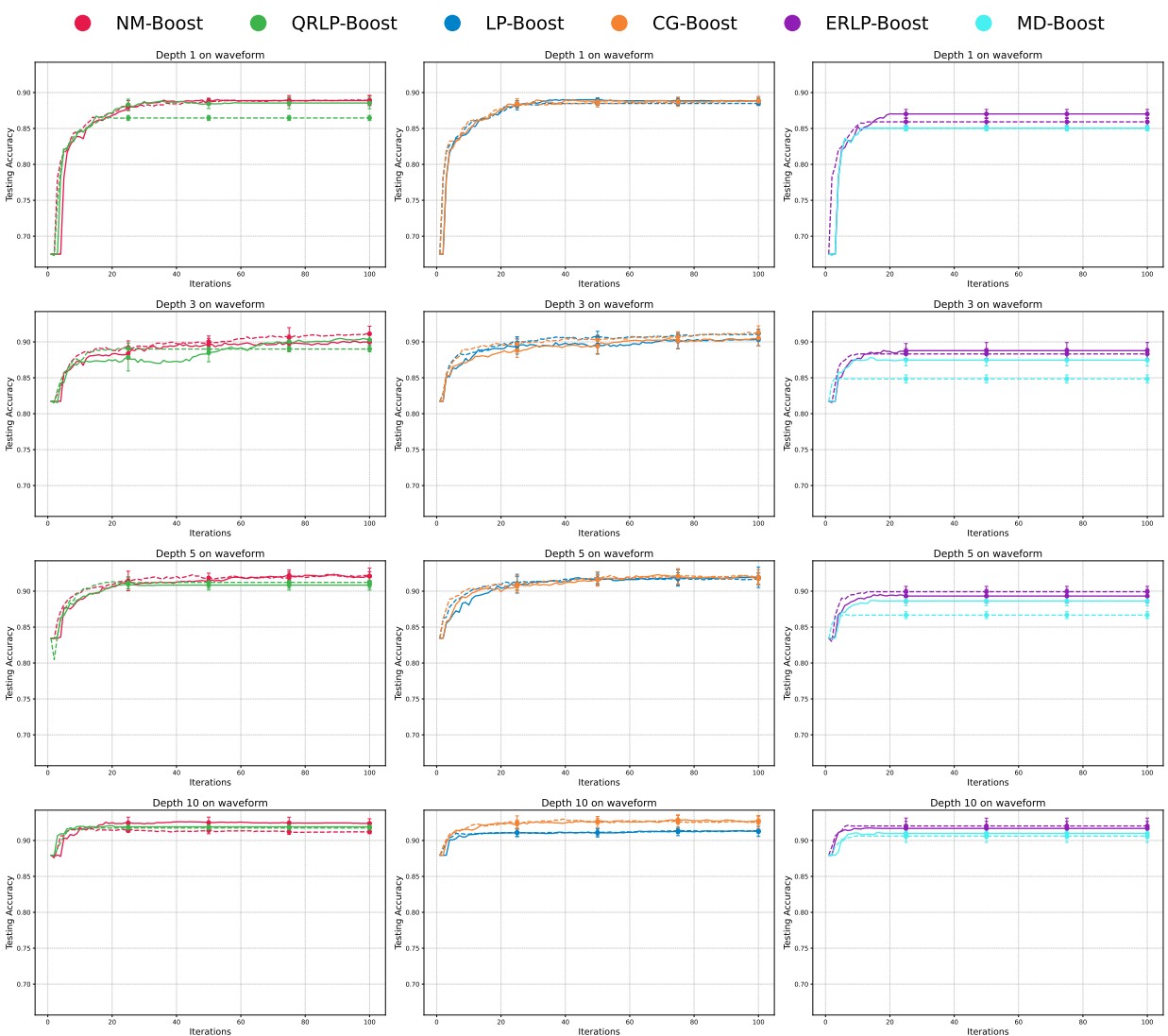

Figure 15: Anytime behavior for the **waveform** dataset and 1, 3, 5, and 10 depth CART decision trees compared to confidence-rated trees (dashed line), error bars indicate standard deviation over 5 seeds. The methods are split over three columns for visualization purposes.

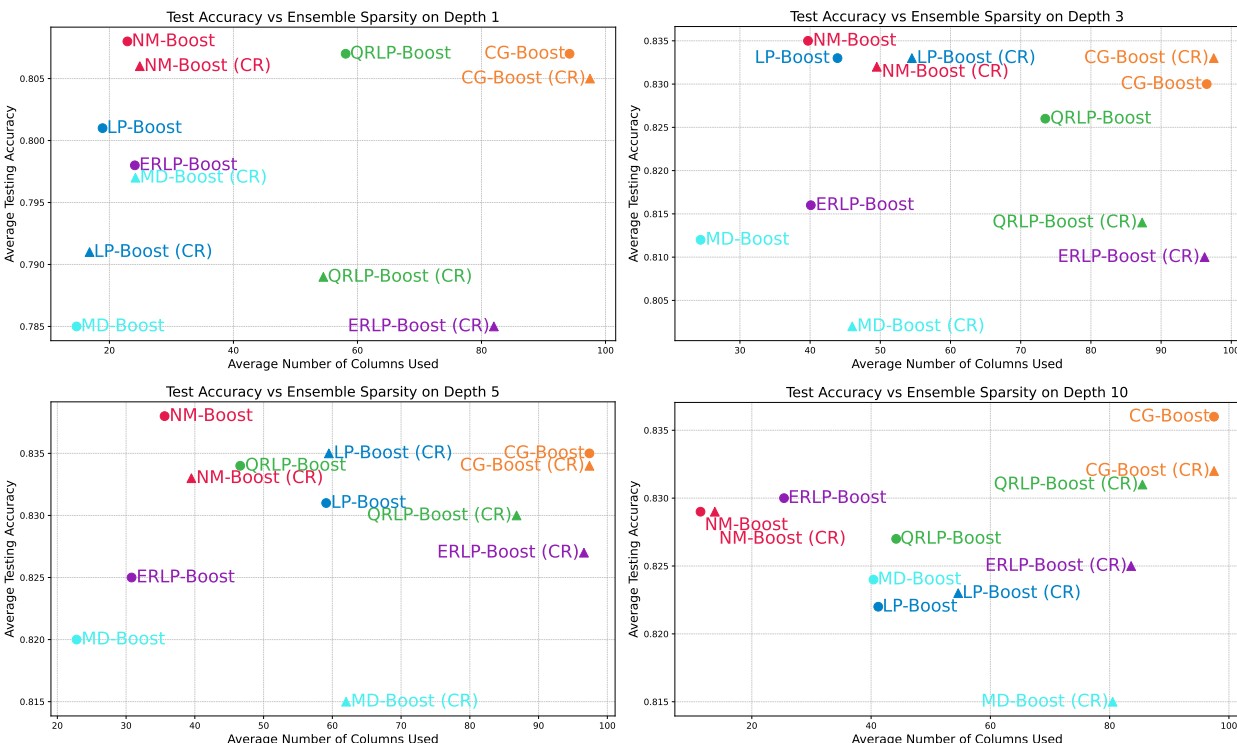

Figure 16: Average testing accuracy compared to average ensemble sparsity over **all datasets** for 1, 3, 5, and 10 depth CART trees with either hard voting or confidence-rated (CR) soft voting.

Table 13: Testing accuracy and number of non-zero weights for different boosting methods using **confidence-rated** CART trees of **depth 1**, averaged over five seeds. A star* marks the best among *totally corrective* methods. The last row shows the mean and median for both statistics.

| Dataset | NM-Boost | | QRLP-Boost | | LP-Boost | | CG-Boost | | ERLP-Boost | | MD-Boost | |
|---|---|---|---|---|---|---|---|---|---|---|---|---|
| | Acc. | Cols. | Acc. | Cols. | Acc. | Cols. | Acc. | Cols. | Acc. | Cols. | Acc. | Cols. |
| banana | $0.625^* \pm 0.016$ | 4 | $0.622 \pm 0.014$ | 4 | $0.625^* \pm 0.016$ | 2 | $0.625^* \pm 0.016$ | 100 | $0.620 \pm 0.016$ | 100 | $0.625^* \pm 0.016$ | 4 |
| breast cancer | $0.785 \pm 0.080$ | 3 | $0.804 \pm 0.058$ | 35 | $0.770 \pm 0.058$ | 0 | $0.792 \pm 0.040$ | 100 | $0.792 \pm 0.058$ | 94 | $0.815^* \pm 0.051$ | 22 |
| diabetes | $0.768^* \pm 0.019$ | 4 | $0.758 \pm 0.021$ | 100 | $0.732 \pm 0.039$ | 4 | $0.751 \pm 0.025$ | 100 | $0.764 \pm 0.020$ | 100 | $0.757 \pm 0.020$ | 17 |
| german credit | $0.807^* \pm 0.026$ | 26 | $0.786 \pm 0.024$ | 100 | $0.732 \pm 0.007$ | 2 | $0.790 \pm 0.027$ | 100 | $0.768 \pm 0.024$ | 100 | $0.805 \pm 0.021$ | 43 |
| heart | $0.822 \pm 0.022$ | 17 | $0.789 \pm 0.028$ | 100 | $0.774 \pm 0.064$ | 9 | $0.833^* \pm 0.035$ | 100 | $0.767 \pm 0.045$ | 73 | $0.826 \pm 0.025$ | 21 |
| image | $0.856^* \pm 0.010$ | 22 | $0.807 \pm 0.010$ | 15 | $0.833 \pm 0.017$ | 13 | $0.843 \pm 0.003$ | 100 | $0.784 \pm 0.024$ | 12 | $0.768 \pm 0.025$ | 12 |
| ringnorm | $0.929 \pm 0.002$ | 60 | $0.871 \pm 0.018$ | 64 | $0.929 \pm 0.002$ | 58 | $0.931^* \pm 0.002$ | 100 | $0.863 \pm 0.006$ | 100 | $0.906 \pm 0.002$ | 30 |
| solar flare | $0.703 \pm 0.071$ | 7 | $0.738^* \pm 0.064$ | 31 | $0.662 \pm 0.091$ | 3 | $0.738^* \pm 0.068$ | 100 | $0.738^* \pm 0.047$ | 85 | $0.710 \pm 0.071$ | 14 |
| splice | $0.943^* \pm 0.007$ | 53 | $0.927 \pm 0.013$ | 83 | $0.942 \pm 0.007$ | 29 | $0.943^* \pm 0.010$ | 100 | $0.915 \pm 0.011$ | 81 | $0.924 \pm 0.009$ | 12 |
| thyroid | $0.940 \pm 0.035$ | 5 | $0.916 \pm 0.032$ | 8 | $0.912 \pm 0.017$ | 4 | $0.944^* \pm 0.032$ | 100 | $0.907 \pm 0.025$ | 64 | $0.944^* \pm 0.032$ | 38 |
| titanic | $0.800^* \pm 0.019$ | 12 | $0.799 \pm 0.021$ | 45 | $0.791 \pm 0.022$ | 4 | $0.793 \pm 0.024$ | 100 | $0.797 \pm 0.021$ | 84 | $0.798 \pm 0.028$ | 22 |
| twonorm | $0.961^* \pm 0.003$ | 60 | $0.917 \pm 0.002$ | 100 | $0.960 \pm 0.005$ | 58 | $0.961^* \pm 0.003$ | 100 | $0.910 \pm 0.003$ | 100 | $0.951 \pm 0.004$ | 46 |
| waveform | $0.889^* \pm 0.006$ | 53 | $0.865 \pm 0.004$ | 100 | $0.885 \pm 0.003$ | 35 | $0.888 \pm 0.004$ | 100 | $0.859 \pm 0.006$ | 100 | $0.850 \pm 0.003$ | 12 |
| adult | $0.814^* \pm 0.003$ | 15 | $0.786 \pm 0.014$ | 5 | $0.812 \pm 0.003$ | 12 | $0.813 \pm 0.003$ | 96 | $0.810 \pm 0.005$ | 96 | $0.812 \pm 0.004$ | 14 |
| compas propublica | $0.659 \pm 0.015$ | 6 | $0.672^* \pm 0.018$ | 100 | $0.656 \pm 0.015$ | 4 | $0.656 \pm 0.015$ | 100 | $0.668 \pm 0.018$ | 65 | $0.672^* \pm 0.017$ | 13 |
| employment CA2018 | $0.740^* \pm 0.003$ | 17 | $0.733 \pm 0.008$ | 18 | $0.740^* \pm 0.003$ | 18 | $0.740^* \pm 0.003$ | 76 | $0.735 \pm 0.003$ | 84 | $0.734 \pm 0.003$ | 11 |
| employment TX2018 | $0.744^* \pm 0.003$ | 18 | $0.742 \pm 0.003$ | 24 | $0.744^* \pm 0.003$ | 19 | $0.742 \pm 0.003$ | 91 | $0.736 \pm 0.003$ | 83 | $0.731 \pm 0.005$ | 10 |
| public coverage CA2018 | $0.696 \pm 0.004$ | 27 | $0.698 \pm 0.005$ | 100 | $0.680 \pm 0.005$ | 3 | $0.680 \pm 0.005$ | 100 | $0.688 \pm 0.006$ | 100 | $0.704^* \pm 0.003$ | 55 |
| public coverage TX2018 | $0.847 \pm 0.004$ | 43 | $0.846 \pm 0.002$ | 24 | $0.829 \pm 0.002$ | 5 | $0.830 \pm 0.002$ | 100 | $0.845 \pm 0.003$ | 96 | $0.848^* \pm 0.002$ | 67 |
| mushroom secondary | $0.787 \pm 0.006$ | 46 | $0.697 \pm 0.073$ | 33 | $0.817^* \pm 0.005$ | 54 | $0.810 \pm 0.004$ | 86 | $0.736 \pm 0.003$ | 23 | $0.753 \pm 0.008$ | 22 |
| Mean/Median | $0.806^*/0.804^*$ | 24.9/17.5 | 0.789/0.788 | 54.5/40 | 0.791/0.782 | 16.8/7 | 0.805/0.802 | 97.5/100 | 0.785/0.776 | 82.0/89.5 | 0.797/0.802 | 24.2/19 |

Table 14: Testing accuracy and number of non-zero weights for different boosting methods using **confidence-rated** CART trees of **depth 3**, averaged over five seeds. A star* marks the best among *totally corrective* methods. The last row shows the mean and median for both statistics.

| | NM-Boost | | QRLP-Boost | | LP-Boost | | CG-Boost | | ERLP-Boost | | MD-Boost | |
|---|---|---|---|---|---|---|---|---|---|---|---|---|
| Dataset | Acc. | Cols. | Acc. | Cols. | Acc. | Cols. | Acc. | Cols. | Acc. | Cols. | Acc. | Cols. |
| banana | $0.670 \pm 0.014$ | 14 | $0.672 \pm 0.015$ | 81 | $0.675^* \pm 0.018$ | 11 | $0.670 \pm 0.014$ | 100 | $0.673 \pm 0.017$ | 100 | $0.670 \pm 0.014$ | 20 |
| breast cancer | $0.838^* \pm 0.019$ | 17 | $0.838^* \pm 0.026$ | 90 | $0.823 \pm 0.035$ | 15 | $0.834 \pm 0.047$ | 100 | $0.804 \pm 0.064$ | 89 | $0.785 \pm 0.028$ | 21 |
| diabetes | $0.752 \pm 0.026$ | 6 | $0.738 \pm 0.017$ | 100 | $0.758^* \pm 0.019$ | 11 | $0.732 \pm 0.038$ | 100 | $0.756 \pm 0.036$ | 100 | $0.730 \pm 0.035$ | 46 |
| german credit | $0.875^* \pm 0.029$ | 50 | $0.835 \pm 0.025$ | 100 | $0.855 \pm 0.028$ | 72 | $0.852 \pm 0.022$ | 100 | $0.827 \pm 0.036$ | 100 | $0.776 \pm 0.030$ | 33 |
| heart | $0.881^* \pm 0.034$ | 7 | $0.867 \pm 0.036$ | 86 | $0.867 \pm 0.040$ | 31 | $0.878 \pm 0.025$ | 100 | $0.852 \pm 0.042$ | 100 | $0.774 \pm 0.072$ | 26 |
| image | $0.947 \pm 0.005$ | 41 | $0.926 \pm 0.015$ | 100 | $0.953 \pm 0.009$ | 39 | $0.954^* \pm 0.007$ | 100 | $0.880 \pm 0.022$ | 89 | $0.876 \pm 0.025$ | 17 |
| ringnorm | $0.937 \pm 0.005$ | 88 | $0.910 \pm 0.003$ | 100 | $0.933 \pm 0.004$ | 69 | $0.938^* \pm 0.003$ | 100 | $0.896 \pm 0.005$ | 100 | $0.887 \pm 0.012$ | 37 |
| solar flare | $0.614 \pm 0.074$ | 11 | $0.655 \pm 0.079$ | 100 | $0.669^* \pm 0.064$ | 7 | $0.648 \pm 0.059$ | 100 | $0.628 \pm 0.122$ | 100 | $0.669^* \pm 0.099$ | 54 |
| splice | $0.975 \pm 0.006$ | 34 | $0.967 \pm 0.008$ | 100 | $0.975 \pm 0.004$ | 89 | $0.976^* \pm 0.005$ | 100 | $0.952 \pm 0.004$ | 100 | $0.935 \pm 0.005$ | 45 |
| thyroid | $0.940^* \pm 0.024$ | 2 | $0.940^* \pm 0.024$ | 100 | $0.940^* \pm 0.024$ | 4 | $0.940^* \pm 0.035$ | 100 | $0.940^* \pm 0.024$ | 50 | $0.940^* \pm 0.024$ | 75 |
| titanic | $0.804 \pm 0.014$ | 11 | $0.810 \pm 0.032$ | 100 | $0.803 \pm 0.027$ | 21 | $0.821^* \pm 0.022$ | 100 | $0.802 \pm 0.026$ | 100 | $0.806 \pm 0.024$ | 100 |
| twonorm | $0.961 \pm 0.003$ | 92 | $0.935 \pm 0.005$ | 100 | $0.959 \pm 0.005$ | 74 | $0.963^* \pm 0.004$ | 100 | $0.927 \pm 0.005$ | 100 | $0.920 \pm 0.029$ | 65 |
| waveform | $0.911 \pm 0.011$ | 94 | $0.890 \pm 0.004$ | 100 | $0.911 \pm 0.006$ | 82 | $0.913^* \pm 0.009$ | 100 | $0.883 \pm 0.007$ | 100 | $0.848 \pm 0.006$ | 45 |
| adult | $0.817 \pm 0.002$ | 88 | $0.817 \pm 0.003$ | 81 | $0.817 \pm 0.001$ | 82 | $0.817 \pm 0.001$ | 96 | $0.816 \pm 0.002$ | 100 | $0.818^* \pm 0.001$ | 41 |
| compas propublica | $0.666 \pm 0.012$ | 21 | $0.668 \pm 0.015$ | 100 | $0.668 \pm 0.013$ | 49 | $0.669^* \pm 0.014$ | 100 | $0.669^* \pm 0.013$ | 100 | $0.661 \pm 0.006$ | 52 |
| employment CA2018 | $0.746^* \pm 0.002$ | 69 | $0.737 \pm 0.009$ | 25 | $0.743 \pm 0.003$ | 74 | $0.743 \pm 0.002$ | 100 | $0.745 \pm 0.002$ | 100 | $0.737 \pm 0.003$ | 16 |
| employment TX2018 | $0.755 \pm 0.001$ | 85 | $0.756^* \pm 0.002$ | 100 | $0.753 \pm 0.003$ | 89 | $0.754 \pm 0.001$ | 91 | $0.752 \pm 0.002$ | 100 | $0.744 \pm 0.005$ | 20 |
| public coverage CA2018 | $0.708 \pm 0.004$ | 92 | $0.702 \pm 0.014$ | 81 | $0.711 \pm 0.004$ | 98 | $0.711 \pm 0.003$ | 100 | $0.705 \pm 0.002$ | 100 | $0.713^* \pm 0.003$ | 95 |
| public coverage TX2018 | $0.852 \pm 0.003$ | 94 | $0.851 \pm 0.001$ | 81 | $0.852 \pm 0.001$ | 92 | $0.852 \pm 0.002$ | 100 | $0.850 \pm 0.002$ | 100 | $0.853^* \pm 0.002$ | 85 |
| mushroom secondary | $0.998^* \pm 0.001$ | 75 | $0.769 \pm 0.085$ | 22 | $0.998^* \pm 0.000$ | 82 | $0.998^* \pm 0.001$ | 86 | $0.852 \pm 0.009$ | 97 | $0.893 \pm 0.033$ | 28 |
| Mean/Median | 0.832/0.845* | 49.5/45.5 | 0.814/0.826 | 87.3/100 | 0.833*/0.837 | 54.5/70.5 | 0.833*/0.843 | 97.5/100 | 0.810/0.821 | 96.2/100 | 0.802/0.796 | 46.0/43 |

Table 15: Testing accuracy and number of non-zero weights for different boosting methods using **confidence-rated** CART trees of **depth 5**, averaged over five seeds. A star* marks the best among *totally corrective* methods. The last row shows the mean and median for both statistics.

| | NM-Boost | | QRLP-Boost | | LP-Boost | | CG-Boost | | ERLP-Boost | | MD-Boost | |
|---|---|---|---|---|---|---|---|---|---|---|---|---|
| Dataset | Acc. | Cols. | Acc. | Cols. | Acc. | Cols. | Acc. | Cols. | Acc. | Cols. | Acc. | Cols. |
| banana | $0.670^* \pm 0.014$ | 14 | $0.670^* \pm 0.014$ | 3 | $0.670^* \pm 0.014$ | 9 | $0.670^* \pm 0.014$ | 100 | $0.670^* \pm 0.014$ | 81 | $0.670^* \pm 0.014$ | 88 |
| breast cancer | $0.830 \pm 0.043$ | 4 | $0.864^* \pm 0.044$ | 100 | $0.838 \pm 0.057$ | 27 | $0.811 \pm 0.012$ | 100 | $0.853 \pm 0.058$ | 100 | $0.777 \pm 0.035$ | 58 |
| diabetes | $0.752^* \pm 0.010$ | 23 | $0.739 \pm 0.034$ | 100 | $0.730 \pm 0.013$ | 75 | $0.735 \pm 0.013$ | 100 | $0.748 \pm 0.024$ | 100 | $0.748 \pm 0.018$ | 83 |
| german credit | $0.882 \pm 0.028$ | 23 | $0.874 \pm 0.024$ | 100 | $0.887^* \pm 0.019$ | 80 | $0.882 \pm 0.029$ | 100 | $0.861 \pm 0.032$ | 100 | $0.825 \pm 0.018$ | 44 |
| heart | $0.841 \pm 0.025$ | 3 | $0.878^* \pm 0.009$ | 100 | $0.878^* \pm 0.030$ | 38 | $0.878^* \pm 0.022$ | 100 | $0.870 \pm 0.042$ | 100 | $0.867 \pm 0.014$ | 72 |
| image | $0.949 \pm 0.005$ | 19 | $0.954^* \pm 0.010$ | 100 | $0.953 \pm 0.006$ | 43 | $0.954^* \pm 0.007$ | 100 | $0.939 \pm 0.007$ | 100 | $0.921 \pm 0.017$ | 78 |
| ringnorm | $0.939^* \pm 0.007$ | 82 | $0.916 \pm 0.006$ | 100 | $0.937 \pm 0.003$ | 70 | $0.937 \pm 0.005$ | 100 | $0.899 \pm 0.008$ | 100 | $0.855 \pm 0.017$ | 47 |
| solar flare | $0.648 \pm 0.059$ | 6 | $0.634 \pm 0.041$ | 100 | $0.641 \pm 0.071$ | 5 | $0.641 \pm 0.071$ | 100 | $0.648 \pm 0.080$ | 100 | $0.669^* \pm 0.056$ | 15 |
| splice | $0.975 \pm 0.006$ | 20 | $0.975 \pm 0.007$ | 100 | $0.980 \pm 0.003$ | 84 | $0.981^* \pm 0.003$ | 100 | $0.971 \pm 0.006$ | 100 | $0.967 \pm 0.006$ | 100 |
| thyroid | $0.940 \pm 0.024$ | 3 | $0.940 \pm 0.035$ | 47 | $0.949^* \pm 0.023$ | 1 | $0.944 \pm 0.028$ | 100 | $0.944 \pm 0.019$ | 57 | $0.921 \pm 0.054$ | 99 |
| titanic | $0.810^* \pm 0.024$ | 6 | $0.800 \pm 0.012$ | 100 | $0.810^* \pm 0.028$ | 14 | $0.799 \pm 0.025$ | 100 | $0.800 \pm 0.025$ | 100 | $0.796 \pm 0.028$ | 48 |
| twonorm | $0.960 \pm 0.003$ | 55 | $0.947 \pm 0.006$ | 100 | $0.960 \pm 0.002$ | 74 | $0.964^* \pm 0.003$ | 100 | $0.938 \pm 0.003$ | 100 | $0.910 \pm 0.012$ | 31 |
| waveform | $0.921^* \pm 0.011$ | 67 | $0.912 \pm 0.008$ | 100 | $0.918 \pm 0.008$ | 76 | $0.920 \pm 0.010$ | 100 | $0.899 \pm 0.007$ | 100 | $0.867 \pm 0.005$ | 92 |
| adult | $0.816 \pm 0.002$ | 82 | $0.815 \pm 0.007$ | 81 | $0.817 \pm 0.002$ | 90 | $0.818^* \pm 0.002$ | 96 | $0.818^* \pm 0.002$ | 100 | $0.818^* \pm 0.002$ | 62 |
| compas propublica | $0.668 \pm 0.014$ | 28 | $0.666 \pm 0.011$ | 100 | $0.668 \pm 0.014$ | 61 | $0.670^* \pm 0.014$ | 100 | $0.667 \pm 0.015$ | 100 | $0.668 \pm 0.016$ | 68 |
| employment CA2018 | $0.748 \pm 0.002$ | 69 | $0.746 \pm 0.006$ | 62 | $0.748 \pm 0.003$ | 75 | $0.749^* \pm 0.002$ | 75 | $0.749^* \pm 0.003$ | 100 | $0.741 \pm 0.003$ | 31 |
| employment TX2018 | $0.756 \pm 0.002$ | 88 | $0.759^* \pm 0.002$ | 100 | $0.758 \pm 0.003$ | 91 | $0.759^* \pm 0.002$ | 91 | $0.757 \pm 0.002$ | 100 | $0.746 \pm 0.006$ | 21 |
| public coverage CA2018 | $0.712 \pm 0.005$ | 96 | $0.707 \pm 0.012$ | 81 | $0.716^* \pm 0.005$ | 99 | $0.714 \pm 0.003$ | 100 | $0.708 \pm 0.005$ | 100 | $0.715 \pm 0.004$ | 62 |
| public coverage TX2018 | $0.849 \pm 0.002$ | 57 | $0.852 \pm 0.002$ | 100 | $0.852 \pm 0.002$ | 98 | $0.852 \pm 0.001$ | 100 | $0.852 \pm 0.002$ | 100 | $0.855^* \pm 0.002$ | 81 |
| mushroom secondary | $0.998 \pm 0.000$ | 44 | $0.944 \pm 0.041$ | 62 | $0.999^* \pm 0.000$ | 80 | $0.999^* \pm 0.000$ | 86 | $0.940 \pm 0.006$ | 94 | $0.959 \pm 0.030$ | 60 |
| Mean/Median | 0.833/0.835 | 39.5/25.5 | 0.830/ 0.858* | 86.8/100 | 0.835*/0.845 | 59.5/74.5 | 0.834/0.835 | 97.4/100 | 0.827/0.853 | 96.6/100 | 0.815/0.821 | 62.0/62 |

Table 16: Testing accuracy and number of non-zero weights for different boosting methods using **confidence-rated** CART trees of **depth 10**, averaged over five seeds. A star* marks the best among *totally corrective* methods. The last row shows the mean and median for both statistics.

| Dataset | NM-Boost Acc. | Cols. | QRLP-Boost Acc. | Cols. | LP-Boost Acc. | Cols. | CG-Boost Acc. | Cols. | ERLP-Boost Acc. | Cols. | MD-Boost Acc. | Cols. |
|---|---|---|---|---|---|---|---|---|---|---|---|---|
| banana | $0.670^* \pm 0.014$ | 14 | $0.670^* \pm 0.014$ | 10 | $0.670^* \pm 0.014$ | 100 | $0.670^* \pm 0.014$ | 100 | $0.670^* \pm 0.014$ | 100 | $0.670^* \pm 0.014$ | 100 |
| breast cancer | $0.819 \pm 0.019$ | 4 | $0.830 \pm 0.024$ | 74 | $0.819 \pm 0.026$ | 28 | $0.808 \pm 0.025$ | 100 | $0.823 \pm 0.035$ | 81 | $0.845^* \pm 0.037$ | 56 |
| diabetes | $0.704 \pm 0.036$ | 2 | $0.740^* \pm 0.026$ | 100 | $0.697 \pm 0.032$ | 63 | $0.734 \pm 0.015$ | 100 | $0.687 \pm 0.043$ | 99 | $0.716 \pm 0.019$ | 100 |
| german credit | $0.866 \pm 0.030$ | 3 | $0.887^* \pm 0.016$ | 100 | $0.876 \pm 0.032$ | 66 | $0.881 \pm 0.024$ | 100 | $0.872 \pm 0.020$ | 87 | $0.871 \pm 0.022$ | 89 |
| heart | $0.837 \pm 0.022$ | 1 | $0.833 \pm 0.031$ | 1 | $0.833 \pm 0.012$ | 1 | $0.841 \pm 0.025$ | 100 | $0.844^* \pm 0.015$ | 1 | $0.589 \pm 0.055$ | 2 |
| image | $0.955^* \pm 0.008$ | 7 | $0.950 \pm 0.011$ | 100 | $0.952 \pm 0.011$ | 47 | $0.952 \pm 0.011$ | 100 | $0.944 \pm 0.009$ | 83 | $0.943 \pm 0.018$ | 99 |
| ringnorm | $0.926 \pm 0.004$ | 21 | $0.924 \pm 0.006$ | 100 | $0.908 \pm 0.006$ | 98 | $0.935^* \pm 0.003$ | 100 | $0.915 \pm 0.008$ | 100 | $0.919 \pm 0.005$ | 88 |
| solar flare | $0.683^* \pm 0.077$ | 7 | $0.683^* \pm 0.026$ | 99 | $0.655 \pm 0.049$ | 10 | $0.648 \pm 0.059$ | 100 | $0.628 \pm 0.063$ | 99 | $0.676 \pm 0.035$ | 75 |
| splice | $0.963 \pm 0.005$ | 2 | $0.970 \pm 0.005$ | 100 | $0.968 \pm 0.009$ | 76 | $0.979^* \pm 0.005$ | 100 | $0.968 \pm 0.010$ | 100 | $0.977 \pm 0.003$ | 100 |
| thyroid | $0.949 \pm 0.023$ | 3 | $0.953^* \pm 0.029$ | 51 | $0.949 \pm 0.023$ | 1 | $0.949 \pm 0.023$ | 100 | $0.944 \pm 0.019$ | 39 | $0.935 \pm 0.034$ | 67 |
| titanic | $0.800 \pm 0.017$ | 12 | $0.774 \pm 0.010$ | 100 | $0.789 \pm 0.036$ | 34 | $0.803^* \pm 0.017$ | 100 | $0.797 \pm 0.027$ | 82 | $0.796 \pm 0.031$ | 100 |
| twonorm | $0.946 \pm 0.005$ | 11 | $0.948 \pm 0.006$ | 100 | $0.932 \pm 0.006$ | 89 | $0.968^* \pm 0.002$ | 100 | $0.949 \pm 0.002$ | 100 | $0.950 \pm 0.002$ | 100 |
| waveform | $0.912 \pm 0.003$ | 9 | $0.917 \pm 0.005$ | 100 | $0.913 \pm 0.002$ | 93 | $0.926^* \pm 0.009$ | 100 | $0.920 \pm 0.011$ | 100 | $0.906 \pm 0.009$ | 87 |
| adult | $0.817^* \pm 0.002$ | 4 | $0.816 \pm 0.002$ | 100 | $0.816 \pm 0.002$ | 90 | $0.816 \pm 0.002$ | 96 | $0.816 \pm 0.002$ | 1 | $0.816 \pm 0.002$ | 96 |
| compas propublica | $0.666^* \pm 0.014$ | 76 | $0.665 \pm 0.014$ | 100 | $0.664 \pm 0.014$ | 86 | $0.666^* \pm 0.014$ | 100 | $0.665 \pm 0.013$ | 100 | $0.664 \pm 0.014$ | 100 |
| employment CA2018 | $0.746 \pm 0.003$ | 24 | $0.747 \pm 0.002$ | 100 | $0.743 \pm 0.002$ | 16 | $0.748 \pm 0.001$ | 76 | $0.749^* \pm 0.002$ | 100 | $0.742 \pm 0.002$ | 64 |
| employment TX2018 | $0.755 \pm 0.003$ | 12 | $0.755 \pm 0.008$ | 81 | $0.750 \pm 0.004$ | 11 | $0.757 \pm 0.003$ | 91 | $0.758^* \pm 0.004$ | 100 | $0.751 \pm 0.003$ | 60 |
| public coverage CA2018 | $0.710 \pm 0.005$ | 38 | $0.711 \pm 0.005$ | 100 | $0.696 \pm 0.007$ | 99 | $0.715^* \pm 0.005$ | 100 | $0.708 \pm 0.006$ | 100 | $0.697 \pm 0.003$ | 61 |
| public coverage TX2018 | $0.849 \pm 0.002$ | 11 | $0.851 \pm 0.003$ | 100 | $0.841 \pm 0.002$ | 97 | $0.855^* \pm 0.002$ | 100 | $0.849 \pm 0.002$ | 100 | $0.847 \pm 0.002$ | 79 |
| mushroom secondary | $0.999^* \pm 0.000$ | 14 | $0.999^* \pm 0.000$ | 100 | $0.999^* \pm 0.000$ | 78 | $0.999^* \pm 0.000$ | 86 | $0.993 \pm 0.004$ | 100 | $0.996 \pm 0.002$ | 86 |
| Mean/Median | 0.829/0.828 | 13.8/10 | 0.831/0.831 | 85.5/100 | 0.823/0.826 | 54.6/64.5 | 0.832*/0.829 | 97.5/100 | 0.825/0.833* | 83.6/100 | 0.815/0.831 | 80.5/87.5 |

## D.7 Experiments with Optimal Trees as Base Learners

We report in Tables 17, 18, 19, and 20 the per-dataset performances of the different considered boosting approaches, for optimal decision trees of depths 1, 3, 5, and 10 (respectively) trained with the `Blossom` algorithm. They complement the results provided in Section 7. Figure 17 shows the accuracy-sparsity trade-off of ODT ensembles in comparison to CART ensembles, aggregated over all datasets.

Table 17: Testing accuracy and number of non-zero weights for different boosting methods using **optimal decision trees** of **depth 1**, averaged over five seeds. **Bold** highlights the best accuracy among *all* methods, while a star* marks the best among *totally corrective* methods. The last row shows the mean and median for both statistics.

| Dataset | NM-Boost Acc. | Cols. | QRLP-Boost Acc. | Cols. | LP-Boost Acc. | Cols. | CG-Boost Acc. | Cols. | ERLP-Boost Acc. | Cols. | MD-Boost Acc. | Cols. | Adaboost Acc. | Cols. |
|---|---|---|---|---|---|---|---|---|---|---|---|---|---|---|
| banana | $\mathbf{0.625}^* \pm 0.016$ | 6 | $0.620 \pm 0.016$ | 9 | $\mathbf{0.625}^* \pm 0.016$ | 1 | $\mathbf{0.625}^* \pm 0.016$ | 100 | $0.620 \pm 0.016$ | 12 | $0.620 \pm 0.016$ | 11 | $\mathbf{0.625} \pm 0.016$ | 100 |
| breast cancer | $\mathbf{0.808}^* \pm 0.065$ | 9 | $0.796 \pm 0.069$ | 35 | $0.789 \pm 0.070$ | 4 | $0.789 \pm 0.044$ | 100 | $\mathbf{0.808}^* \pm 0.055$ | 25 | $0.800 \pm 0.068$ | 18 | $0.789 \pm 0.038$ | 100 |
| diabetes | $0.769 \pm 0.022$ | 5 | $0.758 \pm 0.032$ | 80 | $0.749 \pm 0.024$ | 15 | $0.742 \pm 0.035$ | 100 | $0.765 \pm 0.033$ | 44 | $\mathbf{0.775}^* \pm 0.015$ | 17 | $0.773 \pm 0.026$ | 100 |
| german credit | $0.762 \pm 0.022$ | 7 | $\mathbf{0.811}^* \pm 0.022$ | 80 | $0.797 \pm 0.021$ | 41 | $0.800 \pm 0.024$ | 100 | $0.805 \pm 0.021$ | 48 | $0.775 \pm 0.028$ | 18 | $0.808 \pm 0.029$ | 100 |
| heart | $0.848 \pm 0.052$ | 26 | $0.830 \pm 0.022$ | 27 | $0.815 \pm 0.033$ | 14 | $\mathbf{0.870}^* \pm 0.012$ | 100 | $0.819 \pm 0.022$ | 18 | $0.815 \pm 0.029$ | 13 | $0.807 \pm 0.025$ | 100 |
| image | $\mathbf{0.851}^* \pm 0.013$ | 19 | $0.832 \pm 0.013$ | 36 | $0.847 \pm 0.009$ | 20 | $0.850 \pm 0.013$ | 100 | $0.826 \pm 0.012$ | 23 | $0.820 \pm 0.013$ | 15 | $0.824 \pm 0.016$ | 100 |
| ringnorm | $\mathbf{0.930}^* \pm 0.004$ | 55 | $0.910 \pm 0.004$ | 40 | $\mathbf{0.930}^* \pm 0.004$ | 51 | $\mathbf{0.930}^* \pm 0.002$ | 100 | $0.891 \pm 0.002$ | 27 | $0.889 \pm 0.002$ | 22 | $0.911 \pm 0.003$ | 100 |
| solar flare | $0.703 \pm 0.047$ | 7 | $0.710 \pm 0.077$ | 21 | $0.697 \pm 0.059$ | 6 | $\mathbf{0.724}^* \pm 0.076$ | 100 | $0.717 \pm 0.077$ | 16 | $0.703 \pm 0.083$ | 9 | $0.662 \pm 0.096$ | 100 |
| splice | $0.943 \pm 0.009$ | 55 | $0.946 \pm 0.011$ | 64 | $\mathbf{0.949}^* \pm 0.008$ | 43 | $0.944 \pm 0.003$ | 100 | $0.938 \pm 0.013$ | 28 | $0.930 \pm 0.015$ | 10 | $0.943 \pm 0.009$ | 100 |
| thyroid | $0.921 \pm 0.024$ | 3 | $\mathbf{0.944}^* \pm 0.032$ | 9 | $\mathbf{0.944}^* \pm 0.032$ | 3 | $\mathbf{0.944}^* \pm 0.032$ | 100 | $\mathbf{0.944}^* \pm 0.032$ | 5 | $0.912 \pm 0.027$ | 5 | $0.940 \pm 0.024$ | 100 |
| titanic | $0.801 \pm 0.027$ | 8 | $0.803 \pm 0.022$ | 57 | $0.799 \pm 0.023$ | 3 | $0.806^* \pm 0.021$ | 100 | $0.797 \pm 0.028$ | 33 | $0.794 \pm 0.028$ | 17 | $\mathbf{0.807} \pm 0.029$ | 100 |
| twonorm | $\mathbf{0.961}^* \pm 0.003$ | 61 | $0.941 \pm 0.004$ | 72 | $0.959 \pm 0.004$ | 60 | $\mathbf{0.961}^* \pm 0.003$ | 100 | $0.927 \pm 0.003$ | 36 | $0.922 \pm 0.002$ | 23 | $0.949 \pm 0.004$ | 100 |
| waveform | $\mathbf{0.890}^* \pm 0.007$ | 57 | $0.888 \pm 0.004$ | 82 | $\mathbf{0.890}^* \pm 0.007$ | 52 | $\mathbf{0.890}^* \pm 0.006$ | 100 | $0.878 \pm 0.006$ | 36 | $0.867 \pm 0.007$ | 21 | $0.885 \pm 0.002$ | 100 |
| adult | $0.814 \pm 0.003$ | 17 | $0.816^* \pm 0.002$ | 25 | $0.814 \pm 0.003$ | 14 | $0.814 \pm 0.003$ | 100 | $0.816^* \pm 0.002$ | 27 | $0.812 \pm 0.002$ | 15 | $0.815 \pm 0.002$ | 100 |
| compas propublica | $0.659 \pm 0.019$ | 6 | $0.671^* \pm 0.018$ | 14 | $0.660 \pm 0.013$ | 6 | $0.660 \pm 0.013$ | 100 | $0.671^* \pm 0.018$ | 23 | $0.671^* \pm 0.018$ | 15 | $0.670 \pm 0.017$ | 100 |
| employment CA2018 | $0.740 \pm 0.003$ | 18 | $0.747^* \pm 0.002$ | 70 | $0.740 \pm 0.003$ | 19 | $0.740 \pm 0.003$ | 100 | $0.746 \pm 0.002$ | 64 | $0.737 \pm 0.002$ | 28 | $0.738 \pm 0.002$ | 100 |
| employment TX2018 | $0.744 \pm 0.003$ | 21 | $0.755^* \pm 0.004$ | 75 | $0.744 \pm 0.003$ | 19 | $0.744 \pm 0.003$ | 100 | $0.752 \pm 0.002$ | 61 | $0.745 \pm 0.003$ | 29 | $0.740 \pm 0.002$ | 100 |
| public coverage CA2018 | $0.680 \pm 0.006$ | 7 | $0.704^* \pm 0.003$ | 63 | $0.680 \pm 0.005$ | 3 | $0.680 \pm 0.006$ | 100 | $0.704^* \pm 0.003$ | 75 | $0.704^* \pm 0.003$ | 53 | $0.700 \pm 0.005$ | 100 |
| public coverage TX2018 | $0.831 \pm 0.010$ | 14 | $0.850^* \pm 0.003$ | 100 | $0.829 \pm 0.002$ | 3 | $0.829 \pm 0.002$ | 100 | $0.849 \pm 0.002$ | 75 | $0.847 \pm 0.003$ | 39 | $0.843 \pm 0.003$ | 100 |
| mushroom secondary | $0.816 \pm 0.005$ | 63 | $0.785 \pm 0.004$ | 69 | $0.817^* \pm 0.004$ | 62 | $0.817^* \pm 0.004$ | 100 | $0.771 \pm 0.001$ | 53 | $0.740 \pm 0.007$ | 38 | $0.754 \pm 0.002$ | 100 |
| Mean/Median | 0.805 / **0.811*** | 23.2 / 15.5 | 0.806 / 0.807 | 51.4 / 60.0 | 0.804 / 0.806 | 21.9 / 14.5 | **0.808*** / 0.810 | 100.0 / 100.0 | 0.802 / 0.806 | 36.5 / 30.5 | 0.794 / 0.797 | 20.8 / 17.5 | 0.799 / 0.807 | 100 / 100 |

Table 18: Testing accuracy and number of non-zero weights for different boosting methods using **optimal decision trees** of **depth 3**, averaged over five seeds. **Bold** highlights the best accuracy among *all* methods, while a star* marks the best among *totally corrective* methods. The last row shows the mean and median for both statistics.

| | NM-Boost | | QRLP-Boost | | LP-Boost | | CG-Boost | | ERLP-Boost | | MD-Boost | | Adaboost | |
|---|---|---|---|---|---|---|---|---|---|---|---|---|---|---|
| Dataset | Acc. | Cols. | Acc. | Cols. | Acc. | Cols. | Acc. | Cols. | Acc. | Cols. | Acc. | Cols. | Acc. | Cols. |
| banana | **0.670***±0.014 | 14 | **0.670***±0.014 | 94 | **0.670***±0.014 | 11 | **0.670***±0.014 | 100 | **0.670***±0.014 | 97 | **0.670***±0.014 | 24 | **0.670**±0.014 | 100 |
| breast cancer | 0.811±0.049 | 11 | 0.815±0.022 | 28 | 0.804±0.009 | 26 | 0.800±0.026 | 100 | **0.823***±0.033 | 16 | **0.823***±0.051 | 10 | **0.823**±0.019 | 100 |
| diabetes | 0.749±0.013 | 11 | 0.726±0.009 | 61 | 0.739±0.011 | 32 | 0.738±0.053 | 100 | 0.743±0.017 | 35 | 0.751*±0.007 | 26 | **0.769**±0.019 | 100 |
| german credit | 0.850±0.024 | 58 | 0.832±0.027 | 56 | 0.843±0.039 | 73 | 0.863*±0.022 | 100 | 0.819±0.030 | 32 | 0.810±0.022 | 16 | **0.867**±0.021 | 100 |
| heart | 0.881*±0.015 | 8 | 0.863±0.038 | 22 | 0.870±0.035 | 34 | 0.874±0.038 | 100 | 0.848±0.049 | 17 | 0.841±0.036 | 20 | **0.889**±0.023 | 100 |
| image | 0.942±0.009 | 36 | 0.937±0.011 | 42 | 0.942±0.013 | 47 | **0.943***±0.011 | 100 | 0.901±0.010 | 24 | 0.852±0.048 | 16 | **0.943**±0.011 | 100 |
| ringnorm | **0.933***±0.003 | 68 | 0.923±0.004 | 100 | 0.931±0.005 | 80 | 0.930±0.003 | 100 | 0.912±0.004 | 67 | 0.901±0.002 | 35 | 0.916±0.003 | 100 |
| solar flare | 0.662±0.034 | 10 | 0.593±0.067 | 28 | 0.662±0.103 | 10 | **0.703***±0.071 | 100 | 0.655±0.065 | 21 | 0.690±0.093 | 12 | 0.697±0.051 | 100 |
| splice | 0.971±0.007 | 43 | 0.975±0.007 | 84 | 0.976*±0.007 | 87 | 0.976*±0.005 | 100 | 0.967±0.011 | 55 | 0.959±0.010 | 52 | **0.979**±0.002 | 100 |
| thyroid | 0.940±0.035 | 2 | 0.944±0.024 | 21 | 0.953*±0.025 | 1 | 0.930±0.047 | 100 | 0.940±0.035 | 6 | 0.949±0.031 | 10 | 0.944±0.032 | 100 |
| titanic | 0.813±0.028 | 14 | 0.803±0.024 | 47 | 0.809±0.026 | 24 | 0.803±0.024 | 100 | 0.804±0.027 | 25 | 0.817*±0.028 | 14 | 0.812±0.026 | 100 |
| twonorm | 0.958±0.004 | 92 | 0.953±0.005 | 81 | 0.960*±0.006 | 83 | 0.958±0.002 | 100 | 0.946±0.006 | 42 | 0.929±0.005 | 49 | 0.954±0.004 | 100 |
| waveform | 0.895±0.005 | 49 | 0.894±0.006 | 85 | 0.901±0.001 | 87 | **0.904***±0.008 | 100 | 0.887±0.007 | 31 | 0.880±0.009 | 27 | 0.884±0.008 | 100 |
| adult | 0.809±0.002 | 4 | 0.819±0.002 | 100 | 0.818±0.001 | 86 | 0.816±0.002 | 92 | **0.820***±0.001 | 100 | 0.819±0.002 | 59 | 0.819±0.001 | 100 |
| compas propublica | 0.665±0.010 | 35 | 0.666±0.014 | 79 | **0.670***±0.014 | 46 | 0.667±0.013 | 100 | 0.666±0.014 | 100 | 0.667±0.016 | 42 | 0.668±0.015 | 100 |
| employment CA2018 | 0.736±0.002 | 14 | **0.749***±0.001 | 100 | 0.737±0.002 | 24 | 0.739±0.003 | 41 | **0.749***±0.002 | 100 | 0.735±0.002 | 21 | 0.744±0.002 | 100 |
| employment TX2018 | 0.745±0.003 | 34 | 0.758±0.005 | 100 | 0.747±0.003 | 37 | 0.748±0.002 | 82 | **0.759***±0.005 | 92 | 0.738±0.003 | 18 | 0.754±0.004 | 100 |
| public coverage CA2018 | 0.704±0.005 | 78 | 0.709±0.004 | 100 | 0.703±0.007 | 98 | 0.705±0.007 | 100 | **0.712***±0.003 | 100 | 0.707±0.004 | 90 | 0.711±0.002 | 100 |
| public coverage TX2018 | 0.850±0.002 | 2 | 0.852±0.003 | 100 | 0.850±0.002 | 1 | 0.850±0.001 | 100 | **0.853***±0.002 | 85 | 0.852±0.003 | 35 | 0.851±0.003 | 100 |
| mushroom secondary | 0.968±0.008 | 58 | 0.936±0.010 | 70 | 0.979±0.008 | 64 | **0.982***±0.005 | 66 | 0.875±0.006 | 55 | 0.831±0.035 | 23 | 0.809±0.011 | 100 |
| Mean/Median | 0.828 / 0.831 | 32.0 / 24.0 | 0.821 / 0.825 | 69.9 / 80.0 | 0.828 / 0.831 | 47.5 / 41.5 | **0.830*** / **0.833*** | 94.0 / 100.0 | 0.817 / 0.821 | 55.0 / 48.5 | 0.811 / 0.821 | 29.9 / 23.5 | 0.825 / 0.821 | 100 / 100 |

Table 19: Testing accuracy and number of non-zero weights for different boosting methods using **optimal decision trees** of **depth 5**, averaged over five seeds. **Bold** highlights the best accuracy among *all* methods, while a star* marks the best among *totally corrective* methods. The last row shows the mean and median for both statistics.

| | NM-Boost | | QRLP-Boost | | LP-Boost | | CG-Boost | | ERLP-Boost | | MD-Boost | | Adaboost | |
|---|---|---|---|---|---|---|---|---|---|---|---|---|---|---|
| Dataset | Acc. | Cols. | Acc. | Cols. | Acc. | Cols. | Acc. | Cols. | Acc. | Cols. | Acc. | Cols. | Acc. | Cols. |
| banana | **0.670***±0.014 | 12 | **0.670***±0.014 | 2 | **0.670***±0.014 | 1 | **0.670***±0.014 | 100 | **0.670***±0.014 | 4 | **0.670***±0.014 | 40 | **0.670**±0.014 | 100 |
| breast cancer | 0.838±0.019 | 10 | 0.834±0.025 | 22 | 0.808±0.038 | 32 | **0.845***±0.038 | 100 | 0.804±0.039 | 12 | 0.823±0.039 | 9 | 0.819±0.042 | 100 |
| diabetes | 0.755*±0.033 | 29 | 0.742±0.019 | 33 | 0.739±0.020 | 84 | 0.744±0.021 | 100 | 0.740±0.013 | 24 | 0.742±0.032 | 16 | **0.761**±0.017 | 100 |
| german credit | **0.895***±0.015 | 25 | 0.868±0.022 | 31 | 0.876±0.028 | 73 | 0.871±0.016 | 100 | 0.854±0.011 | 15 | 0.842±0.012 | 18 | 0.893±0.016 | 100 |
| heart | 0.874±0.022 | 4 | 0.870±0.026 | 25 | 0.856±0.040 | 40 | 0.881*±0.009 | 100 | 0.859±0.030 | 17 | 0.844±0.019 | 19 | **0.904**±0.014 | 100 |
| image | 0.952*±0.005 | 24 | 0.947±0.007 | 31 | 0.950±0.008 | 46 | 0.952*±0.003 | 100 | 0.936±0.012 | 17 | 0.911±0.020 | 14 | **0.954**±0.005 | 100 |
| ringnorm | **0.936***±0.004 | 88 | 0.926±0.004 | 98 | 0.935±0.006 | 80 | 0.935±0.003 | 100 | 0.918±0.003 | 64 | 0.892±0.005 | 25 | 0.921±0.006 | 100 |
| solar flare | **0.690***±0.049 | 2 | 0.628±0.086 | 72 | 0.683±0.024 | 72 | 0.662±0.063 | 100 | 0.662±0.063 | 16 | 0.662±0.101 | 12 | 0.683±0.026 | 100 |
| splice | 0.977±0.007 | 24 | 0.979±0.005 | 70 | 0.980*±0.007 | 88 | 0.980*±0.008 | 100 | 0.974±0.006 | 38 | 0.973±0.007 | 31 | **0.984**±0.004 | 100 |
| thyroid | 0.953±0.025 | 2 | **0.958***±0.023 | 19 | 0.953±0.021 | 1 | 0.944±0.035 | 100 | 0.949±0.027 | 4 | 0.935±0.027 | 16 | 0.944±0.032 | 100 |
| titanic | 0.811±0.028 | 8 | 0.812*±0.028 | 39 | 0.808±0.031 | 19 | 0.799±0.036 | 100 | 0.802±0.026 | 19 | 0.806±0.034 | 16 | **0.816**±0.021 | 100 |
| twonorm | 0.961*±0.004 | 58 | 0.956±0.003 | 54 | 0.958±0.003 | 76 | 0.960±0.004 | 100 | 0.949±0.002 | 24 | 0.947±0.007 | 41 | **0.963**±0.004 | 100 |
| waveform | 0.923*±0.010 | 76 | 0.917±0.009 | 37 | 0.916±0.012 | 97 | 0.916±0.012 | 100 | 0.902±0.011 | 26 | 0.892±0.006 | 24 | **0.925**±0.014 | 100 |
| adult | 0.817±0.002 | 98 | 0.817±0.002 | 92 | 0.817±0.002 | 91 | 0.816±0.002 | 100 | 0.817±0.002 | 100 | **0.820***±0.002 | 77 | 0.819±0.002 | 100 |
| compas propublica | 0.668±0.013 | 31 | 0.666±0.009 | 9 | **0.670***±0.014 | 32 | **0.670***±0.013 | 100 | 0.667±0.012 | 3 | 0.669±0.015 | 61 | 0.669±0.015 | 100 |
| employment CA2018 | 0.747±0.003 | 96 | **0.750***±0.002 | 100 | 0.749±0.001 | 98 | 0.749±0.003 | 100 | **0.750***±0.001 | 99 | 0.743±0.001 | 27 | 0.748±0.002 | 100 |
| employment TX2018 | 0.758±0.004 | 91 | 0.759±0.003 | 100 | 0.758±0.003 | 99 | 0.757±0.003 | 100 | **0.761***±0.005 | 86 | 0.750±0.003 | 26 | 0.756±0.005 | 100 |
| public coverage CA2018 | 0.700±0.005 | 15 | 0.704±0.002 | 100 | **0.714***±0.004 | 94 | 0.712±0.002 | 95 | 0.713±0.003 | 100 | 0.707±0.002 | 28 | 0.713±0.005 | 100 |
| public coverage TX2018 | 0.850±0.001 | 3 | **0.854***±0.001 | 100 | 0.852±0.002 | 99 | 0.848±0.002 | 100 | 0.853±0.002 | 62 | 0.850±0.001 | 17 | 0.851±0.003 | 100 |
| mushroom secondary | 0.998±0.000 | 56 | 0.982±0.004 | 45 | **0.999***±0.000 | 91 | **0.999***±0.000 | 90 | 0.937±0.011 | 24 | 0.891±0.033 | 23 | 0.997±0.001 | 100 |
| Mean/Median | **0.839*** / 0.844 | 37.6 / 24.5 | 0.832 / 0.844 | 54.0 / 42.0 | 0.835 / 0.835 | 62.1 / 78.0 | 0.834 / **0.847*** | 99.2 / 100.0 | 0.826 / 0.835 | 37.7 / 24.0 | 0.818 / 0.833 | 27.0 / 23.5 | 0.839 / 0.835 | 100 / 100 |

Table 20: Testing accuracy and number of non-zero weights for different boosting methods using **optimal decision trees** of **depth 10**, averaged over five seeds. **Bold** highlights the best accuracy among *all* methods, while a star* marks the best among *totally corrective* methods. The last row shows the mean and median for both statistics.

| | NM-Boost | | QRLP-Boost | | LP-Boost | | CG-Boost | | ERLP-Boost | | MD-Boost | | Adaboost | |
|---|---|---|---|---|---|---|---|---|---|---|---|---|---|---|
| Dataset | Acc. | Cols. | Acc. | Cols. | Acc. | Cols. | Acc. | Cols. | Acc. | Cols. | Acc. | Cols. | Acc. | Cols. |
| banana | $\mathbf{0.670}^* \pm 0.014$ | 14 | $\mathbf{0.670}^* \pm 0.014$ | 2 | $\mathbf{0.670}^* \pm 0.014$ | 1 | $\mathbf{0.670}^* \pm 0.014$ | 100 | $\mathbf{0.670}^* \pm 0.014$ | 2 | $\mathbf{0.670}^* \pm 0.014$ | 36 | $0.670 \pm 0.014$ | 100 |
| breast cancer | $\mathbf{0.838}^* \pm 0.026$ | 3 | $0.830 \pm 0.043$ | 55 | $0.830 \pm 0.045$ | 26 | $0.823 \pm 0.026$ | 100 | $0.811 \pm 0.027$ | 7 | $0.834 \pm 0.050$ | 36 | $0.826 \pm 0.028$ | 100 |
| diabetes | $0.713 \pm 0.019$ | 3 | $0.700 \pm 0.021$ | 30 | $0.682 \pm 0.041$ | 51 | $0.748^* \pm 0.019$ | 100 | $0.709 \pm 0.010$ | 10 | $0.717 \pm 0.030$ | 56 | $\mathbf{0.760} \pm 0.013$ | 100 |
| german credit | $0.861 \pm 0.021$ | 4 | $\mathbf{0.888}^* \pm 0.019$ | 31 | $0.886 \pm 0.022$ | 75 | $0.877 \pm 0.021$ | 100 | $0.871 \pm 0.040$ | 11 | $0.876 \pm 0.025$ | 18 | $0.881 \pm 0.023$ | 100 |
| heart | $0.856 \pm 0.034$ | 1 | $0.859^* \pm 0.032$ | 1 | $0.844 \pm 0.030$ | 1 | $0.841 \pm 0.025$ | 100 | $0.833 \pm 0.029$ | 1 | $0.844 \pm 0.030$ | 1 | $0.844 \pm 0.030$ | 1 |
| image | $0.949 \pm 0.009$ | 13 | $0.950 \pm 0.004$ | 22 | $0.952 \pm 0.010$ | 60 | $0.953^* \pm 0.007$ | 100 | $0.952 \pm 0.009$ | 9 | $0.951 \pm 0.006$ | 16 | $\mathbf{0.954} \pm 0.008$ | 100 |
| ringnorm | $0.922 \pm 0.006$ | 27 | $0.918 \pm 0.011$ | 20 | $0.910 \pm 0.007$ | 97 | $0.931^* \pm 0.007$ | 100 | $0.922 \pm 0.003$ | 13 | $0.914 \pm 0.009$ | 15 | $\mathbf{0.934} \pm 0.005$ | 100 |
| solar flare | $0.669 \pm 0.047$ | 6 | $0.648 \pm 0.051$ | 99 | $0.641 \pm 0.071$ | 11 | $0.669 \pm 0.083$ | 100 | $0.641 \pm 0.056$ | 72 | $0.697^* \pm 0.040$ | 35 | $0.648 \pm 0.040$ | 100 |
| splice | $0.967 \pm 0.008$ | 2 | $0.967 \pm 0.009$ | 15 | $0.965 \pm 0.012$ | 75 | $\mathbf{0.979}^* \pm 0.004$ | 100 | $0.963 \pm 0.015$ | 9 | $0.971 \pm 0.006$ | 45 | $0.978 \pm 0.002$ | 100 |
| thyroid | $0.944 \pm 0.032$ | 2 | $0.958^* \pm 0.023$ | 19 | $0.953 \pm 0.021$ | 1 | $0.940 \pm 0.024$ | 100 | $0.949 \pm 0.027$ | 4 | $0.940 \pm 0.035$ | 9 | $0.944 \pm 0.032$ | 100 |
| titanic | $0.797 \pm 0.010$ | 16 | $0.790 \pm 0.018$ | 49 | $0.790 \pm 0.041$ | 26 | $0.797 \pm 0.021$ | 100 | $0.803^* \pm 0.025$ | 19 | $0.798 \pm 0.015$ | 25 | $0.802 \pm 0.008$ | 100 |
| twonorm | $0.947 \pm 0.005$ | 14 | $0.951 \pm 0.005$ | 21 | $0.939 \pm 0.008$ | 96 | $\mathbf{0.969}^* \pm 0.003$ | 100 | $0.953 \pm 0.004$ | 53 | $0.955 \pm 0.003$ | 41 | $0.965 \pm 0.003$ | 100 |
| waveform | $0.921 \pm 0.007$ | 25 | $0.918 \pm 0.007$ | 26 | $0.912 \pm 0.008$ | 96 | $0.923^* \pm 0.005$ | 100 | $0.920 \pm 0.012$ | 16 | $0.910 \pm 0.008$ | 17 | $\mathbf{0.925} \pm 0.008$ | 100 |
| adult | $0.815 \pm 0.004$ | 41 | $0.816 \pm 0.002$ | 1 | $0.815 \pm 0.004$ | 88 | $\mathbf{0.817}^* \pm 0.003$ | 74 | $0.816 \pm 0.002$ | 1 | $0.816 \pm 0.003$ | 81 | $0.816 \pm 0.002$ | 100 |
| compas propublica | $0.667^* \pm 0.013$ | 16 | $0.665 \pm 0.013$ | 100 | $0.666 \pm 0.016$ | 1 | $0.667^* \pm 0.013$ | 100 | $0.666 \pm 0.016$ | 3 | $0.664 \pm 0.010$ | 100 | $0.665 \pm 0.014$ | 100 |
| employment CA2018 | $0.746 \pm 0.002$ | 20 | $0.744 \pm 0.002$ | 100 | $0.744 \pm 0.001$ | 1 | $0.752^* \pm 0.001$ | 47 | $0.745 \pm 0.001$ | 58 | $0.748 \pm 0.001$ | 31 | $\mathbf{0.752} \pm 0.000$ | 100 |
| employment TX2018 | $0.752 \pm 0.003$ | 25 | $0.754 \pm 0.003$ | 96 | $0.752 \pm 0.004$ | 1 | $\mathbf{0.764}^* \pm 0.003$ | 99 | $0.755 \pm 0.006$ | 49 | $0.758 \pm 0.003$ | 17 | $0.758 \pm 0.003$ | 100 |
| public coverage CA2018 | $0.701 \pm 0.006$ | 23 | $0.706 \pm 0.006$ | 96 | $0.693 \pm 0.002$ | 88 | $\mathbf{0.721}^* \pm 0.003$ | 100 | $0.709 \pm 0.002$ | 68 | $0.710 \pm 0.004$ | 20 | $0.714 \pm 0.004$ | 100 |
| public coverage TX2018 | $0.851 \pm 0.001$ | 9 | $0.852 \pm 0.003$ | 54 | $0.847 \pm 0.003$ | 1 | $\mathbf{0.856}^* \pm 0.001$ | 89 | $0.849 \pm 0.002$ | 25 | $0.848 \pm 0.003$ | 13 | $0.854 \pm 0.002$ | 100 |
| mushroom secondary | $\mathbf{0.999}^* \pm 0.000$ | 24 | $0.998 \pm 0.000$ | 30 | $\mathbf{0.999}^* \pm 0.000$ | 76 | $\mathbf{0.999}^* \pm 0.000$ | 96 | $0.992 \pm 0.004$ | 14 | $0.989 \pm 0.004$ | 58 | $\mathbf{0.999} \pm 0.000$ | 100 |
| Mean/Median | $0.829 \, / \, \mathbf{0.845}^*$ | 14.4 / 14.0 | $0.829 \, / \, 0.841$ | 43.4 / 30.0 | $0.824 \, / \, 0.837$ | 43.6 / 38.5 | $\mathbf{0.835}^* \, / \, 0.832$ | 95.2 / 100.0 | $0.826 \, / \, 0.825$ | 22.2 / 12.0 | $0.831 \, / \, 0.839$ | 33.5 / 28.0 | $0.834 \, / \, 0.835$ | 95 / 100 |

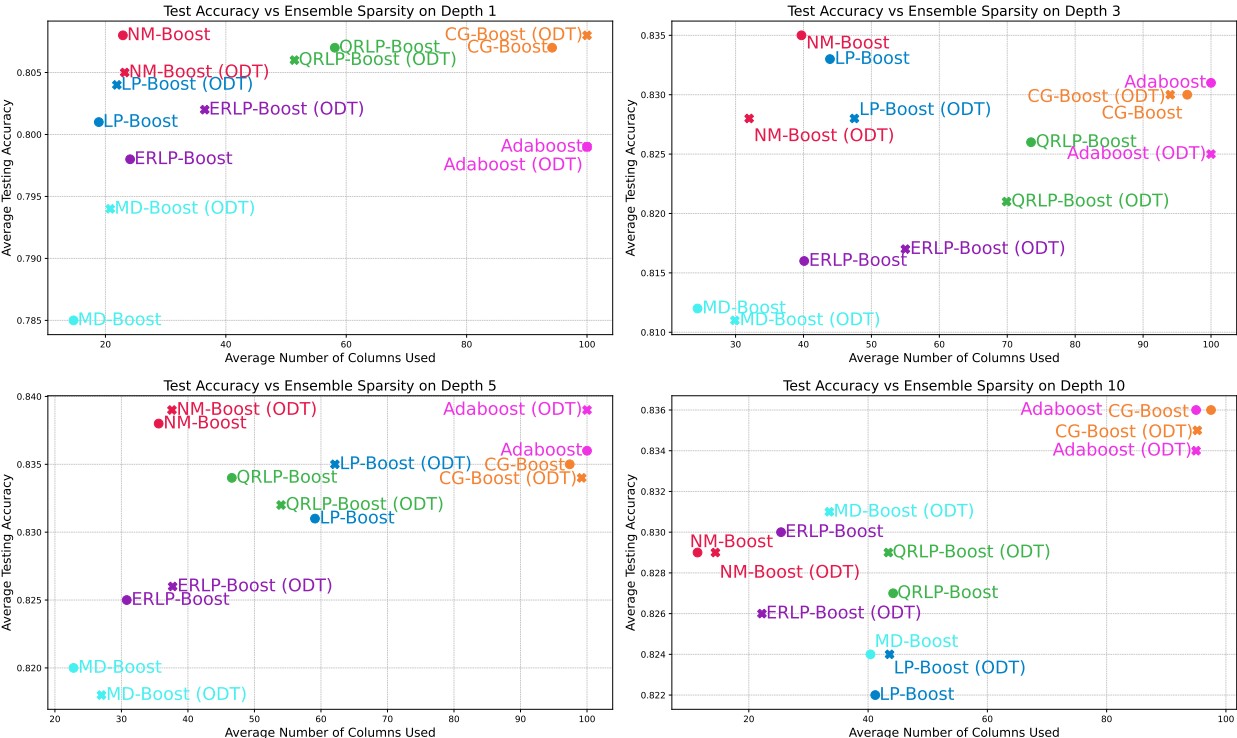

Figure 17: Average testing accuracy compared to average ensemble sparsity over **all datasets** for 1, 3, 5, and 10 depth CART trees or optimal decision trees (ODT).

