# OpenReview forum: "Boosting Revisited: Benchmarking and Advancing LP-Based Ensemble Methods"
_TMLR — Accepted by TMLR_

### Review · Reviewer_B18U · 2025-08-13

**Summary Of Contributions:**

This paper presents a comprehensive empirical study of totally corrective boosting methods based on linear programming formulations, which adjust the weights of all base learners at each iteration. It compares with six LP based methods, NM-Boost, QRLP-Boost, and AdaBoost, XGBoost, and LightGBM. It is evaluated on 20 diverse binary classification datasets.

The paper introduces NM-Boost and QRLP-Boost methods that perform well on shallow trees particularly.

**Audience:**

Yes

**Audience Explanation:**

The empirical study of LP-based boosting methods, the sparsity benefits, margin distributions, and comparisons to heuristics like XGBoost  will be interested to some audience.

However, more applications and discussions on multi-class problem are better.

**Broader Impact Concerns:**

The paper includes the adult dataset for testing, and the concern like bias on rase and gender will be raised. It is better to discuss the risks and mitigation strategies of them.

**Claims And Evidence:**

Yes

**Claims Explanation:**

The paper provides empirical study across 20 diverse datasets with results presented in multiple tables and figures for accuracy, sparsity, margin distributions, and hyperparameter sensitivity. Observations are shown.

However, the paper can be more clear and convicting to show the codes that generate the figures and results in the tables.

**Requested Changes:**

1. Lack of computational cost analysis
The paper does not quantify its computational complexity or runtime compared to heuristic methods like XGBoost and LightGBM.

2. Binary classification limited
The study focuses on binary classification only, limiting its applicability to multi-class or regression tasks, which are more common in applications.

3. The paper lacks discussions on efficient tuning strategies.

---

> ### Author Response · Authors · 2025-09-11
> **Response to reviewer B18U**
>
> Thank you for your comments and suggestions. We hereafter answer the key points raised in your review. We have also uploaded a revised version of our manuscript, in which all modifications appear in blue.
>
> > The paper can be more clear and convicting to show the codes that generate the figures and results in the tables.
>
> We have now shared an anonymized version of our code. It includes the code that has been used for our experiments and can be used to replicate our results, as well as a user-friendly Python library that enables users to easily fit an ensemble using an LP-based method of their choice (all studied methods have been implemented). Upon acceptance, we will also release all experiment result files, but we did not include those now due to data file size limitations.
>
> > 1. Lack of computational cost analysis The paper does not quantify its computational complexity or runtime compared to heuristic methods like XGBoost and LightGBM.
>
> Indeed, we did study and report computational times in the submitted paper, but this content was included in Appendix D.1. We agree that it is important to provide a clear analysis of computational times, and therefore, we have revised the paper to add more discussion of the computational times in the main text at the start of Section 5. We note here that heuristic methods are generally faster, but this comes at the cost of sparsity-accuracy tradeoffs. Totally corrective boosting approaches still achieve reasonable running times, typically on the order of only a few tens of minutes in the worst case. For the sake of brevity, the table with all reported CPU times remains in Appendix D.1.
>
> > 2. More applications and discussions on multi-class problem are better. [...] Binary classification limited The study focuses on binary classification only, limiting its applicability to multi-class or regression tasks, which are more common in applications.
>
> We appreciate the reviewer’s suggestion to extend our study to multi-class classification and regression tasks. These are interesting directions for future work. In this paper, however, we deliberately focused on the binary case for three reasons. First, all closely related studies have been designed for binary classification, and we wished to provide a fair and consistent comparison. Second, broadening the scope to multi-class or regression would substantially increase the size and complexity of the paper, potentially at the expense of readability. Finally, while a straightforward extension for multi-class via a one-vs-rest scheme is possible, this amounts to training a separate ensemble for each class and is therefore both methodologically limited and computationally inefficient. Developing a more efficient and principled multi-class or regression variant would require new formulations (for each studied method), which we regard as an important but separate line of research beyond the scope of the current work.
>
> > 3. The paper lacks discussions on efficient tuning strategies.
>
> We thank the reviewer for raising this point. The details of our tuning strategy are described in the paper (see the start of Section 5 and Appendix C), and we additionally analyzed hyperparameter sensitivity in Section 6.5. In the revision, we clarified our motivation by noting that, since each method involves only a single hyperparameter, an exhaustive sweep over a fixed set of values is both straightforward and effective to capture performance across the relevant range, as discussed at the start of Section 5.
>
> > The paper includes the adult dataset for testing, and the concern like bias on race and gender will be raised. It is better to discuss the risks and mitigation strategies of them.
>
> We thank the reviewer for this important comment. In the revision, we have added a Broader Impact statement at the end of the paper, where we explicitly acknowledge the fairness concerns associated with the adult and compas propublica datasets.

---

### Review · Reviewer_eVip · 2025-08-28

**Summary Of Contributions:**

This paper proposes a thorough evaluation of the performance of boosting approaches. Multiple boosting methods, including two newly introduced ones, are evaluated across diverse datasets and metrics.

**Additional Comments:**

Could you please elaborate on the role of $p_\text{neg}$ in (9)?

**Audience:**

Yes

**Audience Explanation:**

Despite the popularity of deep learning approaches, boosting methods are still considered among the best for modeling tabular data. This work provides a strong foundation for researchers interested in comparing their non-boosting approaches against boosting methods.

**Broader Impact Concerns:**

I didn't find any ethical concerns.

**Claims And Evidence:**

Yes

**Claims Explanation:**

The work provides a thorough analysis of boosting methods using various metrics. Despite the advancements in deep learning approaches, boosting methods remain among the best for modeling tabular data. This work provides an excellent foundation for researchers who are interested in comparing any non-boosting approaches with the benchmark boosting method presented here

**Requested Changes:**

I would request authors to provide the solutions of the two proposed LP problems in (9) and (14), given that LP typically has analytical solutions. If that's not feasible, please explain why.

---

> ### Author Response · Authors · 2025-09-11
> **Response to reviewer eVip**
>
> Thank you for your feedback.  We hereafter address the two key questions raised in your review.
> We have also uploaded a revised version of our manuscript, in which all modifications appear in blue.
>
> > 1. I would request authors to provide the solutions of the two proposed LP problems in (9) and (14), given that LP typically has analytical solutions. If that's not feasible, please explain why.
>
> Thank you for you suggestion. For our two formulations (9) and (14) there is no analytic solution. In general, while closed-form analytic solutions are desirable, they
> unfortunately do not exist for all linear programs, a universal analytic formula is not available except in special, separable cases. In response to your feedback, we now clarified the solution structure of our two formulations in the main text, see Section 4.1 and 4.2. While these characterizations aid interpretation, they do not yield closed-form optimizers nor simpler direct methods than solving the stated programs. Practically, both (9) and (14) are solved efficiently and reliably with a standard LP/QP solver (Gurobi), as confirmed by the reported running times (Appendix D.1).
>
> > 2. Could you please elaborate on the role of $\rho^{neg}$ in (9)?
>
> Thank you for pointing this out. The role of $\rho^{neg}$ was briefly discussed in Section 4.1, but indeed we realized that its contribution to the objective function could be discussed in deeper details. In short, $\rho^{neg}$ isolates the negative part of the margin of each example (with a small offset): $\rho_i^{neg} = 0$ if example $i$ is correctly classified, and $\rho_i^{neg}$ is the (negative) margin between the ensemble's prediction for misclassified example $i$ and the decision boundary otherwise. This way, the optimization problem can assign a stronger penalty to misclassified points. This ensures that NM-Boost not only increases the overall margin distribution but also explicitly reduces the number and severity of negative margins (misclassified samples). We have revised the text in Section 4.1 to better discuss those facts.

---

### Review · Reviewer_Gecy · 2025-08-29

**Summary Of Contributions:**

Contributions:
1. Comprehensive Benchmarking: Perform the large-scale comparison of six LP-based boosting formulations against leading heuristic boosting methods across 20 datasets.
2. New Methods: two novel formulations, NM-Boost and QRLP-Boost.
3. Base Learner Study: Examine the impact of different base learners within boosting frameworks, including standard CART trees and optimal decision trees.
4. Empirical Analyses: Provide in-depth evaluation not just of accuracy, but also sparsity of ensembles, margin distributions, anytime performance, and sensitivity to hyperparameters.


Key Strength:
1. From Contribution 1&3&4, the paper is very comprehensive and comparative. In addition, the Experiment results are also very detailed and convincible.
2. The intuition and introduction of two new methods are very clear and easy to understand. The formula is also very detailed.

Weaknesses:
1. The full code is only "available online upon acceptance". What about using an anonymous way for now? Otherwise, it is hard to justify reproducibility.
2. May discuss more about choosing hyperparameters. In Fig 6, except for QRLP-Boost, the best hyper-parameters for other methods are extreme (largest/smalles among all 10 choices).
3. (Minor) Some figures are hard to identify. For example, in Fig 6, there are too many lines with similar colors. In Fig 9, some outliers make the y-axis very large such that the box is very small.

**Audience:**

Yes

**Audience Explanation:**

And I believe some TMLR's audience should be very interested in LP-Based Ensemble Methods. I believe Boosting remains one of the most widely used ensemble methods in ML. And in this paper, there are comprehensive comparisons to strong baselines.

**Broader Impact Concerns:**

None.

**Claims And Evidence:**

Yes

**Claims Explanation:**

As long as authors can clarify the weaknesses, the conclusion should be sound as it is reproducible and has a very clear evedence.

**Requested Changes:**

See Weaknesses.
1. Reproducibility is important, especially for this paper which examines thorough experiments.
2. Need to explain why for some methods, the best hyperparameter is the "extreme" one among all candidates.
3. (Minor) Some figures could be easier for readers to get the key message.

---

> ### Author Response · Authors · 2025-09-11
> **Response to reviewer Gecy**
>
> Thank you for your comments and positive evaluation of our work. We hereafter address the specific points raised in your review. We have also uploaded a revised version of our manuscript, in which all modifications appear in blue.
>
> > 1. The full code is only "available online upon acceptance". What about using an anonymous way for now? Otherwise, it is hard to justify reproducibility.
>
> Yes for sure, we have now shared an anonymized repository that includes the full code used in our experiments, permitting exact replication of all our results. In addition, we provide a user-friendly Python library that enables users to easily fit an ensemble using an LP-based method of their choice (all studied methods have been implemented). Upon acceptance, we will also release all experiment result files, but we did not include those now due to data file size limitations.
>
> > 2. May discuss more about choosing hyperparameters. In Fig 6, except for QRLP-Boost, the best hyper-parameters for other methods are extreme (largest/smalles among all 10 choices). [...] Need to explain why for some methods, the best hyperparameter is the "extreme" one among all candidates.
>
> We thank the reviewer for this observation. For several methods, the best hyperparameter values indeed lie at the extremes of the tested ranges. These ranges were chosen to match those recommended in the original papers, and they were further refined based on preliminary experiments. In those experiments, we observed that the margin distribution converged at the lower and upper ends of the chosen range. Further decreasing or increasing the parameter did not yield meaningful improvements in test accuracy. Moreover, we deliberately selected this search range to avoid excessively small or large hyperparameter values, which can introduce numerical stability issues. To reflect your comment, we included a clarification regarding hyperparameter tuning at the start of Section 5 in the paper, and provided more details on tuning in Appendix C.
>
> > 3. (Minor) Some figures are hard to identify. For example, in Fig 6, there are too many lines with similar colors. In Fig 9, some outliers make the y-axis very large such that the box is very small. [...] (Minor) Some figures could be easier for readers to get the key message.
>
> We adapted Figure 6 and now use the viridis colormap, which should be better distinguishable and colorblind-friendly. To still distinguish between the three methods (NM-Boost, QRLP-Boost and AdaBoost) we mark the edge of the legend in the respective color that belongs to the respective methods in the rest of the paper. Additionally, we filtered out a few outliers in Figure 9, enhancing readability as the y-axis now runs from -10\% to 10\%.

---

> > ### Comment · Reviewer_Gecy · 2025-09-11
> > **Thank the authors for the response**
> >
> > The rebuttal and additional code sufficiently address my concerns. I have no further issues.

---

### Author Response · Authors · 2025-09-11
**We thank the reviewers for their constructive comments and feedback**

We want to thank all reviewers for their evaluation of our work and the overall positive comments and feedback. We have answered each review individually, providing additional clarifications regarding important questions or limitations. Furthermore, we have uploaded the revised version of our paper, in which all modifications appear in blue. We have also uploaded the anonymized source code used for our experiments, which permits to replicate our results, along with a user-friendly Python library that enables users to easily fit an ensemble using an LP-based method of their choice (all studied methods have been implemented).

---

### Comment · Editors_In_Chief · 2025-12-18

On December 18, at the request of the authors, the EiCs replaced the camera ready PDF. Details and justification of changes are described by the authors below. Changes were checked by the AE and the revised PDF was found to still meet the TMLR criteria.

*The error originates from an external library called Blossom, which we used to obtain optimal decision trees (ODTs). The library was found to contain an implementation mistake in its handling of sample weights. Importantly, Blossom was used for only a small portion of our results (2 out of 18 observations). After becoming aware of this issue, we reran all affected experiments and obtained revised results.*

*Previously, our results suggested that the use of ODTs negatively affected boosting performance compared to heuristic CART trees. With the corrected implementation, we now observe that ODT-based ensembles perform similarly to CART-based ensembles, and in some cases may outperform them, depending on the dataset.*

*The revision includes:*
- *Section 7: Textual changes, updated Figure 10 and 11*
- *Section 8 (Conclusion): Textual change*
- *Appendix D.7: four Tables (Table 17-20) updated and added a figure (Figure 17) for better interpretation of sparsity*
- *We added a post-publication change statement after the acknowledgements explaining the issue and its cause*

*Out of 16 figures and 20 tables in the paper, 2 figures and 4 tables are affected.*

---

### Decision · Action_Editor_76D9 · 2025-10-17

**Recommendation:** Accept as is

**Audience:**

Yes

**Audience Explanation:**

As stated by the reviewers, boosting methods are widely used for binary classification, especially on tabular data. Therefore, this paper will be of interest to (at least) some members of the machine learning community.

**Claims And Evidence:**

Yes

**Claims Explanation:**

This goal of this paper is to provide a comprehensive evaluation of totally corrective boosting methods versus heuristic methods (e.g. XGBoost) on binary classification tasks. The authors test numerous LP-based methods, while also introducing two of their own, and find that they generally outperform the heuristic methods.

The evaluation, though limited to binary classification problems, is quite comprehensive and well designed. The authors test over many datasets, and include tests beyond basic accuracy (anytime performance, hyperparameter sensitivity, etc.). The authors have included high quality code to reproduce experiments and further benchmark, and this code will be made available upon acceptance.

The experiments performed as part of this paper provide accurate and sufficient evidence for the benchmarking goal of this paper.